# Short-term occupations at high elevation during the Middle Paleolithic at Kalavan 2 (Republic of Armenia)

Ariel Malinsky-Buller[1,2]*, Philip Glauberman[3,4], Vincent Ollivier[5,6], Tobias Lauer[7], Rhys Timms[8], Ellery Frahm[9], Alexander Brittingham[10], Benno Triller[11], Lutz Kindler[1], Monika V. Knul[12], Masha Krakovsky[13], Sebastian Joannin[14], Michael T. Hren[15,16], Olivier Bellier[5,17], Alexander A. Clark[8], Simon P. E. Blockley[8], Dimidry Arakelyan[18], João Marreiros[19,20], Eduardo Paixaco[19,20], Ivan Calandra[19], Robert Ghukasyan[2], David Nora[20], Nadav Nir[21], Ani Adigyozalyan[3], Hayk Haydosyan[3], Boris Gasparyan[3]

1 MONREPOS, Archaeological Research Centre and Museum for Human Behavioural Evolution, Neuwied, Germany, 2 The Institute of Archaeology, The Hebrew University of Jerusalem, Jerusalem, Israel, 3 Institute of Archaeology and Ethnography, National Academy of Sciences of the Republic of Armenia, Yerevan, Armenia, 4 Xi'an Jiaotong-Liverpool University, Suzhou, China, 5 CNRS, Aix Marseille Univ, Minist Culture, LAMPEA, Aix-en-Provence, France, 6 Aix Marseille Univ, CNRS, FR ECCOREV, Aix-en-Provence, France, 7 Department of Human Evolution, Max Planck Institute for Evolutionary Anthropology, Leipzig, Germany, 8 Centre for Quaternary Research, Department of Geography, Royal Holloway, University of London, Egham, Surrey, United Kingdom, 9 Yale Initiative for the Study of Ancient Pyrotechnology, Council on Archaeological Studies, Department of Anthropology, Yale University, New Haven, Connecticut, United States of America, 10 Department of Anthropology, Old World Archaeology, University of Connecticut, Storrs, Connecticut, United States of America, 11 Institute of Ancient Studies, Johannes Gutenberg-University, Mainz, Germany, 12 Department of Archaeology, Anthropology and Geography, University of Winchester, Winchester, United Kingdom, 13 Israel Antiquities Authority, Jerusalem, Israel, 14 Institut des Sciences de l'Evolution de Montpellier, Université de Montpellier, CNRS, EPHE, Montpellier, France, 15 Department of Geoscience, University of Connecticut, Storrs, Connecticut, United States of America, 16 Department of Chemistry, University of Connecticut, Storrs, Connecticut, United States of America, 17 Aix Marseille University, CNRS, IRD, INRAE, Coll France, CEREGE, Aix-en-Provence, France, 18 Institute of Geological Sciences, National Academy of Sciences of the Republic of Armenia, Yerevan, Armenia, 19 TraCEr, Laboratory for Traceology and Controlled Experiments at MONREPOS Archaeological Research Centre and Museum for Human Behavioural Evolution, RGZM, Neuwied, Germany, 20 ICArEHB–Interdisciplinary Center for Archaeology and Evolution of Human Behaviour, Universidade do Algarve, Faro, Portugal, 21 Department of Earth Sciences, Physical Geography, Freie Universität Berlin, Berlin, Germany

* malinsky@rgzm.de

**Data Availability Statement:** All relevant data are within the paper and its Supporting Information files.

## Abstract

The Armenian highlands encompasses rugged and environmentally diverse landscapes and is characterized by a mosaic of distinct ecological niches and large temperature gradients. Strong seasonal fluctuations in resource availability along topographic gradients likely prompted Pleistocene hominin groups to adapt by adjusting their mobility strategies. However, the role that elevated landscapes played in hunter-gatherer settlement systems during the Late Pleistocene (Middle Palaeolithic [MP]) remains poorly understood. At 1640 m above sea level, the MP site of Kalavan 2 (Armenia) is ideally positioned for testing hypotheses involving elevation-dependent seasonal mobility and subsistence strategies. Renewed excavations at Kalavan 2 exposed three main occupation horizons and ten additional low densities lithic and faunal assemblages. The results provide a new chronological, stratigraphical, and paleoenvironmental framework for hominin behaviors between ca. 60 to 45

**Funding:** For the essential financial and similar support, we gratefully thank the funding agencies of The Gerda Henkel Stiftung grant n. AZ 10_V_17 and n. AZ 23/F/19, the Leakey Foundation, and the Gfoeller Renaissance Foundation of USA. RT, AC, and SB would like to acknowledge support from the Leverhulme Trust-funded Palaeolithic Archaeology, Geochronology, and Environments of the Southern Caucasus (PAGES) project, and to thank Dr. C. Hayward at the University of Edinburgh for assistance with the microprobe analyses. VO and OB were supported by the ECCOREV Research Federation (Aix-Marseille Université, CNRS) and Labex OT-Med (n° ANR- 11-LABX-0061) funded by the French Government "Investissements d'Avenir" programme of the French National Research Agency (ANR) through the A*MIDEX project (n°ANR-11-IDEX-0001-02).

**Competing interests:** The authors have declared that no competing interests exist.

ka. The evidence presented suggests that the stratified occupations at Kalavan 2 locale were repeated ephemerally most likely related to hunting in a high-elevation within the mountainous steppe landscape.

## Introduction

Modern states of Armenia, Azerbaijan, and Georgia occupy an area ca. 240,000 km², located between the Greater Caucasus Mountains in the north and the Araxes River in the south (Fig 1). This area which geographically covers the southern Caucasus and the north-eastern part of the Armenian highlands encompasses a rugged and environmentally diverse landscape, with a mosaic of different ecological niches and a prominent temperature gradient (Fig 1). Strong seasonal fluctuations as manifested in precipitation and temperature occur across elevations that range from sea level to more than 5000 m asl [1]. As a result, the temporal and spatial accessibility of resources across this terrain sharply fluctuated throughout the year. Those ecological challenges are manifested in the seasonal depletion of resources or opportunities such as animal seasonal migration, most likely impacted hunter-gatherer mobility patterns and lifeways. Hunter-gatherers' decisions regarding their mobility took also in consideration the social factors such as finding mating partners and maintaining demographic viability [2]. The archaeological record preserves an archive that echoes past cumulative decision-making regarding hunter-gatherers mobility and land-use as coping strategies to mitigate those ecological and social risks in order to adapt to specific environmental constraints [3].

Our current knowledge of hominin occupation and settlement dynamics in the southern Caucasus and Armenian highlands during the last 250 ka has many temporal and spatial gaps, as few eco-geographic sub-regions have been studied intensively [4, 5] (Fig 1). However, the majority of the Middle and Upper Paleolithic sites (MP and UP) are dated to ~65–30 ka (late Marine Isotope Stages (MIS) 4 up to 2) [4–16]. The environmental conditions during this time interval indicated by regional long-term climatic records were characterized by abrupt, millennial-scale temperature and precipitation oscillations between cold (stadial) and warm (interstadial) stages [17–25]. Two parts of the research area have been widely studied: the Imereti region in Georgia where mainly cave sites were studied, including Ortvale Klde are located at 530 m asl [5, 13, 26]. The second research area is located around Yerevan in the Hrazdan River valley in Armenia ranging from ca. 900 m asl at Yerevan -1 cave [4, 27, 28], and Lusakert- 1 cave at 1420 m asl [14, 15, 29] and Alapars-1 at 1774 m asl [16]. Other sites have been found in diverse biomes (e.g., Barozh 12 at 1336 m asl [7, 8, 30] or elevation settings (e.g., the low-elevation site of Bagratashen 1 at 457 m asl m asl [12]. Notable exceptions include the relatively higher-elevation sites of Aghitu 3 (1601 m asl) [6, 31], Kalavan 2 [10] at 1640 m asl, and Hovk 1 at 2040 m asl [11, 32]. At Hovk 1, however, the occupations were sporadic, and the fauna mostly accumulated as a result of a sedimentological trap, not anthropogenic agencies [11, 33]. Still, the role of mountainous environments in settlement systems during the Late Pleistocene i.e. the Middle Palaeolithic (MP), is however poorly integrated with current understandings of MP settlement patterns and lifeways.

The open-air MP site of Kalavan 2 (UTM 40T 3821 m E, 450551 m N, Republic of Armenia; Fig 1) is situated at 1640 m asl and is an ideal locality for testing the role of high elevation sites in Late Pleistocene hominin settlement systems. The site is located on the northern slopes of the Areguni Mountains, ca. 8 km north of the northern shores of Lake Sevan, and 70 km northeast of Yerevan (Fig 1). Kalavan 2 is positioned at the confluence of the Barepat River and one of its tributaries. The Barepat is connected to the Kura River watershed by the Getik

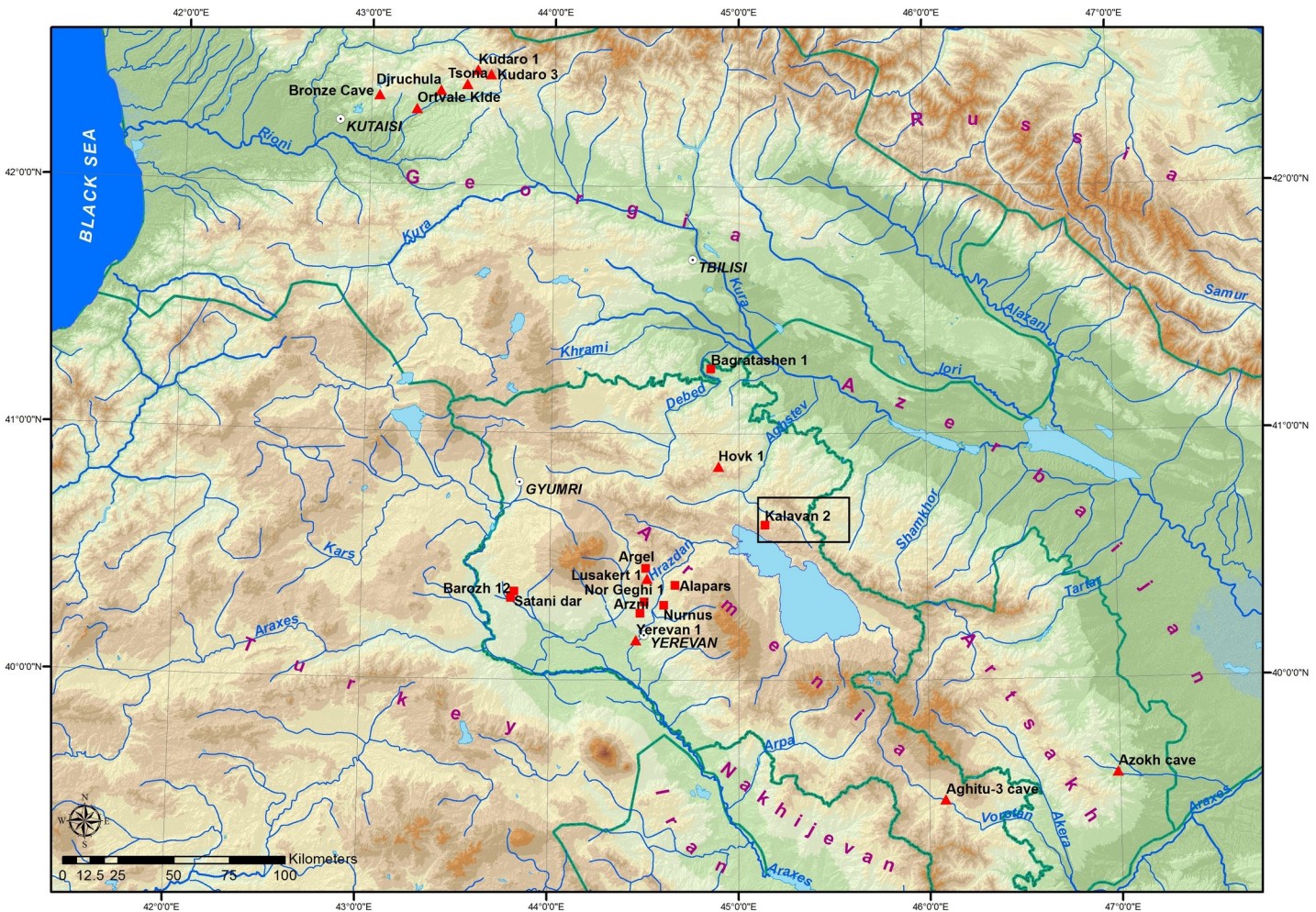

**Fig 1. Location map of Kalavan 2 within southern Caucasus and north-eastern Armenian highlands showing main Lower, Middle, and Upper Paleolithic sites.** Squares = open-air sites, triangles = cave sites.

and Aghstev Rivers (Fig 2). The site was initially excavated by an Armenian-French team in 2006–2007 that exposed MP horizons of lithic artifacts and faunal remains distributed in at least 4 m of stratified alluvial deposits [10].

Here we focus on the results of renewed excavations at Kalavan 2, conducted between 2017 and 2019 (the Institute of Archaeology and Ethnography, Republic of Armenia and Monrepos Archaeological Research Centre and Museum for Human Behavioural Evolution). We first describe the model of elevational mobility strategies and its testable implications for the Kalavan 2 record, particularly concerning the timing and duration of occupation. Next, we update the site's stratigraphy and chronology based on new sedimentological, X-Ray Fluorescence (pXRF) elemental analyses, tephra, and luminescence (pIRIR) dating analyses. We then report our findings based on recently recovered lithic and faunal assemblages. Additionally, two paleovegetation proxies–pollen and leaf waxes–were analyzed to better characterize the environment during MP occupations. Together our results highlight MP hunter-gatherer lifeways, specifically with respect to their subsistence, mobility, and land-use. This record, therefore, enables us to test hypotheses regarding elevation-dependent seasonal mobility and subsistence

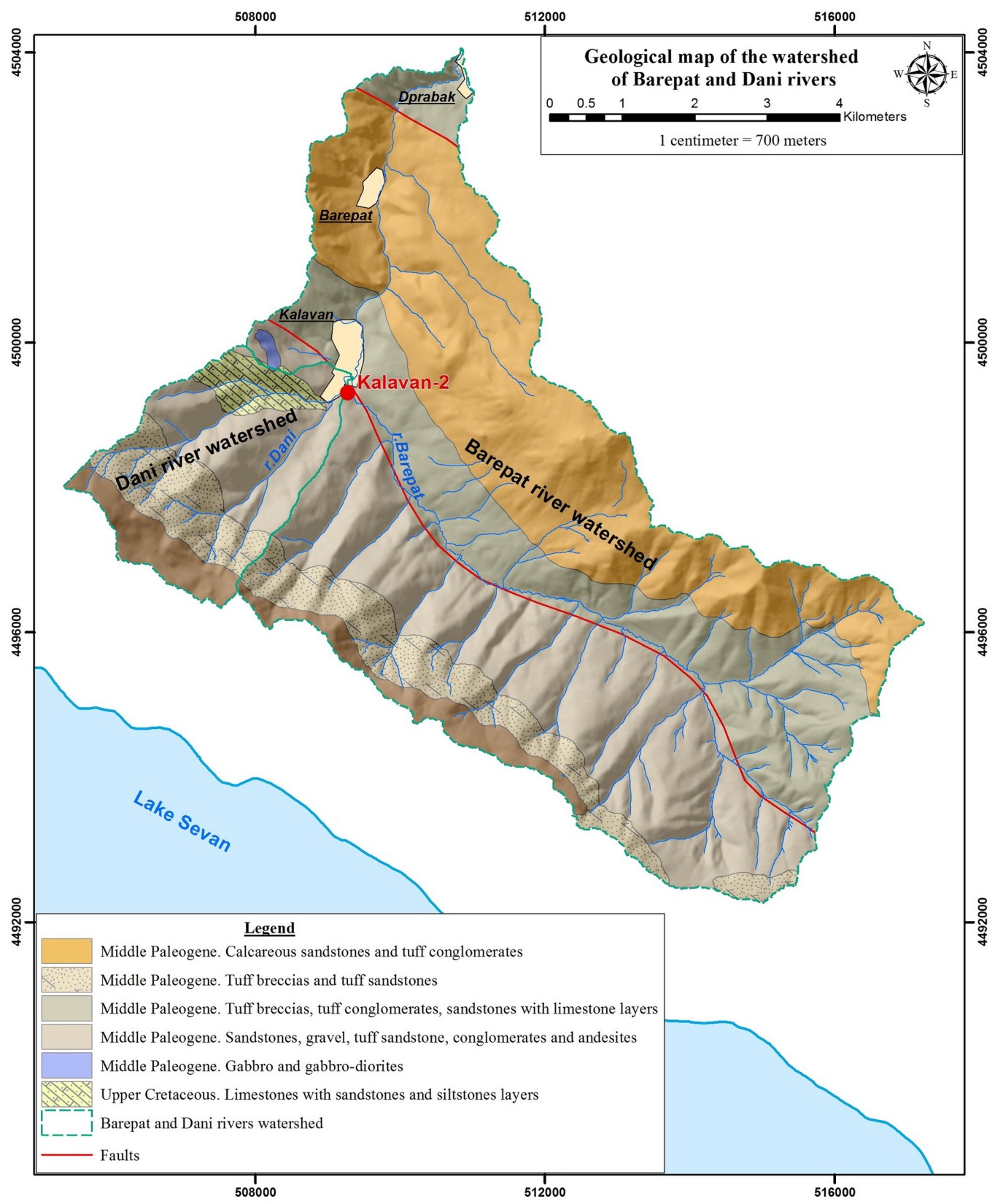

**Fig 2. Geological map of Kalavan area showing the larger watershed of the Barepat and in blue line the Dani watershed.**

strategies and the nature of hominin-environmental interactions in a mountainous environment.

## Testing elevation-dependent mobility strategies

The area between the Greater Caucasus Mountains and the Araxes River is a hotspot of biodiversity, hosting a large number of endemic plant and animal species related to its role as a glacial refugium [1, 34]. Modern climatic records from Armenia reveal large temperature and precipitation gradients across different elevations and strongly seasonal climates. For example, mean July temperatures range from 19 to 33˚C, whereas January temperatures typically span between -4 to -27˚ C, depending on elevation. Elevational gradients have a strong influence on floral and faunal community distributions [1: Fig 15.4]. During the Late Pleistocene, seasonal fluctuations in temperature and humidity were likely even more pronounced than today [17]. During glacial periods, winters were thought to have lasted longer, while summer droughts were more frequent, whereas during interglacials precipitation occurred mainly in spring to early summer [35–37]. These hydrological changes impacted distributions of terrestrial biota, such as shifts in the treeline elevation [23, 24]. The seasonal distribution of resources currently shapes animal migrations [38–42] and most likely will also then have governed the migrations of animals and the hominins that followed them [43, 44].

Mobility is an inherent strategy of population sustainability among hunter-gatherers [2]. It seems as well to be the primary strategy used to locate resources, but social networks are also crucial in providing a "safety net" against fluctuating resource availability [45, 46]. Recent models have identified habitat suitability as the best predictor of hunter-gatherer mobility [47]. Thus, decision-making regarding mobility and settlement strategies likely took into account the variable spatio-temporal access to subsistence resources and/or to maintain social networks [46, 48, 49]. Based on the current local conditions in Armenia (e.g., annual precipitation distributions, snow coverage, seasonal animal migration patterns), we postulate that the Late Pleistocene landscape of Armenia consisted of a fairly predictable, yet topographically segregated, distribution of diverse resources over short distances. Consequently, seasonally fluctuating resource availability along topographic gradients in the region most likely prompted Late Pleistocene hominin groups to adapt through adjustments in their mobility strategies.

We can hypothesize that the settlement strategies of Late Pleistocene hunter-gatherer groups were strongly conditioned by these elevation-dependent seasonal temperature and precipitation gradients [43]. If this hypothesis cannot be refuted, we expect:

i. in winter, higher altitudes being inaccessible, populations would move to lower-elevation areas/enclosed sites, possibly taking advantage of the moderating effects of river valleys and paleo-lake basins;

ii. in spring and autumn, hominin groups would follow migrating game and sporadically extend their mobility range into higher elevations; and

iii. during summer and autumn, hominin groups would spend longer periods at higher altitudes.

Archaeological sites at the highest elevations are expected to be occupied mainly during the summer and not in other seasons. Elevation-dependent seasonal hunter-gatherer mobility creates a set of testable expectations displayed in the archaeological proxies related to the roles of sites within the settlement system [43, 44, 50]. At 1640 m asl, Kalavan 2 is currently situated in a deciduous-mixed forest below the upland meadow-grassland, within an elevational ecotone

[51]. Such an environment is located at the interface of altitudinal zonation of flora and seasonal migration of fauna (see above). This setting provides an ideal case study for integrating high elevation adaptations within hunter-gatherer's settlement systems.

Hunter-gatherers organize settlement their movements along a continuum of logistical and residential mobility [45]. We dichotomize between two hypothetical extremes of this continuum: (i) residential sites ("home bases") and (ii) ephemerally occupied, task-specific sites. Residential sites reflect long-term occupations with varied types of activities, whereas task-specific sites refer to short-term occupations with restricted sets of activities such as hunting kill site. At high elevations during summer, we should expect more ephemeral occupations, specifically task-specific sites that are represented by low-density lithic artifact assemblages exhibiting fragmented core reduction sequences and abundant retouched pieces, highly curated and used for a narrow range of activities (see also [52, 53]). During winter, Late Pleistocene hominin settlement and mobility systems should relocate down to lower elevations, creating longer-duration residential sites. Such sites should contain dense palimpsests of lithic artifacts, encompassing a greater proportion of reduction sequences and tools used for diverse activities (see also [53, 54]). From a faunal perspective, residential sites should contain diverse hunted assemblages with selective transport of meat and fat-rich body parts. At task-specific sites dedicated to hunting, we expect monospecific faunal assemblages containing non-nutritional bone elements [55–57]. These lithic and faunal expectations provide a framework for testing our hypothesis and its expectations, as outlined above.

## The stratigraphy and depositional environments at Kalavan 2

Initial excavations at Kalavan 2 were conducted in 2006–2007 by an Armenian-French team. This team opened three trenches within a Late Bronze Age cemetery [58]. The main excavation area–Trench 2 –exposed 7 m$^2$ to a depth of 99–115 cm below the surface. Within this upper portion of the site's deep stratigraphy, eight layers were identified. An additional 260 cm were excavated in a 1 m$^2$ sounding that exposed twelve additional layers. In total, within the 380 cm thick sequence, 19 stratigraphic layers were identified by the excavators [10]. Layers 19 to 16 are part of an alluvial terrace with major changes in the deposited material and particle size. Layers 15 to 11 were interpreted as a cyclical alluvial sequence with alternating sub-angular gravel and sandy-silt lenses that might correspond to seasonal deposition. Layers 10 to 6 contain a mixture of colluvial/alluvial deposits that have undergone weak pedogenesis. Layers 5 to 3 contain numerous sub-angular to sub-rounded gravels and stones in erosive contact, the product of major hydrological changes that began with torrential slope erosion followed by alluvial deposition. Layer 2 is a silty-sandy deposit with very small and diffuse sub-rounded gravels that represents a low-energy alluvial phase. Layer 1 corresponds to a humic horizon related to recent edaphic conditions with soil development based on the alteration of Layer 2 [10].

In 2017–2019, new excavations were undertaken by our international research group (the permit number in 2017 was N3, in 2018 N28 and 2019 N9). The study of finds from this excavation has been conducted under the same permits. The three original trenches were re-opened and expanded, and an additional trench (Trench 4) was excavated (Fig 3). Trench 2 was expanded to enlarge the sample of archaeological materials, re-evaluate the sedimentary sequence, and collect samples to refine the chronology. Trenches 1, 2, 3, and 4 (T1–T4 hereafter) were excavated to assess the spatial distribution of the archaeological materials and the lateral changes in the depositional sequence (Fig 3). A total area of 50 m$^2$ was excavated (T1 = 17 m$^2$; T2 = 13 m$^2$; T3 = 12 m$^2$; T4 = 8 m$^2$). The distances between trenches and the lack of clear stratigraphic continuation necessitated that each trench has its own numbering of sedimentary units (Tables 1–4; Figs 4–7).

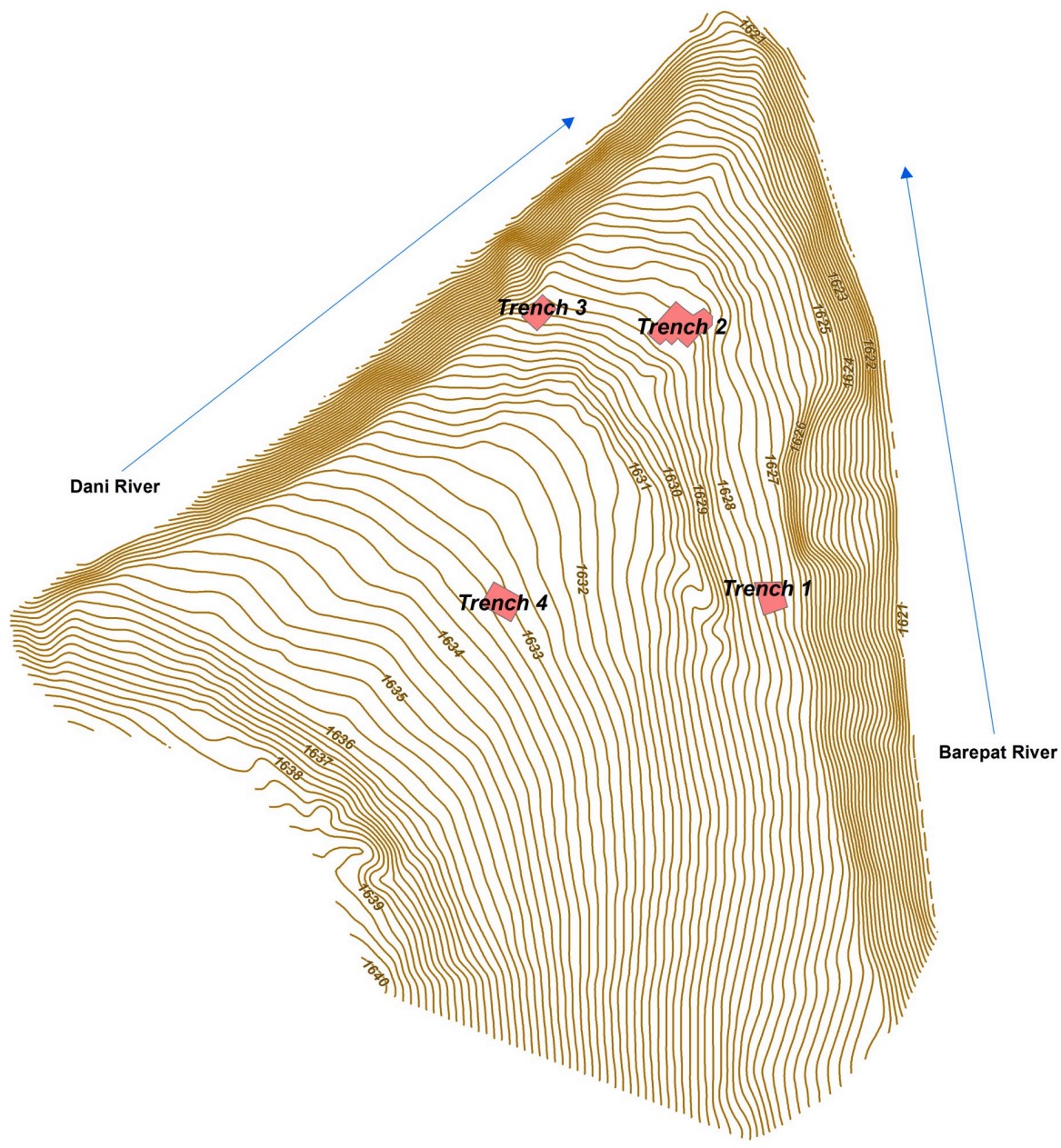

**Fig 3. Plateau topography and location of trenches at Kalavan 2.**

In T1, 17 m$^2$ were excavated to 250 cm below the surface (1626.04–1623.64 m asl). The stratigraphic sequence consists of seven units, alternating between gravels) 1a, 2a, 3a, and 4) and alluvial deposits (1b, 2b, 3b, 5, 6, and 7; for a detailed description of each unit see Table 1; Fig 4A and 4B). Faunal remains and lithic artifacts were recovered from Units 1b, 2b, and the interface of Units 5 and 6, while the highest artifact density appears in Unit 1b (Table 6). The thickness of the archaeological horizons in Units 1b and 2b varies between ca. 10 and 20 cm. The lithic and faunal assemblages in Units 5–6 are dispersed over 40–50 cm at the undulating interface of those units. In Unit 1b, the find density is 307 lithic artifacts and 225 bones per m$^3$, whereas the density in Units 5–6 is much lower, at 34 lithic artifacts and 28 bones per m$^3$ (Tables 6 and 13).

**Table 1. Description and interpretation of sedimentological units (from bottom to top) in Kalavan 2, Trench 1.**

| Unit | Description | Color | pIRIR225 results | Micro-morph | Lithic | Fauna | Sediment sources | Interpretation |
|------|-------------|-------|------------------|-------------|--------|-------|------------------|----------------|
| | Gravels and small pebbles grain size coarsening upwards. Pebbles size at their long axis ranges between 0.5–1 cm. | 7.5 yr 3/3 dark brown | | | | | Dani watershed | No Mg, granulometric classification showing torrential sediment transport. |
| 1b | Silty sand deposit with diffused rounded gravels manganese and iron linear concentrations | 7.5 yr 3/3 dark brown | 54.9 ± 3.8 | North section (MM3) | + | + | Barepat watershed influence | Mg 1,7% gravel abrasion (rounded) showing significant longitudinal transport. More Si than Unit 1a (>20%) and the appearance of aluminum element (Al>8%). The presence/apparition of potassium (K = 1,19%) is probably linked to reworking of the soils and endogenic rocks of the surrounding slopes and watershed. |
| 2a | Gravely pebbly unit with silty sand matrix with heteromeric gravel pebble 3–4 cm, sub-angular sub-rounded. At the basis of the Unit intensive manganese and iron concentration. | 7.5 yr 3/2 dark brown | | North section | | | Barepat watershed influence, discrete mixing with Dani | Mg 1,9% Si >20% Al around 8% Manganese (Mn) and Iron (Fe) concentration showing fluctuations of the water table in a quasi-permanent flowed/saturated sediment. Dani sediment supply could be related to sub-angular elements and a higher rate of carbonated rocks (Ca >6% instead of 1 to 3% in the underlying Unit). The presence of potassium (K) is probably linked to the reworking of the soils and endogenic rocks of the surrounding slopes and watershed. |
| 2b | Silty sand unit with diffused gravels subrounded and rounded, 5 mm, sub polyhedric structure | 7.5yr 3/3 dark brown | | North section | + | + | Dani watershed | Pedogenetic features showing less sediment input from Barepat river indicating the lateral movement of the Barepat riverbed. No Mg. The presence of Al and the poorer Ca rates underline the stasis of the deposition and the remnants and pedogenic evolution of some sediment supplied by the Barepat. The presence of potassium is probably linked to the reworking of the soils and endogenic rocks of the surrounding slopes and watershed. |
| 3a: | Pebble unit with small gravel of 5–6 cm. Silty sand matrix. Calcareous and siliceous blocks 10–40 cm. | 10yr 4/3 brown | | | | | Dani watershed | Same elemental characteristics as Unit 2b. In addition, the presence of bulky calcareous blocks which are the main lithological signature of the Dani river watershed. |
| 3b: | Sandy silty with fine gravel of 1 mm in longest axis, diffused pebbles. In the middle of the unit, there is a line of small rounded sub-rounded pebbles less than 1 cm. in max axis. | 10yr 4/3 brown | 52.7 ± 3.7 | | | | Barepat watershed influence | Presence of Mg (1,4%), higher rates of Si (>20%) and lower rates of Ca (<1,8%). Rounded pebbles showing significant longitudinal transport and a higher quantity of thin grained sediment. The presence of potassium (K) is probably linked to the recycling of the soils and endogenic rocks of the surrounding slopes and watershed. |

*(Continued)*

**Table 1.** (Continued)

| Unit | Description | Color | pIRIR225 results | Micro-morph | Lithic | Fauna | Sediment sources | Interpretation |
|------|-------------|-------|------------------|-------------|--------|-------|------------------|----------------|
| 4: | Gravely layer, pebbles sub-rounded and sub-angular element, heteromeric between 1 cm to 10 cm of max. axis. Sandy matrix and some spots of manganese and altered pebbles | 10 yr 3/4 dark yellowish-brown | | | | | Barepat watershed influence, mixing with Dani | The presence of 1,4% of Mg, high rates of Si (>23%), but higher rates of Ca (>3%), sub-rounded, and sub-angular calcareous pebbles could be the signature of a sedimentary input from the Dani river. The remarkable growth of the Potassium (K = 2,44%) is probably linked to the erosion of the surrounding soils from the Dani watershed. |
| 5: | Sandy silty layer with very small rounded gravel but very diffused, with traces of carbonates associated with the roots. | 10 yr 4/3 brown | 59.4 ± 4.5 | South section (MM4 | + | + | Barepat watershed influence, mixing with Dani | Presence of 1,2% of Mg, still high rates of Si (>19%), and rates of Ca (>3%). Again, sub-rounded and sub-angular calcareous pebbles. Here again, the presence of Potassium is probably more linked to the erosion of the surrounding soils from the Dani watershed. |
| 6: | Sandy layer with little silt with very small sub-rounded gravel | 10yr 5.2 grayish brown | 46.4 ± 4.1 | South section | + | + | Barepat watershed influence, mixing with Dani | Presence of 1,8% of Mg, rates of Si (>10%), and sub-rounded calcareous pebbles. The highest punctual rates of Ca (>12%) could be the signature of a brief Dani river sedimentary input. |
| 7 | Top soil- silty argillaceous clay with polyhedric structure with small gravel angular around 1 cm. in max. axis | 7.5.yr 3/2 dark brown | | | South section | | Mainly Barepat watershed. | The presence of 1,7% of Mg, growing rates of Si (>14%), as well as a significant drop of Ca rates (>6%) and thin grained deposits plead for a major sediment supply from the Barepat watershed. However, the small angular gravels and the re-appearance of K (>1.1%) could indicate a discrete input from the Dani watershed slopes. |

In T2, 13 m$^2$ were excavated to 100 cm below the surface (1629.42–1628.42 m asl). The lower layers exposed in the earlier excavations (Layers 10–20) were not re-exposed during recent excavations. The T2 stratigraphy contains eight sedimentary units. Among them, three phases of gravel were deposited (1a, 2a, and 3a), while two units (4 and 6) are alluvial sediments. Unit 5 comprises of a higher colluvial input. Unit 7 has an isolated lens (ca. 10 cm thick) of coarse- to very-coarse-grained volcanic ash (see Table 2 for more detailed descriptions; Fig 5). Our observations of the exposed stratigraphy are similar to those of the earlier excavations [10], thereby allowing us to draw correlations (i.e., Unit 4 in the present stratigraphy = Layer 7 in the former stratigraphy; Units 5–6 = Layer 5–6; Unit 7 = Layer 2) The lithics and faunal assemblages derive from Units 4, 5–6, and 7. The thickness of these archaeological horizons is usually ≲ 10 cm. Lithic and faunal find densities for these sedimentary units can be found in Tables 6 and 13. The artifact density of layer 7 from previous excavation, as reported in [10] is 145 lithics and 75 bones per m$^3$. In Units 5 and 6, the density of finds is 20 lithic artifacts and 30 bones per m$^3$, and in Unit 7, 135 lithic artifacts and 50 bones per m$^3$ (Tables 6 and 13).

In T3, 12 m$^2$ was excavated to 140 cm below the surface (1628.91–1627.77 m asl). The stratigraphy consists of eight sedimentary units that alternate between alluvial deposits (Units 1, 2b, 3b, and 7) and alluvial/ colluvial gravels (2a and 3a) (see Table 3 for a detailed description; Fig 6). Lithic artifacts were scattered in Units 1 (n = 3), 2b (n = 16), and 4 (n = 21) at an extremely low densities of finds. Fauna remains were only found in Units 2b and 4.

**Table 2. Description and interpretation of sedimentological units (from bottom to top) in Kalavan 2, Trench 2.**

| Unit | Description | Color | pIRIR225 results | Micro-morph | Lithic | Fauna | Sediment sources | Interpretation |
|---|---|---|---|---|---|---|---|---|
| 1a | Pebbles and blocks, sub-rounded heteromeric, 5–30 cm. The gravel matrix sand and gravel, 1 to 5 cm. Angular, subangular, middle size sand, and 1% silt. | 10yr 4/3 brown | | | | | Dani watershed | Calcareous blocks and homogenous elemental composition similar to the sediment supply generally attributed to the Dani watershed. To be noticed the presence of Al (1,1%) which can be linked with a relative proximity of the Barepat water table. The presence of Potassium (K) is probably linked to the erosion of the soils from the surrounding slopes of the Dani catchment. |
| 1b | Silty sand deposits with diffused gravels, 1 mm, sub abraded, structured lightly polyhedric, week pedogenesis. Sandy lens with coarse grain and medium sand | Color: 10yr 3/4 dark yellowish-brown | 55.3 ± 4.1 | | | | Dani watershed | Same characteristics as in Unit 1a. The presence of Potassium (K) is probably linked to the erosion of the soils from the surrounding slopes but also to a weak pedogenetic development observable in the sediment structure. |
| 2a | Gravel- matrix–small pebbles, 5–25 cm, and sands heteromeric, sub-angular to sub-rounded with traces of manganese and iron and diffused carbonates. | 4/2 dark grayish brown | | | | | Dani watershed | Same general characteristics as observed in the previous units. The sub-angular to sub-rounded pebbles indicates a modest longitudinal sedimentary transit following a Dani river origin. The presence of manganese (Mn = 0,8%) is linked to a rise and fluctuations of the water table. |
| 2b | Silty sand with carbonate traces very diffused gravel, linked the contact with another gravel, erosive contact gravel rounded | 10yr 4/3 brown | | | | | Barepat watershed influence | Appearance of the Magnesium (Mg = 1,7%). Decrease in Ca, rounded gravels and fine-grained sediment. |
| 3: | Gravel, sand and gravel matrix, heteromeric 5 mm– 4 cm, sub-angular to sub-rounded, some of the blocks (clasts) size range 10–25 cm. generally sub-rounded and rounded, upper part is weathered. | 10yr 4/3 brown | | | | | Dani watershed | Same general characteristics as observed in the underlying unitsattributed to Dani river. The sub-angular to sub-rounded pebbles and the calcareous blocks indicate a modest longitudinal sedimentary transit in accordance with a Dani river origin. The upper part of the unit shows traces of the progressive influence of an another Barepat sedimentary supply. |
| 4: | Silty sand with small aggregate of secondary carbonatation, with grass roots, structure polyhedric but very light. | 4/3 brown | 56.9 ± 4.4 | MM1 | + | + | Barepat watershed influence | Highest magnesium values (Mg = 2,3%) in the sequence, indicating Barepat influence. Elemental composition and weak pedogenetic evolution, suggests sediment structure/composition could be evidence of a proximal "floodplain" position relative to the Barepat River. |
| 5: | Medium size sand with small gravels ca.1 mm, sub-angular to angular. | 2.5y 5/3 light olive brown | | MM2 | + | + | Barepat watershed influence mixed with Dani sedimentary supply | The low Mg (1,5%) andAl values, and a higher Ca value (9,15%) compared to overlying Unit 4, could indicate a bigger mix between the Barepat and the Dani sediment supply/waters. |
| 6: | Siltly sand with very diffused pebbles, 0.5–2 cm, sub rounded and heterometric. | 10yr 4/3 brown | | MM2 | + | + | Dani watershed | Same general characteristics as observed in units attributed to the Dani river. |
| 7: | Silty clay with diffused gravel, polyhedric | 10yr 4/2 dark grayish brown | 44.1 ± 3.4 | | + | + | Dani watershed | Same general characteristics as observed in units attributed to Dani river. |

(*Continued*)

**Table 2.** (Continued)

| Unit | Description | Color | pIRIR225 results | Micro-morph | Lithic | Fauna | Sediment sources | Interpretation |
|------|-------------|-------|------------------|-------------|--------|-------|------------------|----------------|
| <u>8:</u> | Top soil- silty clay soil more structured than Unit 7, with sub-angular to angular gravels ca 1 mm to 5mm. | 10yr 2/2 –very dark brown | | | | | Dani watershed | Same general characteristics as observed in units attributed to the Dani river. |

In T4, 8 m$^2$ was excavated to 400 cm below the surface (1631.60–1627.60 m asl). The initial accumulation of the sedimentary sequence is alluvial (Units 1 and 2), with Unit 1 containing an abundance of medium-coarse grained volcanic ash. Units 3–6 reflect successive paleosols with a lens of tephra (Unit 6a) composed of medium-grained ash (see Table 4 for a detailed description; Fig 7). Only a few lithic artifacts (Units 2, 5, and 6) and macro-faunal remains (Units 2, 5d, 6b, 7, and 8) were found, although microfaunal remains were abundant throughout the sequence.

**Table 3. Description and interpretation of sedimentological units (from bottom to top) in Kalavan 2, Trench 3.**

| Unit | Description | Color | OSL sample | Lithic | Fauna | Sediment sources | Interpretation |
|------|-------------|-------|-----------|--------|-------|------------------|----------------|
| <u>1</u> | Silty-sand sediments with small carbonated roots with light polyhedric structure | 10yr 3/2 very dark grayish brown | 56.6±4.4 | + | | Barepat watershed influence | Mg (2,5%), Fine-grained sedimentation, floodplain environment. |
| <u>2a</u> | Pebbly gravely with sub-angular—subrounded gravel and pebbles, heteromeric, 2 mm—30 cm. and sand. | Color: 10yr 3/4 dark yellowish-brown | | | | Dani watershed | Same general characteristics as observed in units attributed to Dani river. The presence of sub-angular to angular gravel and pebbles shows a short longitudinal sedimentary transit. |
| <u>2b</u> | Sandy-silt with very diffused angular to sub-angular gravels, 2–3 mm, traces of carbonates roots, polyhedric structure. | 10yr 3/3 dark brown | 62.7 ± 4.5 | + | + | Barepat watershed influence mixed with Dani river | Mg (1,8%), fine- grained sedimentation but angular to sub-angular gravels showing intermittent input from slopes, local channel, and surface runoff. |
| <u>3a</u> | Gravel- pebbly with heteromeric pebbles, subrounded and sub-angular, carbonated. Sandy silt matrix. | 10yr 4/2 dark grayish brown | | | | Dani watershed | Same general characteristics as observed in units attributed to Dani river. The presence of sub-angular to angular gravel and pebbles shows a short longitudinal sedimentary transit. |
| <u>3b:</u> | Sandy-silt with some pebbles diffused sub-rounded 5 mm—3 cm. Traces of root carbonates, carbonate aggregates | 10yr 4–4 Dark yellowish brown | 60.3 ± 4.4 | | | Dani watershed | Same general characteristics as observed in units attributed to Dani river. |
| 4a: | Alluvial reworked volcanic sand with gravels < 1–5 mm | 2.5.y 2/5/1 Black | | + | + | Dani watershed | Same general characteristics as observed in units attributed to Dani river. The presence of sub-angular to angular gravel and pebbles shows a short longitudinal sedimentary transit. |
| <u>4b:</u> | Sand with rare gravels subrounded. 0.5 cm, the unit reworked the previous unit | 4a/ black | | + | + | Dani watershed | Same general characteristics as observed in units attributed to Dani river. The presence of the highest Ca value (6,48%) confirmssediment supply from Dani watershed. |
| <u>5:</u> | Reworked of unit 4b | | | | | Dani watershed | Same general characteristics as observed in units attributed to Dani river. The sediment reworking is visible in the elemental composition |
| <u>6:</u> | | | | | | Dani watershed | Same general characteristics as observed in units attributed to Dani river. |
| <u>7:</u> | Top soil, silty sandy with sub-angular gravels diffused top part polyhedric structure, with charcoal | 10yr 4/2 dark grayish brown | | | | Dani watershed | Presence of sub-angular gravels and discrete development of a soil formation shows a short longitudinal sedimentary transit. |

**Table 4. Description and interpretation of sedimentologicals unit (from bottom to top) in Kalavan 2, Trench 4.**

| Unit | Description | Color | OSL | Lithic | Fauna | Sediment sources | Interpretation |
|---|---|---|---|---|---|---|---|
| 1: | Thin bedded medium sands, diffused and alternated gravels, 2 mm | 5y 2.5/1 black | | | | Dani watershed | Same general characteristics as observed in units attributed to Dani river. Presence of small gravels showing surface runoff from the Dani watershed. |
| 2: | Medium-size sand with sub-rounded, diffuse gravels 1–5 mm | 5y 3–1 very dark black grayish | 59.5 ± 6.1 | + | + | Dani watershed | Same general characteristics as observed in units attributed to Dani river. |
| 3: | Sandy silt unit with very diffuse small gravel subangular, traces of roots and bioturbation with light pedogenesis. | Color: 2.5y 3/2 very dark brown grayish | | | | Dani watershed with Barepat watershed influence | The presence of Mg (1,8%) could indicate a former sedimentary supply from the Barepat mixed with the Dani river. |
| 4: | Sandy silt with small carbonated roots, with sand lenses | 10yr 4/4 dark grayish brown | 51.3 ± 4.8 | | | Dani watershed | Same general characteristics as observed in units attributed to Dani river. Carbonated roots indicate a soil development. |
| 5a: | Silty sand carbonated with a root traces, bioturbation and polyhedric structure | 10yr 5.2 grayish brown | | + | | Dani watershed with Barepat watershed influence | The presence of Mg (1,5%) indicates a discrete sedimentary supply from the Barepat mixed with Dani river input. |
| 5b: | Sandy-silt unit with carbonate nodules, sub-angular gravel heteromeric 1–3 mm and discrete polyhedric structure | 10yr 4/2 dark grayish brown | | + | | Dani watershed | Most of the unit characteristics are observed in units attributed to Dani river. |
| 5c: | Sandy silt with carbonate root casts with gypsum, polyhedric structure | 10yr 3/2 very dark grayish brown | | + | | Dani watershed | Same general characteristics as observed in units attributed to Dani river. Gypsum underline the influence of the alluvial context (humidity variation) in the pedogenesis. |
| 5d | Sandy-silt with very diffuse, rounded gravels | 10yr 4/3 brown | | | + | Barepat watershed influence | The presence of the highest Mg values in the sequence (2,1%) indicate a sedimentary supply from the Barepat. Fine-grained sedimentation and rounded gravels support this observation. |
| 6a: | sandy silt unit, ash and, reworked dark sand | 10yr4-2 dark grayish brown | | | | Dani watershed | Some similar characteristics as observed in units attributed to Dani river but with the presence of ash and dark sand. |
| 6b: | Silty-sand with roots traces and very diffuse, small aggregates of sediment | 10yr 4–3 brown | | + | + | Dani watershed | Same general characteristics as observed in units attributed to Dani river. |
| 7: | Silty sand with root traces, little carbonate, polyhedric structure | 10yr 5–2 grayish brown | | + | + | Dani watershed | some similar characteristics as observed in units attributed to Dani river but with light pedogenic process. |
| 8: | silty sandy roots with carbonates polyhedric structure | 10yr 4/3 brown | 36.2 ± 2.7 | | + | Dani watershed | Same general characteristics as observed in units attributed to Dani river, roots presence indicate the first development of pedogenesis. |
| 9: | Sand with small carbonates aggregate | 10yr 4/3 brown | | | | Barepat watershed influence and Dani input. | Presence of Mg (1.3%) and a fine-grained sedimentary supply. High Ca value (9,15%) indicates weathering potentially associated with sedimentary supply from the Dani. |
| 10a: | Silty sand layer carbonated with small and diffused sediment aggregates, light polyhedric structure with bioturbation | 10yr 6/3 pale brown | | | | Dani watershed | Same general characteristics as observed in units attributed to Dani river. Bioturbation and polyhedric structure underline the first step of the development of a soil. |
| 10b: | Top soil: silty sand with small clay component, sub-angular gravels 0.5 to 2 cm, blocky structure | 7.5 yr 3/2 Dark brown | | | | Dani watershed | Same general characteristics as observed in units attributed to Dani river, then development of a top soil. |

## Materials and methods

A total of 2532 lithic artifacts from 13 sedimentary units (Tables 6–12) and 2163 bones were recovered from 15 sedimentary units (Table 13). Artifacts larger than 20 mm were mapped three-dimensionally using Total Station instruments (Leica FlexLine TS07), and each item was given a unique ID number according to a number bank for each square. All excavated

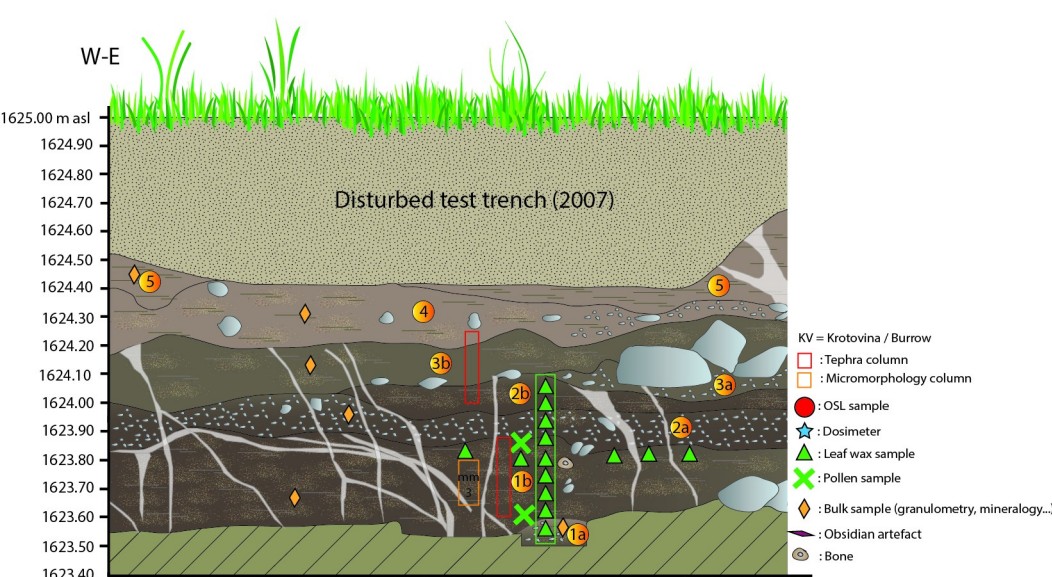

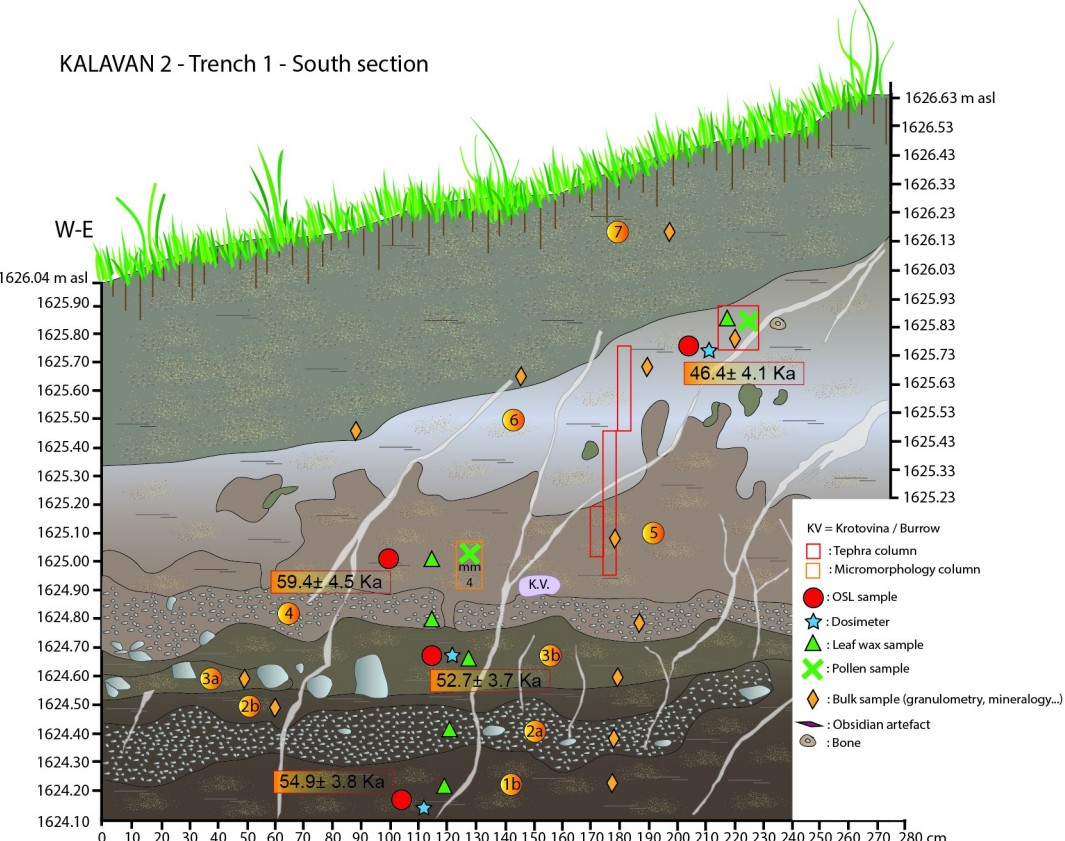

**Fig 4.** A: Section drawing of the Kalavan 2, Trench 1 north section including sample locations; B: Section drawing of the Kalavan 2, Trench 1 south section including sample locations.

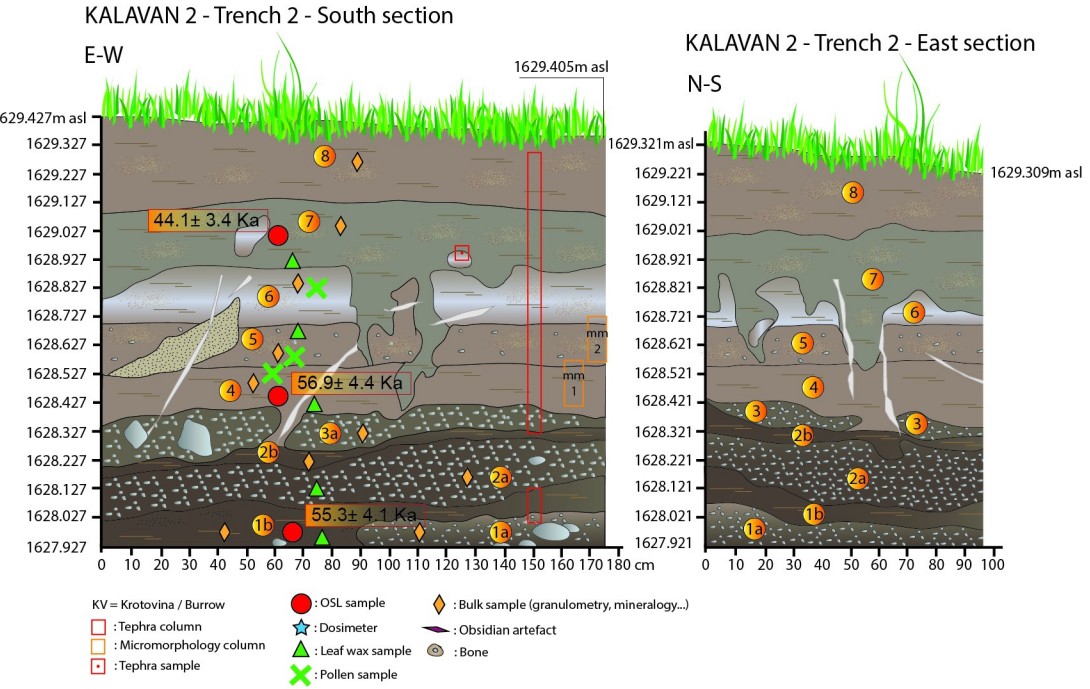

**Fig 5. Kalavan 2, Trench 2 east and south sections with sample locations.**

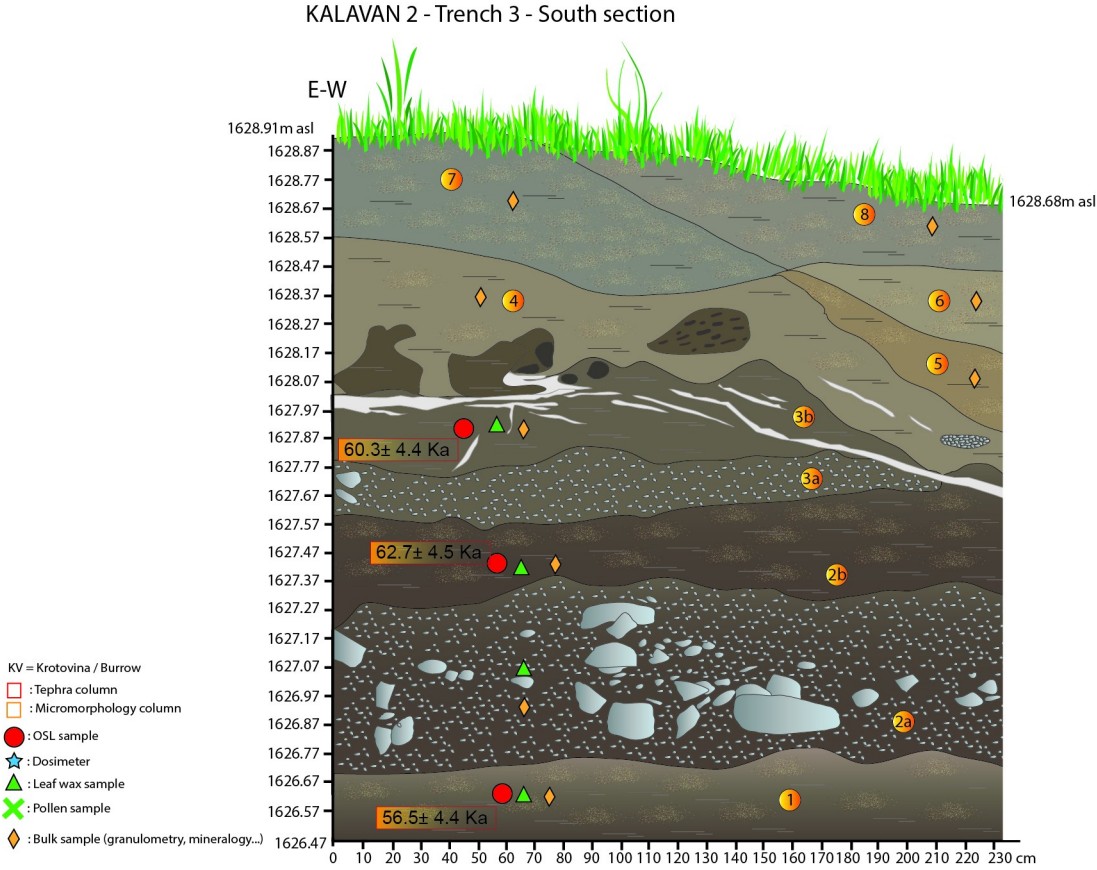

**Fig 6. Kalavan 2, Trench 3 south section with sample locations.**

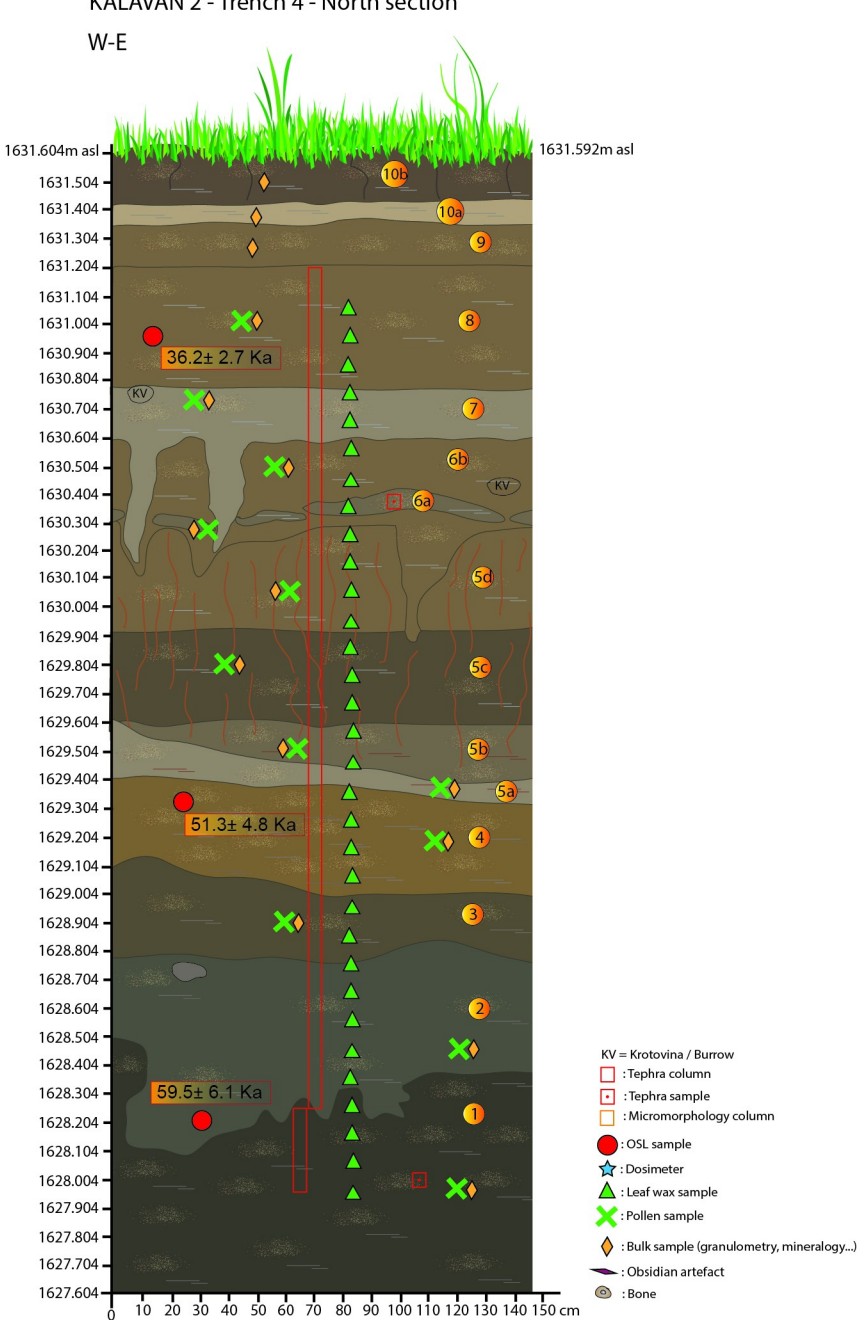

**Fig 7. Kalavan 2, Trench 4 north section with sample locations.**

sediments were dry-sieved through 5 mm mesh; a ca. 10% sample was wet sieved. In T4, two sub-squares were sampled for wet sieving over the full sequence (2 m$^3$ –one metric ton of sediments).

Two methods were used to characterize the sediments: Portable X-ray fluorescence (pXRF) (see S2 File for the pXRF protocols) and micromorphological thin sections (see S2 File for the micromorphological methods). Two dating methods were employed: post-infrared infrared stimulated luminescence (pIRIR) and tephrochronology characterizing both visible and

**Table 5. pIRIR225 results (4–11 μm) from Kalavan-2.**

| Field code | Sample ID | Total doserate (mGy/a) | De pIRIR225 (Gy) | pIRIR225 age (ka) |
|---|---|---|---|---|
| Kal-2-1-0 | 1681 | 2.9 ± 0.3 | 135 ± 1.1 | 46.4 ± 4.1 |
| Kal-2-1-1 | 1682 | 3.3 ± 0.3 | 198 ± 1.1 | 59.4 ± 4.5 |
| Kal-2-1-2 | 1683 | 4.1 ± 0.3 | 214 ± 1.5 | 52.7 ± 3.7 |
| Kal-2-1-3 | 1684 | 3.6 ± 0.2 | 198 ± 1.9 | 54.9 ± 3.8 |
| Kal-2-2-1 (2017) | 1685 | 3.9± 2.7 | 109 ± 6.6 | 32.2 ± 3.2 |
| Kal-2-2-1 / 2018 | 1688 | 3.9± 2.7 | 10.6 ± 0.1 | 2.7 ± 0.2 |
| Kal 2-2-1/ 2018 | 1904 | 3.1 ± 0.2 | 138 ± 1.2 | 44.1 ± 3.4 |
| Kal-2-2-2 | 1686 | 3.6 ± 0.3 | 204 ± 1.4 | 56.9 ± 4.4 |
| Kal-2-2-3 | 1687 | 3.8 ± 0.3 | 208 ± 1.8 | 55.3 ± 4.1 |
| Kal 2-3-1/ 2018 | 1906 | 3.6 ± 0.3 | 224 ± 3.0 | 62.7 ± 4.5 |
| Kal-2-3-2 | 1689 | 3.6 ± 0.3 | 218 ± 1.7 | 60.3 ± 4.4 |
| Kal-2-3-3 | 1690 | 3.4 ± 0.3 | 195 ± 2.6 | 56.5 ± 4.4 |
| Kal-2-4-2 | 1692 | 2.6 ± 0.2 | 134 ± 1.4 | 51.3 ± 4.8 |
| Kal 2-4-1 | 1691 | 3.5 ± 0.3 | 127 ± 0.6 | 36.2 ± 2.7 |
| Kal 2-4-3 | 1693 | 3.0 ± 0.2 | 177 ± 11.0 | 59.5 ± 6.1 |

microscopic ash layers (tephra and cryptotephra, respectively), attempting building a relative, comparable chronology that can be linked to the volcanic origins (see S1 File for details of the methodologies of the two methods).

The lithic artifacts were inventoried according to the variables and attributes used in the analyses of Middle Paleolithic lithic assemblages (see [59] and references therein). The methodology for use-wear analysis on a sample of lithic artifacts is specified in S3 File. The protocols for pXRF obsidian sourcing are based on those published [60–62]. Faunal analyses focused on taxonomy, anatomical breakdown, and taphonomic history of the specimens are described in detail in S4 File. The protocols for faunal and micro-faunal are described in S4 File.

The archaeological material presented in this study is stored at the Institute of Archaeology and Ethnography, Republic of Armenia in Yerevan. The material is available for study with authorization from the Institute of Archaeology and Ethnography, Republic of Armenia in Yerevan. All other relevant data are within the manuscript and its Supporting Information files.

## Geomorphology and sedimentology

Kalavan 2 is located at the confluence of the Barepat River and a tributary locally known as the Dani River. The reconstructed depositional dynamics at Kalavan 2 reflect inputs from an alluvial fan of the Dani catchment and the wider watershed of the Barepat River (Fig 2). The Barepat River is a third-order stream [63], and its catchment covers 58.8 km$^2$, descending from 2300 to 1217 m asl over 16 km (maximum slope = 28%; average slope = 7.5%). The Dani River is a second-order stream and its watershed covers 3.3 km$^2$, descending from 2423 to 1660 m asl over 3.4 km (maximum slope = 49%; average slope = 22%). The expected contribution of the Dani River should manifest as rapid flood events with poor clast sorting, including larger, more angular clasts mixed with colluvial inflow from the slopes. The Barepat River's expected input would be deposits of smaller grain sizes (due to the wider distal floodplain) that should be better sorted and typically alluvial with much more rounded and/or sub-rounded clasts. The Dani watershed transports three main lithologic types: Cretaceous limestone, Paleogenic sandstone, and various volcanogenic rocks. The Barepat watershed also encompasses conglomerates and metamorphic rocks as well as a few Cretaceous limestones within its upstream

**Table 6. Quantities and densities of lithic artifacts in the different trenches and sedimentological units at Kalavan 2.**

| T1 | Unit 5–6 (8.5 m$^3$) | Items ≥2 cm | Items ≤2 cm | Density per M$^3$ ≤2 cm | Total | Density of lithic artifacts per M$^3$ |
|---|---|---|---|---|---|---|
| Obsidian | N | 49 | 71 | | **120** | |
| | % | 40.8 | 59.2 | | 100 | |
| Non-Obsidian | N | 30 | 68 | | **98** | |
| | % | 30.6 | 69.4 | | 100 | |
| **Total** | N | **79** | **139** | **16** | **218** | **27** |
| | % | 36.2 | 63.8 | | 100 | |
| | Unit 1a (3.4 m$^3$) | Items ≥2 cm | Items ≤2 cm | Density per M$^3$ ≤2 cm | Total | Density of lithic artifacts per M$^3$ |
| Obsidian | N | 748 | 131 | | 879 | |
| | % | 85.1 | 14.9 | | | |
| Non-Obsidian | N | 77 | 88 | | 165 | |
| | % | 46.7 | 53.3 | | 100 | |
| **Total** | N | **825** | **219** | **64** | **1044** | **307** |
| | % | 79.0 | 21.0 | | 100 | |
| T2 | Unit 4 (2.6 m$^3$) | Items ≥2 cm | Items ≤2 cm | Density per M$^3$ ≤2 cm | Total | Density of lithic artifacts per M$^3$ |
| Obsidian | N | 62 | 46 | | **108** | |
| | % | 57.4 | 42.6 | | 100 | |
| Non-Obsidian | N | 153 | 208 | | **361** | |
| | % | 42.4 | 57.6 | | 100 | |
| **Total** | N | **215** | **254** | **98** | **469** | **180** |
| | % | 45.8 | 54.2 | | 100 | |
| | Unit 5–6 (3.9 m$^3$) | Items ≥2 cm | Items ≤2 cm | Density per M$^3$ ≤2 cm | Total | Density of lithic artifacts per M$^3$ |
| Obsidian | N | 23 | 39 | | **62** | |
| | % | 37.1 | 62.9 | | 100.0 | |
| Non-Obsidian | N | 2 | 14 | | **16** | |
| | % | 12.5 | 87.5 | | 100 | |
| **Total** | N | **25** | **53** | **14** | **78** | **20** |
| | % | 32.1 | 67.9 | | 100 | |
| | Unit 7 (2.6 m$^3$) | Items ≥2 cm | Items ≤2 cm | Density per M$^3$ ≤2 cm | Total | Density of lithic artifacts per M$^3$ |
| Obsidian | N | 161 | 123 | | **284** | |
| | % | 56.7 | 43.3 | | 100 | |
| Non-Obsidian | N | 23 | 45 | | **68** | |
| | % | 33.8 | 66.2 | | 100 | |
| **Total** | N | **184** | **168** | **65** | **352** | **135** |
| | % | 52.3 | 47.7 | | 100 | |

basin. Furthermore, the Barepat watershed includes numerous spring and travertine deposits (Fig 2).

During the Late Pleistocene, Kalavan 2 was located at the end of the Dani watershed at the meeting point of the Dani catchment alluvial fans with the Barepat River, creating alternating braided channels and thalwegs in response to shifting hydroclimatic and meteorological conditions. As attested in the excavation trenches, sedimentary dynamics between these rivers changed over time. The ground surface of the site area overlooks the valley of the Dani River by ca. 8 m above the current stream and ca. 18 m above the current Barepat stream. The Trenches 1–4 lie in different positions relative to the Dani and Barepat rivers. T1 is the closest to the Barepat while T3 is closest to the Dani (Fig 3). Both T2 and T4 are located between the two, yet the current surface of T4 is 2 m higher in elevation than T2. The distance between T1

**Table 7. The techno-typological composition of T1-Units 5–6,2b and 1b lithic assemblages.**

| T1 | Unit 5–6 | | | | Unit 2b | | | | Unit 1b | | | |
|---|---|---|---|---|---|---|---|---|---|---|---|---|
| | Obsidian | | Non-obsidian | | Obsidian | | Non-obsidian | | Obsidian | | Non-obsidian | |
| | N | % | N | % | N | % | N | % | N | % | N | % |
| Core | 1 | 0.8 | - | - | - | - | - | - | 3 | 0.3 | - | - |
| Core-on- flake | - | - | - | - | - | - | - | - | 1 | 0.6 | 1 | 0.6 |
| Cortical element | - | - | - | - | - | - | - | - | - | - | 2 | 1.2 |
| Flake | 28 | 23.3 | 54 | 55.1 | - | - | 4 | 44.4 | 51 | 5.8 | 65 | 39.4 |
| Kombewa flake | 2 | 1.7 | | 0.0 | - | - | - | - | - | - | - | - |
| Levallois flake | 2 | 1.7 | 4 | 4.1 | - | - | - | - | - | - | - | - |
| Levallois point | - | - | - | - | - | - | - | - | - | - | 2 | 1.2 |
| Levallois blade | - | - | - | - | - | - | - | - | - | - | 1 | 0.6 |
| Blade | 2 | 1.7 | 4 | 4.1 | - | - | - | - | 2 | 0.2 | 2 | 1.2 |
| Bladelet | 4 | 3.3 | 2 | 2.0 | - | - | - | - | - | - | - | - |
| Naturally backed knife | - | - | - | - | - | - | - | - | - | - | 1 | 0.6 |
| Core trimming element | 3 | 2.5 | 2 | 2.0 | - | - | - | - | 2 | 0.2 | 1 | 0.6 |
| Tool | 15 | 12.5 | 2 | 2.0 | 3 | 4.2 | 2 | 22.2 | 23 | 2.6 | 7 | 4.2 |
| Chip | 49 | 40.8 | 30 | 30.6 | 69 | 95.8 | 3 | 33.3 | 748 | 85.1 | 77 | 46.7 |
| Shaping flake | 14 | 11.7 | | 0.0 | - | - | - | - | 50 | 5.7 | - | - |
| Chunk | - | - | - | - | - | - | - | - | - | - | 1 | 0.6 |
| Hammerstone | - | - | - | - | - | - | - | - | - | - | 4 | 2.4 |
| Total | 120 | 100 | 98 | 100 | 72 | 100 | 9 | 100 | 879 | 100 | 165 | 100 |

and T2 is ca. 25 meters, while T3 and T2 are closer to each other, less than 5 meters apart. The distance between T2 and T4 is ca. 30 meters.

Two methods were used to characterize the sediments. First, portable X-ray fluorescence (pXRF) was used to analyze 43 sediment samples from four trenches (see S2 File for the pXRF

**Table 8. The techno-typological composition of Trench T2-Unit 4 lithic assemblage.**

| T2—Unit 4 | Obsidian | | Non-obsidian | |
|---|---|---|---|---|
| | N | % | N | % |
| Core | - | - | 5 | 1.4 |
| Cortical element | - | - | 3 | 0.8 |
| Flake | 10 | 9.3 | 145 | 40.2 |
| Kombewa flake | 1 | 0.9 | 5 | 1.4 |
| Levallois flake | - | - | 1 | 0.3 |
| Levallois point | - | - | 2 | 0.6 |
| Levallois blade | - | - | 2 | 0.6 |
| Blade | 2 | 1.9 | 16 | 4.4 |
| Bladelet | - | - | 2 | 0.6 |
| Naturally backed knife | - | - | | 0.0 |
| Core trimming element | 2 | 1.9 | 7 | 1.9 |
| Tool | 12 | 11.1 | 11 | 3.0 |
| Chip | 62 | 57.4 | 153 | 42.4 |
| Shaping flake | 19 | 17.6 | 6 | 1.7 |
| Hammerstone | - | - | 2 | 0.6 |
| Chunk | - | - | 1 | 0.3 |
| Total | 108 | 100 | 361 | 100 |

Table 9. The techno-typological composition of T2 -Units 5–6 lithic assemblage.

| T2- Unit 5–6 | Obsidian | | Non-obsidian | |
|---|---|---|---|---|
| | N | % | N | % |
| Core | - | - | - | - |
| Cortical element | - | - | - | - |
| Flake | 12 | 19.4 | 9 | 56.3 |
| Kombewa flake | - | - | - | - |
| Levallois flake | - | - | - | - |
| Levallois point | - | - | - | - |
| Levallois blade | - | - | - | - |
| Blade | - | - | 1 | 6.3 |
| Bladelet | - | - | | 0.0 |
| Naturally backed knife | - | - | - | - |
| Core trimming element | - | - | 2 | 12.5 |
| Tool | 3 | 4.8 | | 0.0 |
| Chip | 23 | 37.1 | 2 | 12.5 |
| Shaping flake | 24 | 38.7 | - | - |
| Hammerstone | - | - | - | - |
| Chunk | - | - | 2 | 12.5 |
| Total | 62 | 100 | 16 | 100 |

protocols). Utilizing pXRF, we sought to geochemically differentiate between sediments derived from the Barepat watershed and those from the Dani. Second, micromorphological analysis was conducted on four thin sections from the T1 and T2 sedimentary units (also see S2 File for the micromorphological methods). These analyses aimed to better understand the site formation processes.

Table 10. The techno-typological composition of Trench T2-Unit 7 lithic assemblage.

| T2- Unit 7 | Obsidian | | Non-obsidian | |
|---|---|---|---|---|
| | N | % | N | % |
| Core | 1 | 0.4 | 1 | 1.5 |
| Cortical element | 1 | 0.4 | 3 | 4.4 |
| Flake | 30 | 10.6 | 25 | 36.8 |
| Kombewa flake | - | - | - | - |
| Levallois flake | 1 | 0.4 | 1 | 1.5 |
| Levallois point | - | - | 1 | 1.5 |
| Levallois blade | - | - | 1 | 1.5 |
| Blade | - | - | 7 | 10.3 |
| Bladelet | 5 | 1.8 | 1 | 1.5 |
| Naturally backed knife | - | - | - | - |
| Core trimming element | 6 | 2.1 | - | - |
| Tool | 12 | 4.2 | 3 | 4.4 |
| Chip | 161 | 56.7 | 23 | 33.8 |
| Shaping flake | 66 | 23.2 | 2 | 2.9 |
| Hammerstone | - | - | - | - |
| Chunk | 1 | 0.4 | - | - |
| Total | 284 | 100 | 68 | 100 |

**Table 11. The techno-typological composition of Trench T3- Units 4, 2b and 1 lithic assemblages.**

| T3 | Unit 4 | | Unit 2b | | Unit 1 | |
|---|---|---|---|---|---|---|
| | Obsidian | Non-obsidian | Obsidian | Non-obsidian | Obsidian | Non-obsidian |
| | N | N | N | N | N | N |
| Core | - | - | - | - | - | - |
| Cortical element | - | - | - | - | - | - |
| Flake | 3 | 3 | - | 10 | - | - |
| Kombewa flake | - | - | - | - | - | - |
| Levallois flake | - | - | - | - | - | - |
| Levallois point | - | - | - | - | - | - |
| Levallois blade | - | - | - | - | - | 1 |
| Blade | - | - | - | 2 | - | - |
| Bladelet | 1 | - | - | - | - | - |
| Naturally backed knife | - | - | - | - | - | - |
| Core trimming element | 1 | - | - | - | - | - |
| Tool | 3 | - | 2 | - | - | |
| Chip | 10 | - | - | 2 | - | 2 |
| Chunk | - | - | - | - | - | - |
| Total | 18 | 3 | 2 | 14 | - | 3 |

**pXRF sediments results.** For all trenches, the major detectable elements of sedimentary deposits (Table 5) are Si, Al, Ca, Fe, and K. Light elements (LE; H through Na) are not measured but constitute between 57 and 88% of the samples by mass. Mg is also present but only at detectable concentrations in certain sedimentary units.

**Table 12. The techno-typological composition of Trench T4- Units 7, 6 and 5 lithic assemblages.**

| T4 | Unit 7 | | Unit 6 | | Unit 5 | | Unit 2 | |
|---|---|---|---|---|---|---|---|---|
| | Obsidian | Non-obsidian | Obsidian | Non-obsidian | Obsidian | Non-obsidian | Obsidian | Non-obsidian |
| | N | N | N | N | N | N | N | N |
| Core | - | - | - | - | - | - | - | - |
| Cortical element | - | - | - | - | - | - | - | - |
| Flake | - | - | - | 3 | 6 | 2 | 2 | 1 |
| Kombewa flake | - | - | - | - | - | - | - | - |
| Levallois flake | - | - | - | - | - | - | - | - |
| Levallois point | - | - | - | - | - | - | - | - |
| Levallois blade | - | - | - | - | - | - | - | - |
| Blade | - | - | - | - | - | - | - | - |
| Bladelet | - | - | - | - | 1 | - | - | - |
| Naturally backed knife | - | - | - | - | 1 | - | - | - |
| Core trimming element | - | - | - | - | 1 | - | 1 | - |
| Tool | - | - | 2 | - | 4 | - | - | - |
| Chip | 1 | 4 | - | - | 30 | 9 | 5 | - |
| Chunk | - | - | - | - | - | - | - | - |
| Total | 1 | 4 | 2 | 3 | 41 | 11 | 8 | 1 |

**Table 13. Number of bones and density per m³ from Kalavan 2 by trench and sedimentological unit.**

| T1 | Unit | 5/6 | 3b | 3a | 2b | 1b | total |
|---|---|---|---|---|---|---|---|
| | NSP | 183 | 17 | 13 | 74 | 764 | **1051** |
| | % of T1 | 17.4 | 1.6 | 1.2 | 7.0 | 72.7 | **100** |
| | % of total | 8.5 | 0.8 | 0.6 | 3.4 | 35.3 | **48.6** |
| | Density per M³ | 22 | - | - | 15 | 90 | |
| T2 | Unit | 7 | 5/6 | 4 | - | - | |
| | NSP | 129 | 117 | 854 | - | - | **1100** |
| | % of T2 | 11.8 | 10.7 | 77.6 | - | - | **100** |
| | % of total | 5.9 | 5.6 | 39.5 | - | - | **50.1** |
| | Density per M³ | 50 | 30 | 328 | - | - | |
| T3 | Unit | 4 | 2b | | | | |
| | NSP | 1 | 3 | - | - | - | **4** |
| | % of T3 | 25 | 75 | - | - | - | **100** |
| | % of total | 0.1 | 0.1 | - | - | - | **0.2** |
| T4 | Unit | 8 | 7 | 6b | 5d | 2 | |
| | NSP | 3 | 2 | 1 | 1 | 1 | **8** |
| | % of T4 | 37.5 | 25 | 12.5 | 12.5 | 12.5 | **100** |
| | % of total | 0.1 | 0.1 | 0.1 | 0.1 | 0.1 | **0.4** |

In T1, the mean LE values are the lowest (64.2%), while Si, Al, and K have the highest mean values (Si = 18%; Al = 6.2%; K = 2.4%). The mean elemental values are comparable to T1, however, its Fe values are the highest measured in all trenches (μ = 4.0%). In contrast, T3 and T4 are different from T1 and T2, including higher LE mean values (T3 = 85.1% and T4 = 82.4%). T3 has the lowest mean values for the major elements (Si = 4.3%; Al = 1.5%; Ca = 3.4%; K = 0.4%). T4 is very similar to T3 in terms of elemental values, but it does have a higher mean Ca content (4.9%).

To summarize, the pXRF data reflect the spatial relationships of the trenches: the major-element chemistries of T1 and T2 are more similar to each other, whereas T3 and T4 are chemically similar to one another (See S2 File, Fig 1). T1 and T2 are closer to the current Barepat riverbed, while T3 and T4 are closer to the Dani (Fig 2). An elemental signature of the Barepat appears to be higher Mg content. The sources of Mg are most likely spring-derived (see [64] reference therein). Many springs related to local tectonic faulting and fracturation feed into the Barepat River watershed (n = 31 for the Barepat; n = 7 for the Dani River watersheds), likely contributing Mg into the sedimentary input. The lack of Mg in the gravel units (i.e., Units 1a and 3a in T1; Units 1a, 2a, and 3a in T2) supports this interpretation. Those units most likely formed as an outcome of flooding events within the Dani watershed given that the large, angular calcareous blocks could only derive from upstream in the Dani catchment (**Fig 2**).

**Micromorphological analysis results.** Four thin section samples from T1 (n = 2) and T2 (n = 2) were examined (Sp. 2 Fig 2 in S2 Fig). A summary of our findings is presented here, while additional details can be found in the supplementary materials (S2 Table 2 in S2 Table and S2 File, Fig 2). All descriptions follow Stoops and Jongerius [65].

T1 Unit 1b (sediment sample MM3; thin sections MM3.1 and MM3.2) exhibits a random distribution with volcanic, metamorphic, and sedimentary subrounded heterogeneous grains in the matrix (S2 Figs 2, 3 in S2 Fig). MM3.1 is principally composed of fluviatile deposits with weak traces of pedogenesis. It also includes micro-fragments of obsidian flakes and bone (S2

Fig 2 in S2 Fig). MM3.2 has a similar fabric and texture as MM3.1 but with Mn, less evidence of pedogenesis, and more colluvial sediments in a fluviatile deposition (S2 Fig 3 in S2 Fig).

T1 Unit 5 (sample MM4) has a random distribution of grains (Sp. 2 Fig 4 in S2 Fig). It has a slightly denser grain texture than MM3 and includes more sub-angular to angular material with light rock alteration (i.e., low degree of chemical or mechanical erosion). Structures in MM4.1 are indicative of fluviatile and colluvial deposits with evidence of bioturbation (S2 Fig 4 in S2 Fig), but without notable pedogenetic development. MM4.2, relative to MM4.1, shows a slightly denser grains texture within the matrix and the appearance of reworked soil elements or aggregates with secondary veins of gypsum (S2 Fig 5 in S2 Fig). Despite some bioturbation traces, the fluviatile component remains dominant in the structure and organization of the sediments (S2 Fig 5 in S2 Fig).

T2 Unit 4 at the contact with Unit 5 (sample MM1) exhibits a lower frequency of grains in the matrix (S2 Figs 6, 7 in S2 Fig) than do MM3 and MM4 in T1 (S2 Figs 2–5 in S2 Fig). Grains in MM1 are rounded to sub-rounded and coated by secondary veins of gypsum/calcite (S2 Figs 6, 7 in S2 Fig). MM 1.1 indicates an alluvial deposit mixed with light colluvial supply. Little evidence of pedogenesis and bioturbation is visible. Some evidence of sediment compaction indicates short-term top-soil formation (Sp. 2 Fig 6 in S2 Fig). MM1.2 contains fewer grains than MM1.1 (S2 Figs 6 vs. 7 in S2 Fig). Gypsum and calcite are also present, as are micro-fragments of sub-angular bones. MM1.1 also indicates an alluvial deposit mixed with colluvial elements, and sediment compaction is visible as well (S2. Fig 7 in S2 Fig).

T2 Unit 5 at the contact with Unit 6 (sample MM2) has a random to clustered basic distribution pattern (S2 Figs 8, 9 in S2 Fig). MM2.1 shows poorly sorted sub-rounded particles and light rotation structures, attesting to chemical and/or mechanical erosion. Such erosion shows grain crushing, fragmentation by ice lensing, grain stacks, and smooth fractures typical of frost microstructures. This thin section is mainly constituted by alluvial deposits affected by preliminary to medium frost action (Sp. 2 Fig 8 in S2 Fig). MM2.2 exhibits clustered basic distribution patterns with sandy lenses and sub-rounded particles, and it has reworked sub-angular blocky peds and channel biostructures due to bioturbation. Hence, it reveals an alluvial deposit that reworked soil peds and was affected by a light post-depositional pedogenesis (S2 Fig 9 in S2 Fig).

Most thin sections from T1 and T2 show fluvisols, colluvial soils, or alluvial deposits with poorly evolved pedogenesis. The results of the analyses agree with the stratigraphic observations, recording the discontinuous geomorphic dynamics of an alluvial fan at the confluence of the Dani and Barepat Rivers during the Late Pleistocene. However, some specific observations can be made. In T2 Unit 4, there is some evidence of compaction, which could correspond to human and/or animal activity (trampling) during a short time on a soft, saturated sediment [66]. In T2 at the contact of Units 5 and 6, frost action reveals short-term weather trends such as a severe winter season. This frost feature appears very light without heavy modification of the sedimentary structure (no ice-wedge casts or sand wedges, fragipan horizons, flat bottomed cryoturbations, gelifluction deposits, etc.). During short cold events, when flows of water are less uniformly distributed, crystals such as gypsum or carbonates are formed, as has been observed in the field and under the microscope [67]. This type of crystallization does not appear in all thin sections (e.g., MM4.1, MM1.1, MM1.2). There is no evidence for a periglacial climate or long-term cold climatic conditions, given the evidence of frost, which was only preserved in MM1.1 and MM1.2. Thus, these findings support short periods of frost similar to the conditions prevailing today in the Kalavan area during winter.

## Chronology of the sedimentary sequence

The prior site chronology was based on an analysis of four [14]C samples from the 2006–2007 excavations. Their results placed the MP occupation between 42 and 16 ka; however, these

ages have since been questioned due to laboratory challenges and stratigraphic inconsistencies (see S1 File). Our project used two dating methods to establish a new chronological framework for the sedimentary sequences under investigation, and refine the timing of MP occupations. The first method is luminescence dating, which is based on the accumulation of trapped charge in natural dosimeters such as quartz or feldspar minerals. It has the potential to determine the last sunlight exposure and, therefore, the burial ages of sediment grains [68] (see S1 File for details). The second method is tephrochronology, the goal of which is to create a relative, comparable chronology that can be linked to the volcanic origins of visible and microscopic ash layers (tephra and cryptotephra, respectively). Our protocols for the visible tephra followed standard sedimentological methods [69], while our crypto-tephra analytical methods followed Lane et al. [70] and Blockley et al. [71]. Specifically, volcanic glass shards extracted from the sedimentary sequence were measured for major and minor elements using electron microprobe analysis (EMPA) [72] (see S1 File for the complete methods).

**Luminescence chronology.** Fifteen post-infrared infrared stimulated luminescence (pIRIR) samples have been analyzed [73, 74] on fine-grain (4–11 μm) polyminerals. Figs 4–7 and Tables 1–4 document the stratigraphic locations of pIRIR sampling in the various trenches. Age estimates from the bottom- and middle parts of the sedimentary sequences (T1 Units 1b-5, T2 Units 1b-4, T3 Units 1-3b, and T4 Units 2–4) range between 59.4 ± 4.5 and 51.3 ± 4.8 ka and overlap within the uncertainty intervals. Hence, the luminescence ages do not allow us to distinguish among several periods of sedimentation, but they point to the formation of the bottom- and middle sedimentary units at ca. 55 ka, during early MIS 3. The latest ages corresponding to the last MP occupations are ca. 45 ka (T1 top of Unit 6 = 46.4 ± 4.1 ka and T2 Unit 7 = 44.1 ± 3.4 ka). All ages are shown in Table 5 and Figs 4–7.

**Tephra results and interpretation.** Three visible volcanic ash horizons were identified across the trenches at Kalavan 2. In T2 Unit 7 the tephra is composed of coarse–very coarse-grained ash and forms an isolated lens (ca. 10 cm thick) (T17-0174) (Fig 5). In T4, two ash layers were identified: a lower tephra composed of medium-coarse grained ash forming a constituent of Unit 1 and (T17-0172); and a second tephra composed of medium-grained ash, and forming Unit 6a (ca. 5 cm thick), (T17-0173) (Fig 7). Mean analyses from the vitreous phase of this tephra are listed in S1 Table 1 in S1 Table, with the complete chemical datasets presented in S1 Fig 5 in S1 Fig and S1 Table 2 in S1 Table. Concentrations of cryptotephra shards were detected in almost all of the 10-cm scan samples from T1 and 2 (S1 Fig 6 in S1 Fig). In T1, two distinct zones of glass shards were identified: a lower one at 2.56–2.86 m and an upper one at 2.24–2.50 m below ground surface. Refining the most concentrated samples from the zones revealed two peaks in shard concentration: a lower one at 2.62–2.64 m (T18-0098; Unit 1b; 1379 shards/g) and an upper one at 2.32–2.34 m below ground surface (T18-0095; Unit 3b; 2912 shards/g). In T2, glass shards were detected in every scan sample between 0.06–1.04 m below the ground surface; however, a greater concentration was found between 0.46–0.66 m below the ground surface (S1 Fig 6 in S1 Fig). This interval was refined to a 2-cm resolution, which highlighted two peaks in glass shard concentration: a lower one at 0.64–0.66 m (T18-0110; Unit 6; 1839 shards/g) and an upper one at 0.54–0.56 m below ground surface (T18-0105; Unit 7; 2146 shards/g) (S1 Fig 6 in S1 Fig).

*Chemical interpretation.* Data from the analysis of the vitreous phase of these tephra are presented in S1 Fig 5 in S1 Fig and S1 Table 2 in S1 Table.

The potential source regions for volcanic material deposited at Kalavan 2 and over the Armenian highlands are particularly wide-ranging. Local volcanic centers are numerous (see [75–77]), and dominant atmospheric circulation patterns place the Armenian highlands downwind of Mediterranean, Anatolian, and Carpathian sources, as well as in the likely dispersal envelope of volcanoes from the Greater Caucasus [1, 78].

Two of the three visible tephra (T17-0172 and T17-0174) have a mildly alkaline trachyandesite geochemical signature, whereas the third is mildly alkaline trachyte/ trachydacite (T17-0173). Their compositions and visibility in the trench suggest that they originate from a relatively proximal volcanic source. During the Late Pleistocene, centers in the Gegham Highlands are known to have produced lavas with an alkaline trachyandesitic composition [75–77], so it is possible that these eruptions (ca. < 50 ka) generated plumes of ash that dispersed over the short distance across Lake Sevan to the site (ca. 40–60 km). Unfortunately, at present, the lack of grain-specific elemental data and detailed tephrostratigraphies from these volcanic centers in the Gegham Highlands limit the ability to support or refute such a hypothesis. Other potential source candidates are volcanoes located in the Eastern Anatolian Volcanic Province (EAVP), which produced trachyandesites and trachydacites throughout the Pleistocene [79–82]. One possible correlation is the Çekmece Formation (S1 Fig 7 in S1 Fig): a series of eruptions thought to originate from Nemrut volcano near Lake Van [83, 84]. The estimated age of this series is debated, but it is thought to have occurred sometime ca. 60–30 ka [84]. Such a broad age estimate reflects the current stratigraphic uncertainty of this volcanic series, but the estimate overlaps well with the pIRIR dates at Kalavan 2.

Several cryptotephra populations exhibit a distinct peralkaline rhyolite signature (T18-0098_a; T18-0095_a; T18-0110_a; T18-0106_a; T18-0105_a,b; (S1 Fig 5 in S1 Fig; S1 Table 1 in S1 Table). In the local-medial environment, one prominent volcano is known to have yielded tephras of this composition: Nemrut in the EAVP. This volcano has been particularly active during the last ca. 200 ka, producing several large eruptions (VEI≥5), including the HP-10/AP-6 eruption (ca. 61.6 ka), Çekmece Formation (ca. 60–30 ka), Tatvan Ignimbrite (ca. 45 ka) and Nemrut Formation (ca. 30 ka) [47–51]. Establishing if the cryptotephra at Kalavan 2 relates to any of these large-scale eruptions is difficult at present due to a lack of chemical data for the vitreous phase of Nemrut eruptions and the ongoing chrono-stratigraphic refinement of these events (see [84]). Nevertheless, based on peralkalinity, a tentative link between Nemrut and the Kalavan peralkaline cryptotephras can be proposed, but an exact match remains elusive.

Three of the cryptotephras (T18-0098_b, T18-0095_b, T18-0110_b) are calc-alkaline rhyolite with chemical signatures very similar to tephras from local-medial volcanic centers in Armenia [75, 77] and the EAVP [79, 82–91] as well as more distant volcanoes from the Aeolian Islands (ca. 2500 km east of Kalavan) [92], the East Carpathians (ca. 1700 km northwest of the site) [93–96], and the Central Anatolian Volcanic Province (CAVP; ca. 940–860 km east of Kalavan) [78, 97–102]. At present, it is not possible to provide a more reliable conclusion regarding the provenance of the Kalavan 2 calc-alkaline cryptotephras.

## Lithic artifact assemblages

During the previous excavations in T2, lithic artifact assemblages were recovered from five layers: Layers 2, 3, 6, and 7 over the entire 7 m² excavation area and Layer 11, which was exposed only in a deep test pit [10]. The lithics from Layers 1 and 2 were a mix of Medieval, Late Bronze Age, Chalcolithic ceramics, and MP artifacts, whereas lithic artifacts from Layers 3, 6, and 7 were attributed solely to the MP. Among the MP artifacts, the dominant raw material is obsidian (ca. 59% in Layer 7, for example). Sedimentary rocks including silicified limestone, chert, flint, and jasper as well as very few volcanic rocks such as basalt are also present [12, Table 1]. The authors stressed the dominance of the recurrent convergent unipolar Levallois flaking method and laminar production. Retouched pieces were classified mainly as side-scrapers and retouched Levallois points [10].

In the current excavations, lithic artifacts were found in all trenches in 13 sedimentary units (Tables 7–12). Artifacts from T1 Unit 1b, 2b, and T2 Unit 4 and 7 are vertically dispersed over

ca. 10 to 20 cm. The few artifacts from T4 and T3 are vertically dispersed over 40–50 cm thick sedimentary units. In T3, only two artifacts were recovered in Unit 1, 16 in Unit 2b, and 21 in Unit 4 (Table 11). Similarly, low numbers of artifacts were found in T4 Unit 2 (nine artifacts, of which five are < 2 cm), Unit 5 (53 artifacts; 39 of them < 2 cm), and Units 6 and 7 (five artifacts each) (Table 12). The low densities of small, dispersed artifacts in T4 and T3 mainly represent episodic use, maintenance, and discard events. In T1, the lithic artifacts were dispersed over ca. 40–50 cm at the undulating interface of Units 5–6. The artifact density is relatively low: 34 artifacts/m$^3$ (Fig 4; Tables 6 and 7). Lithic artifacts retain well-preserved dorsal scars and edges, and only minor apparent post-depositional alterations in all assemblages, except for artifacts from T1 at the Unit 5–6 interface. The edges of the artifacts from this interface are slightly more abraded, suggesting that they were more likely reworked and in secondary depositional context. All assemblages display MP techno-typological affinities and a low frequency of Levallois flaking, consistent with previous observations [10]. The only later intrusions at the site stem from Late Bronze Age burials that cut into the Paleolithic layers. No diagnostic artifacts indicating later prehistoric periods were found.

The recovered lithic artifacts were made from seven materials: obsidian, basalt, dacite, welded tuff, chert, limestone, and an unidentified metamorphic rock. Here we distinguish between obsidian and the rest of the materials (i.e., non-obsidian). Obsidian sourcing was conducted using pXRF (see below). Sourcing the non-obsidian materials is still in the initial stages and, as such, will not be detailed here. The frequencies of raw material types vary unit-by-unit among excavated assemblages (Tables 7–12).

Obsidian is the most common material with abundances ranging from 55% to 82% of the artifacts. A notable exception occurs in T2, Unit 4, where obsidian artifacts are much less common: 23.5% (Tables 7–12). This differs from the results of past excavations in the same layer, where obsidian constituted 59% of the artifacts [14: Table 1].

Within the obsidian component in all units, there are no indications of the first stages of core preparation and reduction, as there is a small number of cores and core trimming elements (CTEs) (Tables 7–12). Obsidian debitage consists mainly of flakes < 2 cm (up to 75% of artifacts in the assemblages; Tables 7–12). In the earlier excavations, flakes < 2 cm amounted to 61.7% of the obsidian component in T2 Layer 7 (Unit 4 in our new stratigraphic scheme) [10]. Among the obsidian retouched pieces, there is an extremely narrow range of tool types, and the majority can be classified as retouched points ([103]; see also [10], Fig 9, 1–7, and also Fig 8: 6,7).

Several technological features indicate a range of flaking practices associated with retouched obsidian pieces. "Shaping flakes" is an inclusive term used here to define all by-products of tool modification. Their identification is based on the presence of retouch scars on their dorsal faces. Retouch negatives are defined by their small size (< 0.5 cm), regularity in shape and distribution, and intrusiveness (Fig 8: 2–3). Shaping flakes retain prior retouch located on either their proximal (Fig 8: 2–3) or lateral edges (Fig 8: 1, 5). Most shaping flakes are < 2 cm in maximal dimension but are distinguished from "chips" that do not preserve visible retouch scars. Some of the shaping flakes were removed using soft-hammer percussion (antler or soft-stone [104] Figs 2–3), as many of them have diffuse bulbs of percussion (Fig 8: 2–5). The suggested use of soft hammer percussion to remove shaping flakes differs from that observed on most of the obsidian flakes, which exhibit characteristics of hard hammer percussion [104] Figs 2–3), such as prominent bulbs of percussion ([10], Fig 8: 6).

Shaping flakes may have commonly removed the tips of retouched convergent pieces. Some of the removed tips were detached from the direction of a side blow (Fig 8: 4–5). One shaping flake from T2 Unit 7 removed the distal part of a retouched point (Fig 8: 1). However, despite constant modification by retouching and removals of tips and retouched edges, the tool

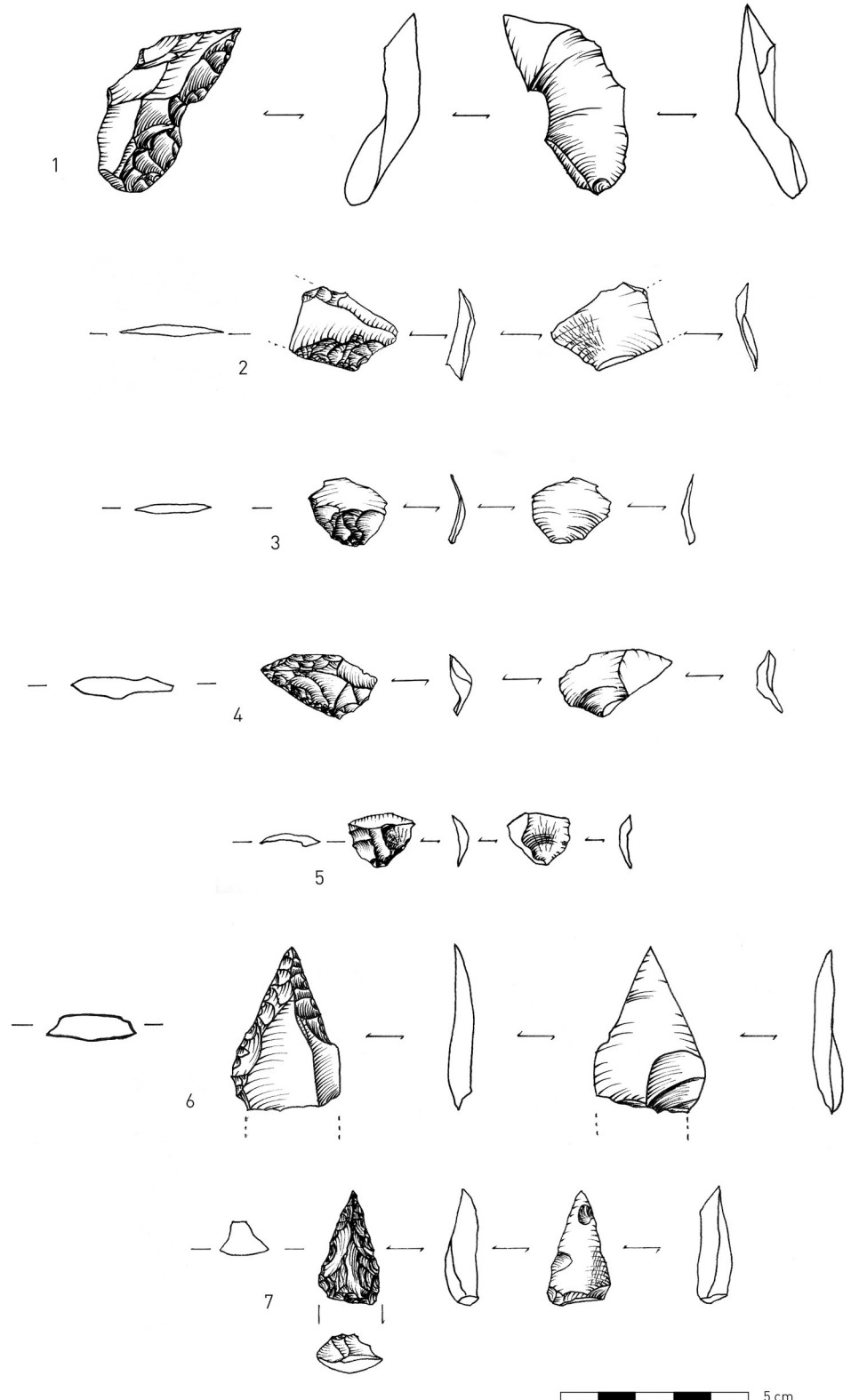

**Fig 8. Obsidian lithic artifacts.** 1, 4–5. Shaping flakes that removed the tips of convergent retouched points; 2–3. Shaping flakes; 6–7. Retouched points. (Illustration: P. Glauberman).

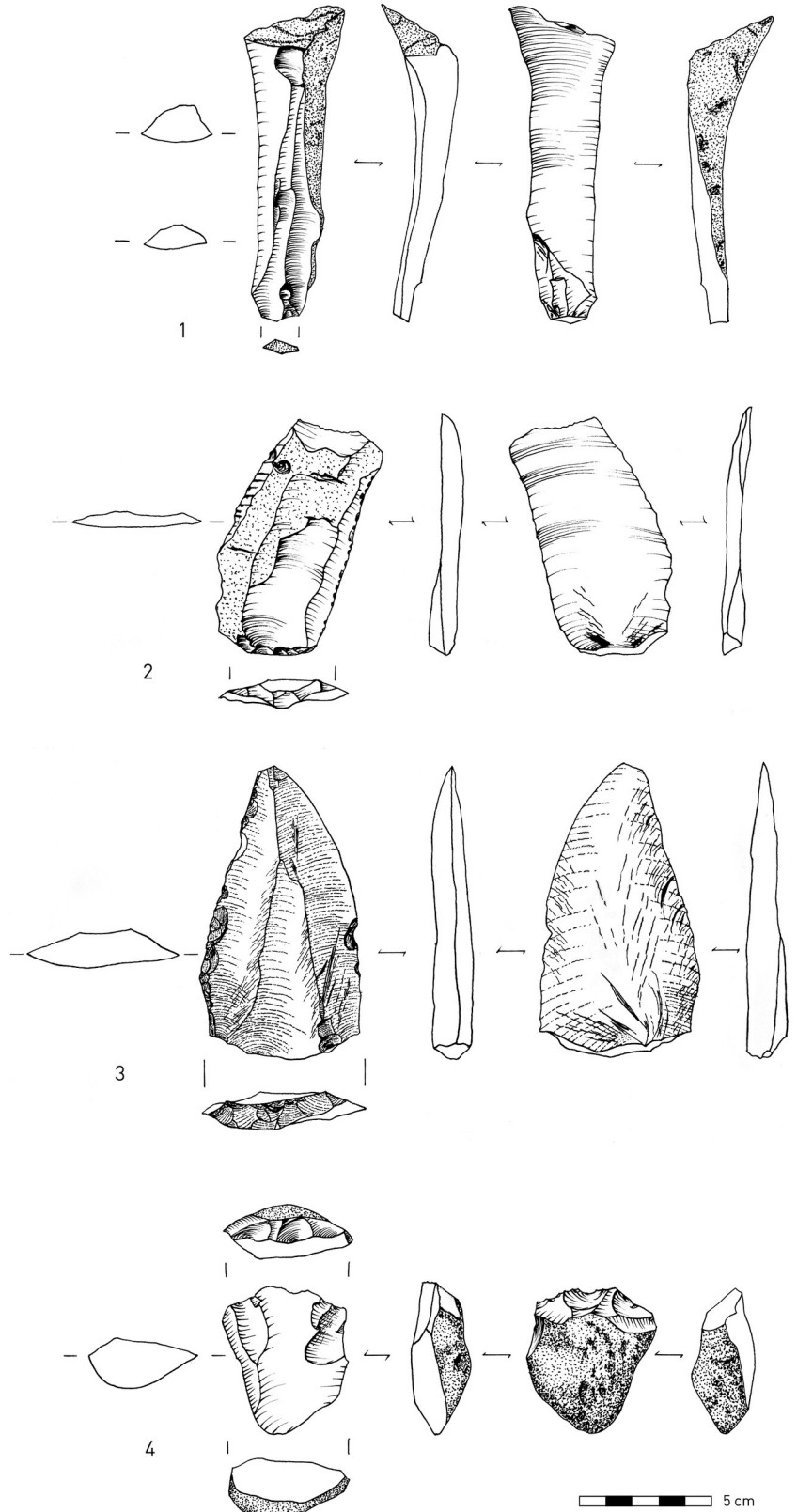

**Fig 9.** 1. Non-obsidian lithic artifacts: blade made on welded tuff; 2. Cortical flake on welded tuff; 3. Levallois point made on basalt; 4. Core on split pebble made on metamorphic material. (Illustration: P. Glauberman).

typology (i.e., a prevalence of pointed retouch) appears not to have changed through cycles of retouch and rejuvenation until discard. Further analysis and refitting will test these hypotheses.

The non-obsidian component exhibits greater indications of core reduction as attested by higher frequencies of cores, CTEs, and hammerstones. Moreover, there are also few indications of initial phases of core reduction, as indicated by a few cores (Fig 9: 4) and primary elements (flakes with >50% cortex, Fig 9: 2), (Tables 7–11). The few cores at the site were mostly made on flakes, while some were made on split cobbles (Fig 9: 4). Non-obsidian assemblages are characterized by flakes, Kombewa flakes, and Levallois flakes (Tables 7–10; Fig 9: 2–3). Levallois points appear in lower frequencies, for example, in Unit 1 in T1 (Fig 9: 3). In T2, Unit 4, sixteen blades (Fig 9: 1) comprise 4.4% of the assemblage, similar to findings from the prior excavations [[10] Fig 4, 8–7, 18]. Refitted blades were also found [[10] Fig 4, 15]. In contrast to the obsidian assemblages, the frequency of the objects < 2 cm among the non-obsidian artifacts in all assemblages is lower and never exceeds 50% (Tables 7–12).

In brief, the behavioral signal in all units indicates high intensities of obsidian tool reduction with repeated cycles of shaping and rejuvenation, while the non-obsidian component suggests core reduction and blank production occurred on-site.

**Lithic use-wear analysis.** A pilot study considered the preservation of use-wear on Kalavan 2 artifacts and the feasibility of its identification. Twenty-two obsidian and two non-obsidian artifacts were studied from T1 Unit 1b, while twelve obsidian and eight non-obsidian artifacts were studied from T2 Unit 4. Each analyzed artifact was individually packed in a plastic bag filled with ~ 100 ml of demineralized water and a non-ionic detergent (BASF Plurfac LF901, 1 g/l = 1% w/v; BASF SE, Ludwigshafen, Germany). The closed bags were put into a preheated ultrasonic bath (EMAG Emmi 20HC). The samples were left in the bath for 15 min at 40°C and 100 KHz (see S3 File for the protocols and methodology of the use-wear analysis). This sample included the most representative techno-typological categories. Obsidian artifacts (n = 34): including flakes (n = 2), fragments smaller than 2 cm pieces (21), shaping flakes (5), and tools (6), as well as non-obsidian flakes (8) and cores (2) (see S3 File for the protocols and results of the analysis).

Obsidian surfaces and edges, due to the brittle properties of the raw material, are more prone to post-depositional alteration compared to non-obsidian materials. The non-obsidian artifacts, though, did not appear to preserve any microwear traces except for one core on a split pebble. Macroscopic traces of percussive activities were identified on this chert core on a split pebble (Fig 9: 4). Among the analyzed obsidian artifacts, nine out of 34 presented specific distributions and orientations of striations and wear polish, indicative of use. Such microwear traces were observed on obsidian scrapers, points, and shaping flakes. Previous research [105, 106] suggests that such orientations, distributions, and types of polish and/or striations on scrapers can be related to hard animal material (Fig 10A and 10B). On retouched points and shaping flakes, the use-wear patterns point to woodworking activities [105, 106] (Fig 10C and 10D). These preliminary findings hint that the shaping flakes and the point fragments may reflect continuous rejuvenation during activities performed on worked materials and motions related to woodworking. This proposal is preliminary, and must still be further tested through detailed techno-typological, use-wear, and refit analyses on a larger and more comprehensive sample size.

In T2, Unit 4, a group of four stone objects, unique in raw material among lithic artifacts (limestone and dacite) and sizes in comparison to the bulk lithic assemblage, were found in close spatial association near a concentration of broken bones (Fig 11; see SP3). A preliminary analysis shows the presence of damaged areas characterized by surface crushing/fracturing and abrasive traces (i.e., striations, scratching), which are reflected in abrupt alterations on the

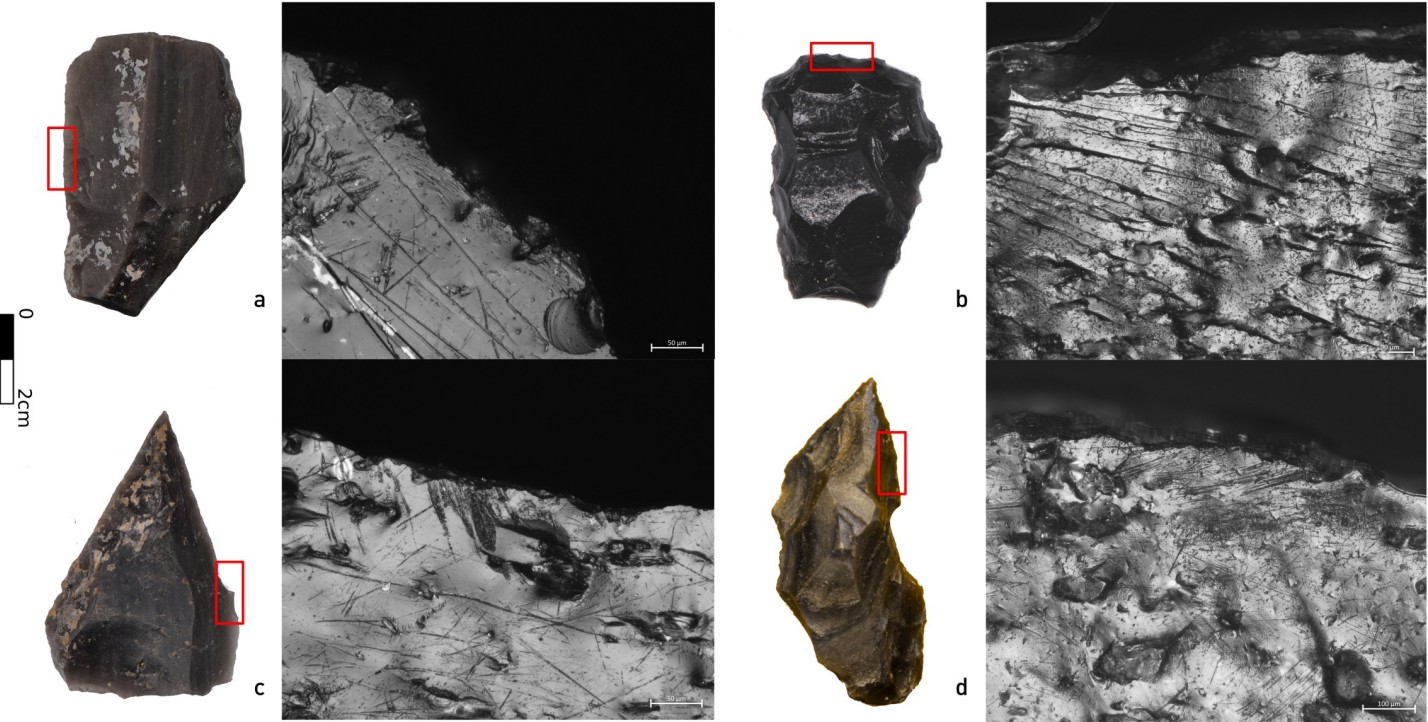

**Fig 10. Fragment of a retouched blade (upper left) and end-scaper (upper right) with striations and polish associated with hard animal material.** A retouched point (bottom left, Fig 8:6) and a shaping flake of a retouched point (bottom right, Fig 8:1) showing striations and polish diagnostic of woodworking activities.

surface topography (Fig 11). The location and distribution of these alterations suggest that the rocks could have been used as anvils for percussive activities. Prior use-wear and residue studies, as well as experimental work on limestone and dacite, have linked pitted anvils to plant processing [107, 108], bone processing [109–112], or bipolar lithic production [113]. Since there is no evidence for bipolar production in the lithic assemblages, the most likely functions of the pitted stones were for other purposes, such as processing of vegetal materials and/or faunal resources, which will be tested in the future (see [114]). Further microscopic analyses supported by experiments using materials with similar features are possible avenues for better understanding the function of these artifacts.

## Obsidian sourcing

Geochemically matching obsidian artifacts to their volcanic origins across the Armenian highlands offers a means to reconstruct the mobility of the site's occupants. We analyzed 928 obsidian artifacts following our published pXRF protocols [60–62]. The identified obsidian sources principally reflect two mountainous areas of central Armenia (Fig 12; S3 File, Fig 1)–the Tsaghkunyats (Kamakar, Ttvakar, and Damlik) and Gutansar, Hatis, and Gegham volcanic range (Geghasar)–followed by sources in the Syunik area of southern Armenia (Satanakar and Sevkar). Artifacts from more distant obsidian sources in western Armenia (Pokr Arteni), eastern Turkey (Digor 1 in the Kars region), and southern Georgia (Chikiani) are also present within the analyzed sample. Fig 13 breaks down the major source areas in each trench by stratigraphy to highlight diachronic trends, such as, for example, the different proportions of source areas in T2 Units 4 and 7.

These results allow us to largely describe the movements of the Kalavan 2 visitors in what is now central Armenia, in particular the vicinity of the Hrazdan River, which acts as an interface

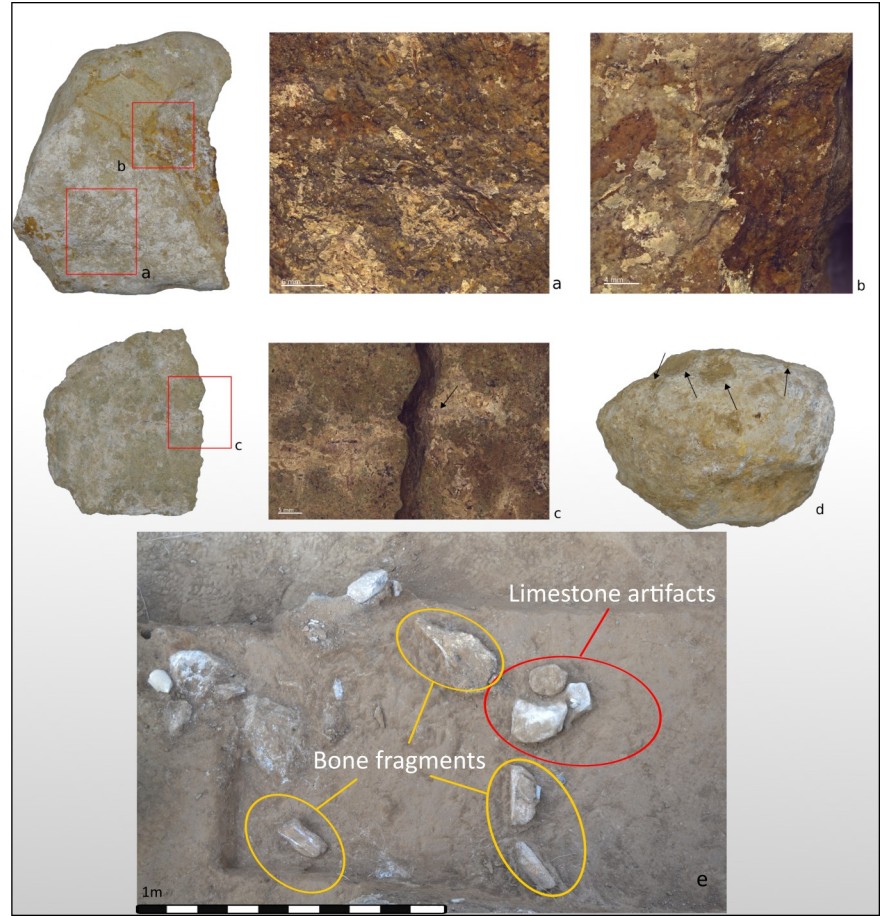

**Fig 11.** Limestone pitted anvil showing evidence of a) impact marks and b) surface crushing. Bottom: c) Limestone slab with abraded surface and percussion impact feature; d) Limestone pitted anvil with several impact marks and multidirectional flake scars resulting from percussive actions (arrows indicate the orientation of the flakes removal); e) The archeological context of the pitted anvils (red circle indicating pitted anvil; yellow circles indicating bone fragments).

between the Gegham and Tsaghkunyats ranges. Gutansar, Hatis, and the three Tsaghkunyats sources are all ca. 50–60 km linearly from Kalavan 2; however, these distances increase to ≥ 120–140 km on foot based on topography. The Syunik sources and Pokr Arteni are 120–130 km from the site linearly, but each of these distances is ≥ 200 km along the most direct routes. The two farthest sources are Chikiani in Georgia (150 km linearly, ≥ 240 km on foot) and Kars-Digor 1 in Turkey (160 km linearly, ≥ 200 km on foot).

Fig 14 breaks down the sourcing results by lithic artifact class for all four trenches. For simplicity, obsidian sources are grouped by their source areas (e.g., Kamakar, Ttvakar, and Damlik as the Tsaghkunyats sources). The knapping quality of the obsidian (e.g., glassiness, lack of mineral phenocrysts) does not differ among the identified sources. Thus, the variation in the raw material abundance does not reflect a preference based on flaking properties or mechanical performance. Instead, it reflects mobility patterns coupled with spatiotemporally distributed production and discard of different tools as needed.

It would be expected that, for any particular lithic class, Geghasar, as well as Gutansar and Hatis obsidians should be most abundant, followed by Tsaghkunyats obsidians and, finally, Syunik obsidians. Most artifact classes indeed fit such a pattern as demonstrated in Fig 14. An

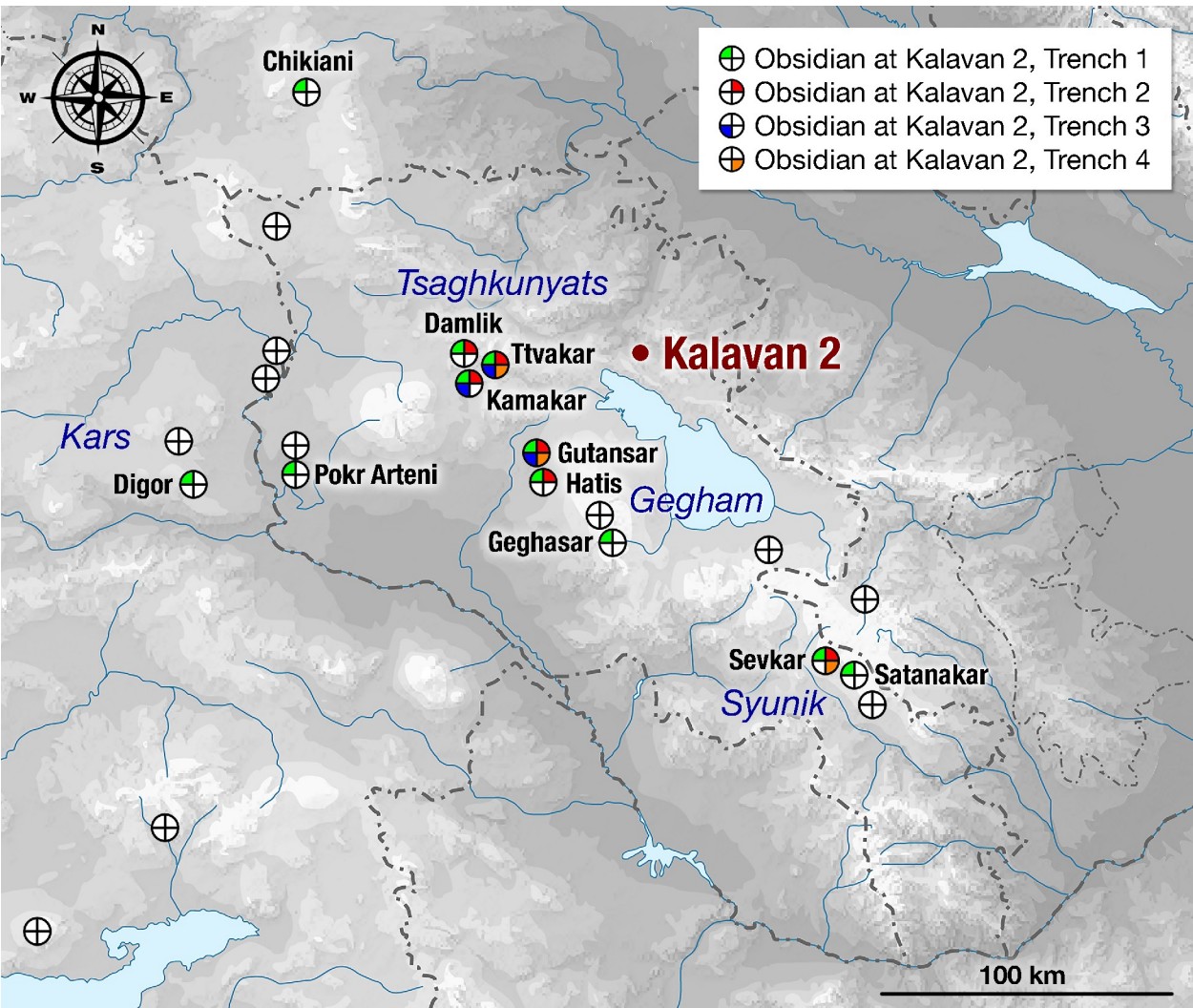

**Fig 12. Locations of obsidian sources relative to Kalavan 2 (SRTM3 digital elevation data).** Presence or absence of the obsidians at Kalavan 2 is represented by each source's circular symbol. The empty circles correspond to obsidian sources that are not currently reflected in the Kalavan 2 assemblage.

exception is a Levallois flake from the Sevkar source in the Syunik area. Bladelets are another exception: the numbers of Geghasar, Gutansar, Hatis, and Tsaghkunyats bladelets are equal, but the sample is small. Thus, we do not interpret such an observation as a stronger preference for Tsaghkunyats obsidian for bladelets relative to other lithic classes. The three distant obsidian sources are each represented by one (Pokr Arteni and Kars-Digor 1) or two (Chikiani) artifacts smaller than 2 cm, whereas a single tool was transported from Pokr Arteni and Chikiani.

## Faunal remains

The faunal assemblage from the prior excavations consists mainly of bovid (cf. *Bos primigenius*), horse (*Equus caballus*), wild goat/ibex (*Capra* sp.), and red deer (*Cervus elaphus*). The anthropogenic origin of this assemblage was suggested by the presence of cut marks on a small number of bones and a few impact fractures on bovid bones [10]. Our excavations yielded > 2163 bone specimens (i.e., number of specimens [NSP], following Lyman [115]:

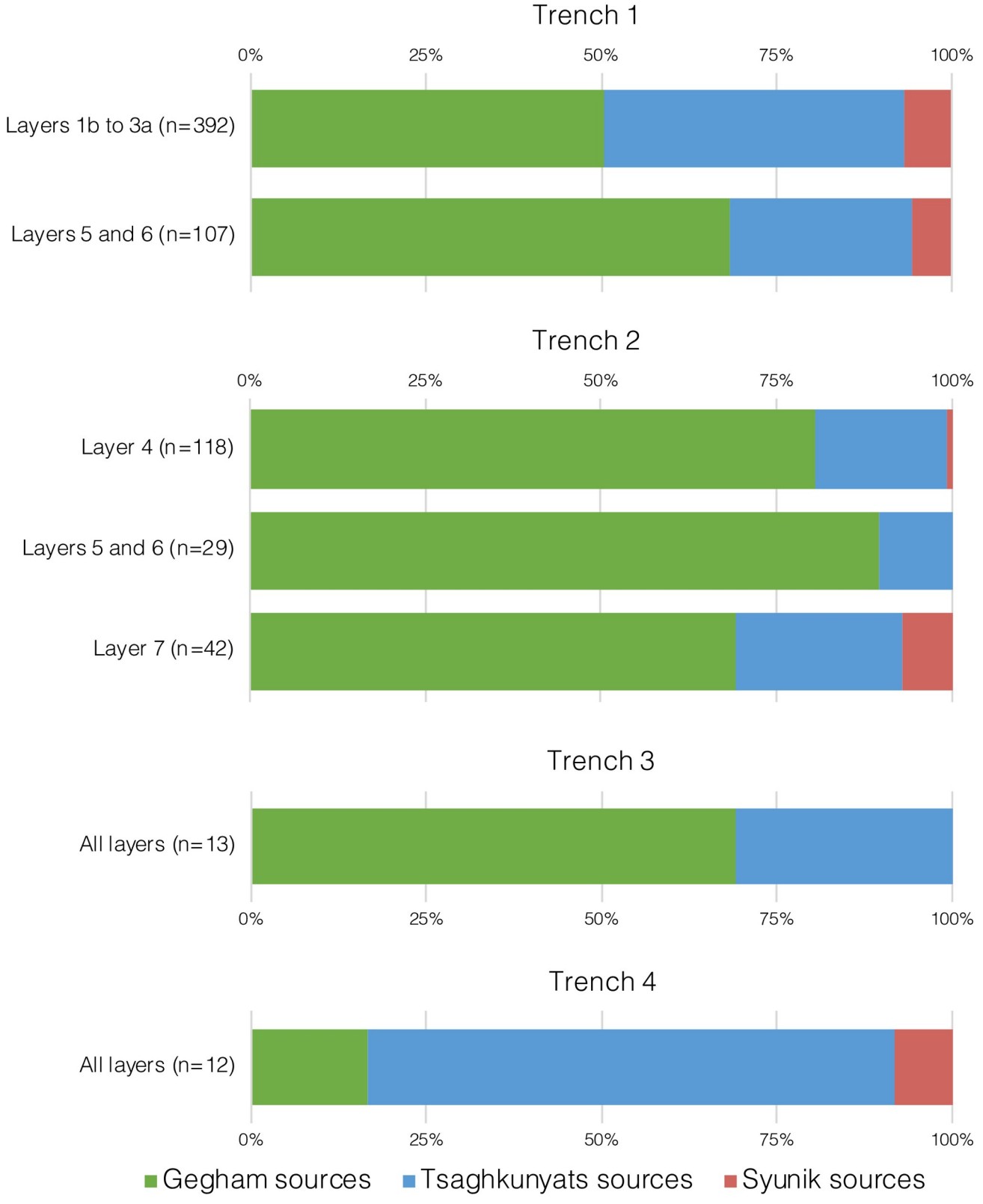

**Fig 13. Obsidian sources and the corresponding source areas identified at Kalavan 2 by trench.**

Table 13). The bones were cleaned in the field, and a basic taxonomic/taphonomic analysis was performed (see S4 File for methods). Faunal remains were excavated in all trenches and retrieved from 15 units (Table 13). The two richest faunal assemblages derive from T2 Unit 4 (330/m$^3$; 39.5% of the entire NSP) and T1 Unit 1b (225/m$^3$; 35.3% of the entire NSP). The assemblages consist mostly of highly fractured long bones in different weathering stages (Table 14) and display various stages of weathering: outer surfaces are often heavily affected by root etching, decalcification, and carbonate coating. Taxonomic and bone element identifications, as well as preserved modifications, are limited. Thus, anthropogenic modifications are scarce. No carnivore gnawing was observed, while a minor frequency of rodent gnawing could be identified (T1 Unit 5: n = 2; T1 Unit 3a: n = 1; T1 Unit 1b: n = 3; T2 Unit 4: n = 3).

To date, 79 bones have been assigned to taxon (Table 15). Among the larger ungulates, *Bos/Bison* and a horse (*Equus* sp.) are most prevalent. Three molars belong to *Equus hemionus*, indicating the presence of an additional smaller Equid among the bones. Ten bones, including an antler fragment, belong to *Cervus elaphus*, and nine bones to the subfamily Caprinae (*Capra* and/or *Ovis*). Due to the limited taxonomic identification, bones were attributed to size classes (i.e., categorizing the different life weights of different species; see S4 Table 1 in S4 Table

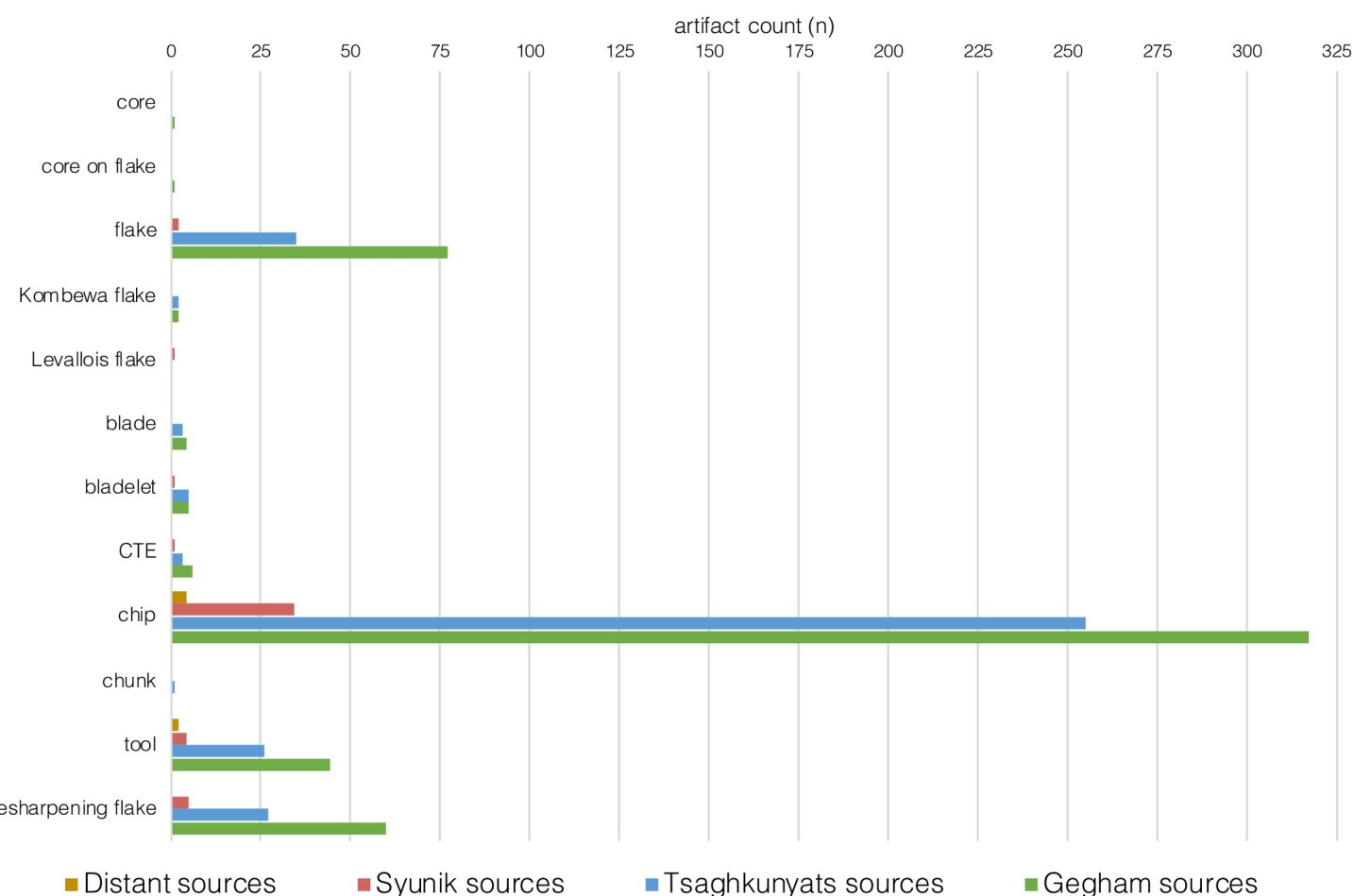

**Fig 14. Obsidian source areas identified in all trenches by lithic class.**

**Table 14. Number of individual specimens (NISP) according to anatomical region by trench and sedimentological unit.**

| Anat. region | T1 | | | | | | |
|---|---|---|---|---|---|---|---|
| Unit | 1b | 2a | 2b | 3a | 3b | 5 | |
| Cranial | 12 | 1 | 3 | | | 4 | **20** |
| Axial | 11 | 1 | 3 | 1 | | 2 | **18** |
| Long bone | 83 | 2 | 2 | 2 | 1 | 6 | **96** |
| Autopodium | 8 | | 1 | 2 | | 2 | **13** |
| | 114 | 4 | 9 | 5 | 1 | 14 | **147** |
| | T2 | | | | | | |
| Unit | 3 | 4 | 5 | 5/6 | 6 | 7 | |
| Cranial | | 17 | | 1 | | 1 | **19** |
| Axial | | 4 | | | | | **4** |
| Long bone | 2 | 10 | 1 | | | | **11** |
| Autopodium | | 4 | | | 1 | | **5** |
| | **2** | **35** | **1** | **1** | **1** | **1** | **39** |
| | T3 | | | | | | |
| Unit | 2b | 4 | | | | | |
| Cranial | 2 | | | | | | **2** |
| Axial | 1 | | | | | | **1** |
| Long bone | | 1 | | | | | **1** |
| Autopodium | | | | | | | |
| Total | **3** | **1** | | | | | **4** |
| | T4 | | | | | | |
| | 2 | 5d | 6b | 7 | 8 | | |
| Unit | | | | | 1 | | **1** |
| Cranial | | 1 | | | 1 | | **2** |
| Axial | | | 1 | | | | **1** |
| Long bone | 1 | | | 2 | 1 | | **4** |
| Autopodium | | | | | | | **0** |
| Total | **1** | **1** | **1** | **2** | **3** | | **8** |

Anat. region = anatomical region.

[116, 117]). Small cervids (roe deer), goat, and sheep with a weight range of 20–120 kg are attributed to size class 2. Size class 3 represents a range of 120–260 kg of medium ungulates (e.g., red deer, reindeer, fallow deer, wild ass, boar). The size class 4 category includes the larger horses and *Bos/Bison* (ca. 300–1000 kg). Thus, 677 additional bones could be assigned to a size class based on bone robusticity, cortical bone thickness, and morphology (Table 16). The results confirm the predominance of larger taxa (*Bos/Bison* and *Equus* sp.) in all faunal assemblages.

Cut marks were observed on at least three long bone fragments with well preserved cortical surfaces from T2 Unit 4 (Fig 15). In addition, ten bones with possible cutmarks were found in T1 Unit 1b. Long bone shafts with percussion impacts, possibly resulting from bone breakage for marrow extraction were found in T2 Units 4 and 5 and T1 Unit 1b (Fig 16). Furthermore, eight bones with possible impacts, as well as a few bones that demonstrate color changes potentially related to burning and/or staining [118, 119], were found. In T1, such bones were retrieved from Units 5 (n = 4), 2b (n = 1), and 1b (n = 12). Some were calcined, similar to observations made by the prior excavators where they interpreted seven surface modifications as traces of burning [10]. These bones suspected of having been burnt will be investigated

**Table 15. NISP of Species by trench and sedimentological unit.**

| T 1 | Species | | | | | |
|---|---|---|---|---|---|---|
| Unit | Bos/Bison | Equus sp. | Equus hemionus | Cervus elaphus | Capra/Ovis | Total |
| 5 | 1 | | | | 1 | 2 |
| 3a | 1 | | | | | 1 |
| 2b | 3 | 1 | | | | 4 |
| 2a | 1 | | | | | 1 |
| 1b | 17 | 7 | | 5 | 2 | 31 |
| Total | 23 | 8 | | 5 | 3 | 39 |
| T2 | Species | | | | | |
| Unit | Bos/Bison | Equus sp. | Equus hemionus | Cervus elaphus | Capra/Ovis | |
| 7 | | 1 | | | | 1 |
| 5/6 | | | 1 | | | 1 |
| 5 | 1 | | | | | 1 |
| 4 | 16 | 6 | 2 | 4 | 1 | 29 |
| Total | 17 | 7 | 3 | 4 | 1 | 32 |
| T3 | Species | | | | | |
| Unit | Bos/Bison | Equus sp. | Equus hemionus | Cervus elaphus | Capra/Ovis | |
| 2b | 2 | | | | | 2 |
| Total | 2 | | | | | 2 |
| T4 | Species | | | | | |
| Unit | Bos/Bison | Equus sp. | Equus hemionus | Cervus elaphus | Capra/Ovis | |
| 8 | | | | | 3 | 3 |
| 6b | | | | | 1 | 1 |
| 5d | | | | | 1 | 1 |
| 2 | | | | 1 | | 1 |
| | | | | 1 | 5 | 6 |

using Fourier Transform infrared spectrometry (FTIR) as a means to either validate or refute hypotheses of thermal alteration [120].

Following the protocols of Sánchez-Hernández [121, 122], we sought to discern the seasons at which the animals died by combining meso- and microwear analyses. For animals with seasonal diets, mesowear analyses show an average-diet signal, and the microwear signal corresponds to the seasonal diet. For example, in a given layer, if the animals died in different seasons (e.g., a long-term occupation site), the microwear signal would encompass the full dietary range, and microwear and mesowear analyses would converge. But if all animals died in the same season, be it in the same year or over several different years (e.g., a short-term seasonal hunting site), the microwear signal would correspond to this particular season only, and signals from microwear and mesowear would instead diverge. Ten teeth have been recovered from three trenches (*Bos/Bison*: n = 4; *Equus hemionus*: 3; *Cervus elaphus*: 1; *Equus* sp.: 2), and only six are in suitable condition for wear analysis. Just four individual teeth can be studied using both wear methods (see S4 Table 2 in S4 Table), so the current sample size is not sufficient to produce meaningful results based on these methods (see S4 File). Thus, the raw data were not analyzed.

## Micro-faunal remains

The micro-faunal analysis (see S4 File for detailed methods) is based on NISP and the minimum number of individuals (MNI) obtained from 138 molars (Table 17). NISP and MNI include all identifiable dental elements, including upper and lower molars. For MNI, the most

**Table 16. Number of specimens (NSP) of size class by trench and sedimentological unit.**

| T1 | Size class | | | | | | |
|---|---|---|---|---|---|---|---|
| **Unit** | **4** | **3 or 4** | **3** | **2 or 3** | **2** | **1** | **Total** |
| **5** | 5 | 5 | 5 | 11 | 4 | | **30** |
| **3b** | 1 | | | | | | **1** |
| **3a** | 3 | 3 | 2 | | | | **8** |
| **2b** | 6 | 2 | 2 | | | | **10** |
| **2a** | 2 | | 1 | 2 | 1 | | **6** |
| **1b** | 106 | 37 | 30 | 25 | 21 | | **219** |
| **Total** | **123** | **47** | **40** | **38** | **26** | | **274** |
| **T2** | Size class | | | | | | |
| **Unit** | **4** | **3 or 4** | **3** | **2 or 3** | **2** | **1** | |
| **7** | | 3 | | 1 | | | **4** |
| **6** | | 11 | | | 1 | | **12** |
| **5/6** | 2 | 8 | 1 | | 1 | | **12** |
| **5** | 1 | 9 | | | | | **10** |
| **4** | 123 | 148 | 36 | 16 | 27 | 1 | **351** |
| **3** | | | 2 | | | | **2** |
| Total | **126** | **179** | **39** | **17** | **29** | **1** | **391** |
| **T3** | Size class | | | | | | |
| **Unit** | **4** | **3 or 4** | **3** | **2 or 3** | **2** | **1** | **Total** |
| **4** | | | | 1 | | | 1 |
| **2b** | 2 | 1 | | | | | 3 |
| **Total** | **2** | **1** | | **1** | | | **4** |
| **T4** | Size class | | | | | | |
| **Unit** | **4** | **3 or 4** | **3** | **2 or 3** | **2** | **1** | **Total** |
| **8** | | | | 3 | | | **3** |
| **7** | | | | | 2 | | **2** |
| **6b** | | | | 1 | | | **1** |
| **5d** | | | | 1 | | | **1** |
| **2** | | | 1 | | | | **1** |
| **Total** | **-** | **-** | **1** | **5** | **2** | **0** | **8** |

abundant single element is used (e.g., all lower-left M1, all upper-right M1). The microfaunal assemblage of the site is dominated by the common hamster (*Cricetus cricetus*) and also includes the Turkish hamster (*Mesocricetus brandti*), molevole (*Ellobius lutescens*), European watervole (*Arvicola terrestris*), common or field vole (*Microtus arvalis/agrestis*), and single occurrences of the common shrew (*Sorex* cf. *araneus)* and marmot (*Marmota* sp.*).

In T4, a large number of post-cranial elements are attributed to the common hamster, and the high rate of completeness, lack of digestive marks, and grouping of skeletal elements suggests a non-predatory driven assemblage (following [123]). The presence of unfused bones in the distal femur, proximal tibia, and distal humerus (S4 Table 3 in S4 Table) indicate the presence of juvenile hamsters (6–12 months old) [124]. Thus, post-cranial elements in T4 suggest that the common hamster may have been accumulated via fossorial death during winter months. T1 to T3, in contrast, are mainly predator assemblages based on prevalent digestive corrosion and breakage (except for elements from the common hamster). Thus, it is assumed that the small mammals are contemporaneous with their respective units, although no changes are observed in faunal composition among units (Table 17).

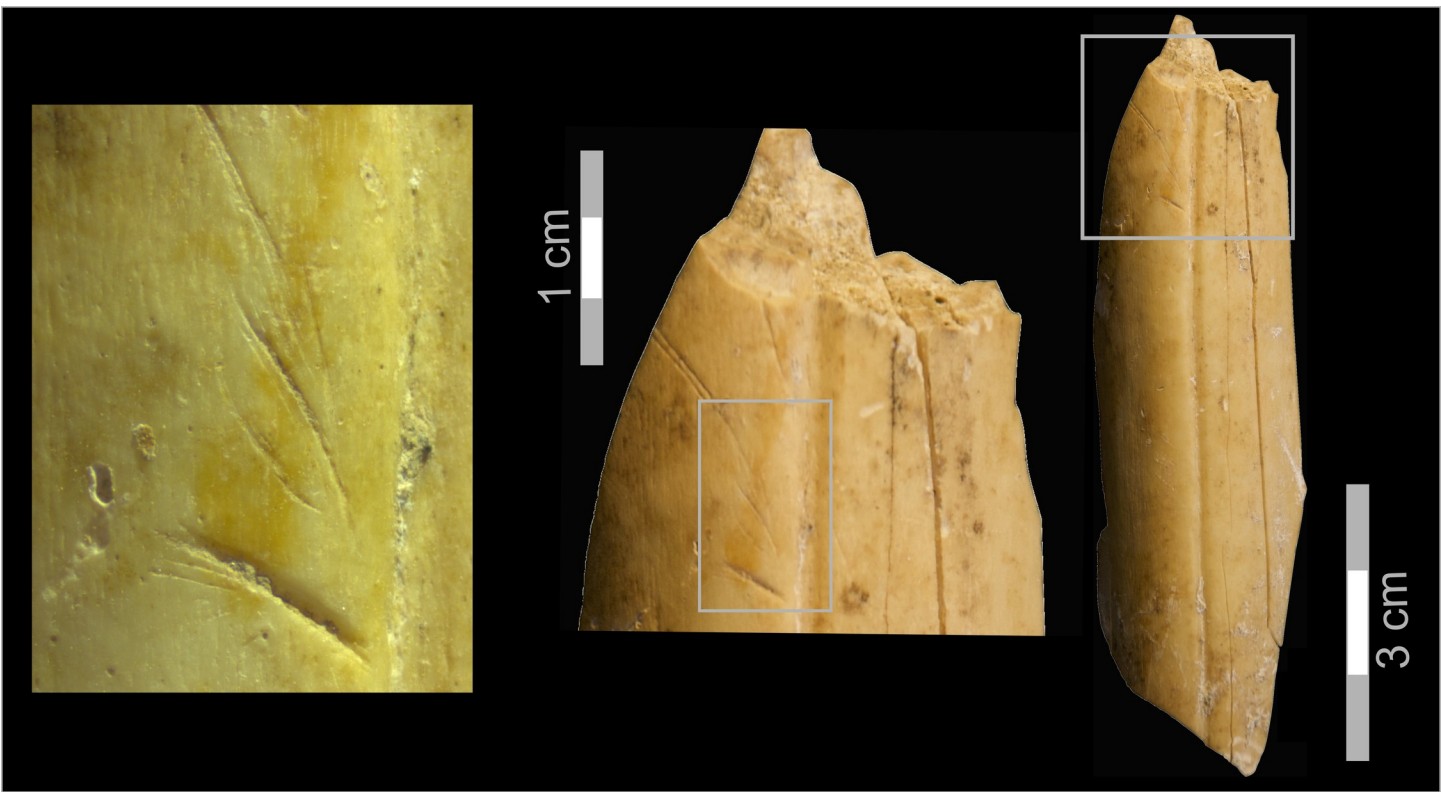

**Fig 15. Kalavan 2 cervid (red deer sized) metacarpus with multiple cut marks (Id: N50-44).**

## Paleovegetation proxies

Two proxies were analyzed to reconstruct the environment during site occupations. The first is identifying pollen preserved in the sediments, and the second is analyzing leaf wax-derived lipid biomarkers. Among wax lipids, *n*-alkanes have become popular leaf-derived biomarkers due to their relatively high persistence against degradation (i.e., insolubility in water, chemical inertness) and their potential to serve as molecular proxies for paleovegetation and paleoclimate reconstruction.

**Pollen.** The studied sediment samples from all trenches mostly contained 150 pollen grains or fewer. Only two samples from T4 Unit 5 had more than 150 grains (S2 Table 1 in S2 Table). The results from T1, T2, and T3 suggest either no preservation of pollen or biased preservation dominated by Asteraceae Cichorioideae (T2 Unit 6, T4 Unit 7, T4 Unit 5c, T4 Unit 5b). The pollen from Chicoroideae is one of the known resistant pollen types with *Tilia*, Brassicaceae, Caryophyllaceae, Asteraceae that persists despite unfavorable preservation conditions [125]. In the two counted T4 samples, the proportion of Asteraceae Chicoroideae is lower (14.8 and 62.5%) than in the previously mentioned samples. The amount of tree pollen is not significant, while the steppic taxa of Amaranthaceae, *Centaurea*, and *Artemisia* are more prevalent (S2 Table 1 in S2 Table). The depositional conditions of T1, T2, and T3 differ from those of T4. At T4, possible successive paleosols accumulation provided less oxic conditions that enabled pollen grain preservation; however, the studied pollen from T4 is most likely modified from the original assemblage due to post-depositional processes. Recorded taxa characterized an open steppic landscape.

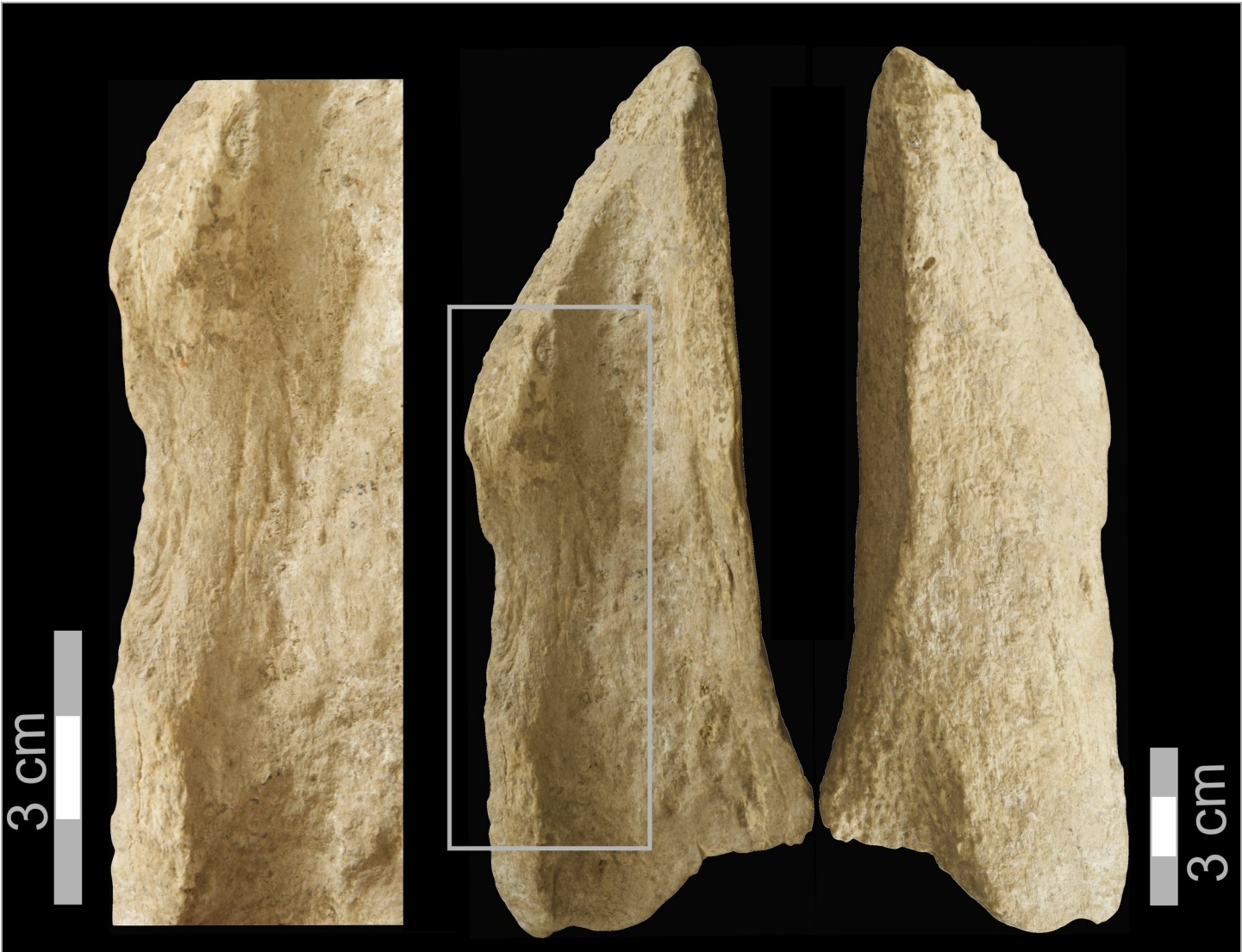

**Fig 16. Kalavan 2: Large bovid femur diaphysis with impact notch (Id: A77-1).**

**Leaf wax (*n*-alkanes) analysis.** In the Armenian highlands and the southern Caucasus, the distribution of *n*-alkanes has been demonstrated to differentiate between soils formed under deciduous vegetation and grasses [23, 126]. Soils under grasses typically have higher average chain length (ACL) values (~30), and the most abundant *n*-alkane is $n$C$_{31}$. Soils under deciduous trees have lower ACL values (~29), and the most abundant n-alkane is $n$C$_{29}$. Grassy soils also typically have $n$C$_{27}$/$n$C$_{33}$ ratios <1, whereas deciduous soils have $n$C$_{27}$/$n$C$_{33}$ ratios >1. There is no significant difference in a metric known as odd-over-even predominance (OEP) between deciduous (8.6) and grassy (8.4) soils. Such high OEP values, though, indicate that modern soils in the Armenian highlands and Southern Caucasus have not undergone significant bacterial or thermal alteration, which tends to lower OEP values [127].

Seventeen samples were taken from 17 sedimentary units in T1-T3 (see S2 File and Table 18 for method and sampling). The distribution of *n*-alkanes shows fluctuating environmental conditions during sedimentation and occupation of the site (Table 18). High OEP

**Table 17. Analysed dental elements per trench and unit.**

**T1**

| Unit | cf. Cricetus | | Cricetus cricetus | | Microtus arvalis/agrestis | | Mesocricetus | | Microtus sp. | | Arvicola terrestris | | Ellobius lutescens | | cf. Ellobius | | Marmota sp. | | Total | |
|---|---|---|---|---|---|---|---|---|---|---|---|---|---|---|---|---|---|---|---|---|
| | NISP | MNI | NISP | MNI | NISP | MNI | NISP | MNI | NISP | MNI | NISP | MNI | NISP | MNI | NISP | MNI | NISP | MNI | NISP | MNI |
| Unit 1b | | | | | | | | | | | | | | | | | | | | |
| 5 or 6 | | | 2 | 2 | 1 | 1 | 1 | 1 | | | 1 | 1 | | | | | | | 5 | 5 |
| Total | | | 2 | 2 | 1 | 1 | 1 | 1 | | | 1 | 1 | | | | | | | 5 | 5 |

**T2**

| Unit | cf. Cricetus | | Cricetus cricetus | | Microtus arvalis/agrestis | | Mesocricetus | | Microtus sp. | | Arvicola terrestris | | Ellobius lutescens | | cf. Ellobius | | Marmota sp. | | Sorex araneus | | Total | |
|---|---|---|---|---|---|---|---|---|---|---|---|---|---|---|---|---|---|---|---|---|---|---|
| | NISP | MNI | NISP | MNI | NISP | MNI | NISP | MNI | NISP | MNI | NISP | MNI | NISP | MNI | NISP | MNI | NISP | MNI | NISP | MNI | NISP | MNI |
| 7 | | | 1 | 1 | | | | | | | 3 | 2 | | | | | | | | | 4 | 3 |
| 4 | 1 | | 8 | 5 | | | | | | | | | 6 | 3 | 1 | 1 | | | | | 16 | 9 |
| 3 | | | | | 1 | 1 | | | 1 | 1 | | | | | | | | | | | 2 | 2 |
| 2 | | | | | 2 | 2 | | | | | | | | | | | | | | | 2 | 2 |
| Total | 1 | 0 | 9 | 6 | 3 | 3 | 0 | 0 | 1 | 1 | 3 | 2 | 6 | 3 | 1 | 1 | 0 | 0 | 0 | 0 | 24 | 16 |

**T4**

| Unit | cf. Cricetus | | Cricetus cricetus | | Microtus arvalis/agrestis | | Mesocricetus | | Microtus sp. | | Arvicola terrestris | | Ellobius lutescens | | cf. Ellobius | | Marmota sp. | | Sorex araneus | | Total | |
|---|---|---|---|---|---|---|---|---|---|---|---|---|---|---|---|---|---|---|---|---|---|---|
| | NISP | MNI | NISP | MNI | NISP | MNI | NISP | MNI | NISP | MNI | NISP | MNI | NISP | MNI | NISP | MNI | NISP | MNI | NISP | MNI | NISP | MNI |
| 6a | | | 2 | 1 | | | | | | | | | | | | | | | | | 2 | 1 |
| 5d | 1 | 1 | 2 | 1 | 2 | 1 | | | | | | | | | | | | | | | 5 | 3 |
| 5c | 3 | 1 | 14 | 4 | 6 | 4 | 2 | 1 | 6 | 1 | 2 | 1 | | | | | | | | | 33 | 12 |
| 5b | | | 7 | 2 | 2 | 2 | | | 2 | 1 | | | | | | | | | 1 | 1 | 12 | 6 |
| 5a | 2 | 1 | 8 | 2 | 1 | 1 | | | | | | | | | | | | | | | 11 | 4 |
| 4 | 1 | 1 | 10 | 5 | 2 | 1 | | | 1 | 1 | 5 | 3 | 1 | 1 | | | 1 | 1 | | | 21 | 13 |
| 3 | 2 | 1 | 3 | 2 | | | 1 | 1 | 3 | 1 | 2 | 1 | 1 | 1 | | | | | | | 12 | 7 |
| 2 | 1 | 1 | 3 | 2 | | | | | | | | | 1 | 1 | 2 | 1 | | | | | 7 | 5 |
| Total | 10 | 6 | 49 | 19 | 13 | 9 | 5 | 4 | 12 | 4 | 9 | 5 | 3 | 3 | 2 | 1 | 1 | 1 | 1 | 1 | 103 | 51 |
| Total of excavation | | | | | | | | | | | | | | | | | | | | | #REF! | #REF! |

The NISP (Number of Identified Specimens) and MNI (Minimum Number of Individuals) are shown per taxon.

**Table 18. Results of *n*-alkane quantification from sedimentary units in Trenches 1, 2 and 3 at Kalavan-2 1: Estimate of percent grass from Bliedtner et al 2018.** Values of >100% are possible due to uncertainties in end-member values, see Bliedtner et al 2018 for a full explanation. 2: Landscape interpretation based on grass estimates: deciduous (0–33% grass), mixed (33–66%) and grass (>66%).

| Trench | Unit | ACL | OEP | Grass (%)[1] | Landscape[2] |
|--------|------|-----|-----|----------|-----------|
| 1 | 6 | 29.8 | 9.9 | 75% | Open |
| | 5 | 29.3 | 9.1 | 52% | Mixed |
| | 4 | 29.8 | 10.6 | 69% | Grass |
| | 3b | 29.7 | 11.0 | 63% | Mixed |
| | 2a | 29.7 | 10.3 | 66% | Grass |
| | 1b | 29.6 | 7.2 | 77% | Grass |
| 2 | 7 | 30.5 | 9.6 | 101% | Grass |
| | 5 | 29.3 | 6.6 | 17% | Deciduous |
| | 4 | 29.5 | 9.2 | 62% | Mixed |
| | 2a | 29.7 | 4.7 | 87% | Grass |
| | 1b | 29.1 | 5.3 | 62% | Mixed |
| 3 | 4a | 30.1 | 12.1 | 90% | Grass |
| | 3b | 29.2 | 8.1 | 50% | Mixed |
| | 3a | 29.8 | 10.8 | 73% | Grass |
| | 2b | 29.2 | 7.8 | 18% | Deciduous |
| | 2a | 30.1 | 8.2 | 92% | Grass |
| | 1 | 29.9 | 8.1 | 94% | Grass |

values throughout the entire sequence indicate little to no post-depositional alteration of the *n*-alkanes, suggesting a well-preserved environmental signal (S2 Fig 10 in S2 Fig). Most samples align with present-day grassy landscapes or indicate a mix of grass and deciduous sources of *n*-alkanes (S2 Fig 11 in S2 Fig), as the concentration of $nC_{31}$ and $nC_{33}$ is typically higher than $nC_{27}$ and $nC_{29}$). In the units dated ca. 50–60 ka, most samples either overlap with present-day grass soils or fall between the disruption of grass and deciduous soils with one exception (T2 Unit 2b), which has a low ACL value that overlaps with present-day deciduous soils (S2 Fig 11 in S2 Fig).

## Discussion

Our data elucidate hominin behavior at Kalavan 2 and its environmental background. New sedimentological, chronometric, and paleoenvironmental proxy data permit us to contextualize hominin occupations and assess site formation processes. Archaeological evidence would test seasonal occupational patterns hypothesized across the region's topographic gradients. This evidence further indicates hominin activities, intensity of site occupation, and the role of the site within regional late MP settlement dynamics.

### Depositional environments and site formation

Four excavated trenches exposed two parts of the paleo-landscape. Trenches 1–3 suggest an outcome of two types of dynamics at the confluence of the smaller Dani catchment meeting the larger Barepat River catchment. The Dani catchment produced alluvial fans which created smaller-scale braided channels. For example in T1, the main channel width of such a braided fan was ca. 3 m. These channels were most likely originating from the Dani catchments. Flooding from the small and steep Dani catchment most likely represents episodic snow and rain melt overflowing from its mountainous catchment area. The Barepat watershed is larger, contains more springs, and the slope of the river valley is less steep than the Dani (Fig 2). Barepat

derived sediments are generally finer than those from the Dani, indicating different depositional energies and valley geometry in the two catchments. Our pXRF analyses of the excavated sediments document the two watershed sources in the different trenches. Within the sedimentary sequences of our trenches, we attribute the more gravelly/stony sediments and those without Mg in T1 (Units 1a, 3a) and T2 (Units 1a, 2a, 3a) to the Dani. In some cases, given the confluence context, gravelly deposits are a result of simultaneous inflows from both rivers. The T4 sedimentary sequence differs from those in T1, T2, and T3 in that no major gravel unit occurs in T4. The T4 deposits regularly display a medium- to low-/very-low-energy alluvial dynamic or surface runoff, reworking colluvial-alluvial inputs of endogenous rocks from nearby outcrops. The succession of paleosols indicates that the location of T4 was a more stable section of the landscape.

Micromorphological analyses from T1 Unit 1b and T2 Unit 4 indicate weak pedogenesis during periods of morphogenic stability. Lateral migrations of braided channels, as manifested in the gravely units of T1 to T3, reflect short-term deposition events. These dynamics probably indicate that the periods of stasis were short and did not allow the complete development of soils. The intricate contacts between the deposits from Dani catchment with fine-grained deposits of the Barepat and the spatial variations of those sedimentations impede a more refined correlation of sedimentary units among the excavation trenches.

The chronology of sedimentation in T1-T3 based on the pIRIR dates indicates depositional ages of ca. 60–45 ka, that is, at the end of MIS 4 and beginning of MIS 3. T4 Units 1–4 are penecontemporaneous with the formation of the alluvial fans deposits in T1-T3. The paleosol sequence of Units 5a-5d in T4, (ca. 51–36 ka) overlaps with the T1-T3 sedimentary sequence but also accumulated slightly later.

Reconstructing the environments during hominin occupations is complex. For example, paleovegetation proxies differ in spatial representativeness. Pollen reflects the wider region since pollen grains are transported partly by wind, while leaf wax remains represent accumulation within the watersheds of the Dani and/or Barepat Rivers. Sampling in the modern watershed does not show the Dani River significantly incorporating leaf waxes from the alpine vegetation once the stream passes below the treeline. This indicates that, in this setting, leaf waxes as a proxy are not influenced as much by up-stream vegetation changes, but rather by more local vegetation at and around the site (Brittingham et al., in prep). Pollen from the T4 deposits (51–36 ka) suggests an open landscape and unlike the current forested environment of Kalavan today, trees may have been present in deep valley at lower altitudes of the Barepat watershed. Due to the preservation and taphonomic limitations, palynological data from T4 was unable to offer unequivocal support for a treeless landscape. Analyses of $n$-alkanes ($n$C$_{25}$-$n$C$_{35}$) from T1-T3 shows that most samples reflect grassy landscapes or a mix of deciduous and grassy, with one exception (T2 Unit 2b), which is similar to modern deciduous soils.

Future chain length and isotopic analysis of $n$-alkanes ($n$C$_{25}$-$n$C$_{35}$) from T4, in comparison with those from T1-T3, will help to test the suggested environmental reconstruction of these two portions of this landscape. The sampling of present-day soil and stream sediments along the Dani catchment will yield a framework for a detailed comparison of the present-day sediments and the effects of sediments transport on the leaf-wax composition with those sampled from the trenches [23, 126]. The results of these analyses will enable a more nuanced examination of the $n$-alkane variations along topographic gradients in the landscape surrounding Kalavan 2 [128].

The faunal assemblages include a diverse species of ungulate fauna, such as *Bos/Bison*, *Equus* sp., *Equus hemionus*, *Cervus elaphus*, and *Capra/Ovis*. Bovids (*Bos primigenius* or *Bison*) and red deer (*Cervus elaphus*) are quite tolerant of a range of micro-habitats. Caprinae is more prevalent in mountainous and rocky habitats. Horse and onager prefer open steppes and

grassland, and the onager is adapted to rather dry conditions [40, 129]. Such a diversity of habitats may reflect seasonal ecological conditions where specific habitats of different species overlapped at Kalavan 2. Alternatively, the faunal assemblage may be biased by hominin hunting behaviors [130] (see below). In T4, the few faunal remains that could be identified, including those that could be assigned to size class, mainly represent red deer (*Cervus elaphus*) and goat/ibex (*Capra* sp.), while the large size classes of *Bos/Bison* are absent (Tables 13 and 14).

Small mammal bones are dominated by the common hamster, followed by the Turkish hamster, European watervole, common/field vole, and single occurrences of the common shrew and marmot. T4 may indicate intrusive remains of common hamster that died in their burrows during winter, as their burrows can be of considerable complexity and size [131]. Testing intrusiveness could be resolved with 14C dating as demonstrated by Royer et al. [132] and Rofes et al. [133]. T1 and T2 represent predator-accumulated assemblages and are not dominated by the common hamster and provide a more diverse species composition, together with signs of digestion and breakage. These micro-faunal remains in T1 and T2 may, therefore, reflect deposition during the time of the sediment accumulation.

Today the common hamster is extinct in Armenia, but relict populations exist in southern Russia [134]. At the nearby Hovk 1 cave site (2040 m asl), where predator-accumulated assemblages were unearthed, common hamster remains were found in levels from MIS 5 to 4/3. [135]. The European watervole and vole (either common or the field) are absent from Hovk 1 [135] and the Upper Paleolithic (UP) cave site of Aghitu 3 [6]. The common hamster, Turkish hamster, marmot, mole vole, and common vole tend to inhabit steppic or grassland environments [131, 136–139]. The water vole occurs near streams, while the field vole and shrew are associated with moist meadows and forest steppes [140–142]. This taxa combination indicates an open environment with a water body nearby, as supported by our other environmental proxies. Further in-depth study of the common hamsters at Kalavan 2 will also include direct radiocarbon dating to refine their history of accumulation and site formation processes in general.

Regional MIS 3 climatic records are characterized by abrupt temperature and precipitation oscillations over millennia, indicative of changes between cold (stadial) and warm (interstadial) stages [17–22]. During stadial periods, the winters are thought to have been longer and summer droughts more frequent, while there was increased spring and early summer precipitation during interstadial periods [35–37]. Ollivier [143] suggested that these seasonal conditions during interstadials provided greater runoff and sedimentary transfers, which supplied torrential geomorphic components (e.g., pebbely deposits and developed alluvial fans) observed within the Kalavan 2 trenches.

Between 60 and 45 ka, the Kalavan 2 inhabitants repeatedly occupied parts of the local alluvial landscape (i.e., T1 and T2). The formation processes that preserve or distort archeological horizons are also an outcome of similar landscape geomorphological processes [144]. The highest density archaeological horizons were stratigraphically confined within a relatively thin horizon ca. 10–20 cm thick. 92% of the lithic artifacts were derived from three horizons (T1 Unit 1b – 57.1%; T2 Unit 4–19.9% and Unit 7–15.0). Similarly, 74.8% of the faunal NSP were found in two of those horizons (T2 Unit 4–39.5% and T1 Unit 1b - 35.3%). Some archaeological horizons are confined laterally as well (e.g., the horizons in T2 do not continue into T3 five meters apart). Moreover, as can be observed by the differences between T2 Unit 4 and the previous excavations of the same archeological horizon as manifested in the density of finds (both lithic and fauna) as well as the relative frequency of the different raw materials frequencies hint toward a possible lateral changes within a single paleo-surface.

These spatial variations as seen in the different trenches suggest temporal concentration of an anthropogenic non-random distribution of both lithic and faunal remains (Tables 6 and

13). Only a few horizons suggest a secondary redistribution of finds (e.g., the wavy contact at the interface of Units 5 and 6 in T1, with artifacts vertically distributed over ca. 50 cm).

Fauna from T1 Unit 1b and T2 Unit 4 are characterized by high fragmentation, resulting in limited taxon identification. The few anthropogenic modifications of fauna were found in these two units, in which long bone shafts dominate (Table 14). Such a pattern can be explained by three possible scenarios: (1) poor syn- and post-depositional conditions for bone perseveration [145, 146]; (2) bone accumulation as a result of a high hydro-dynamic input [146, 147]; and (3) selective transport of animal elements by hominins [148]. Micromorphological analyses, however, reveal the initial stages of stabilization and no indications of hydro-dynamic sorting. Moreover, lithics < 2 cm in maximal dimensions occur in high frequencies (i.e., T1 Unit 1b: 85.1% of the obsidian, 46.7% of the non-obsidian assemblages; T2 Unit 4: 57.4% of the obsidian, 42.4% of non-obsidian assemblages). In experimental core reduction and tool production assemblages, this size class comprises ca. 60–80% of lithic artifacts [149, 150]. Given that lithics < 2 cm are more susceptible to displacement than larger artifacts [149–151], the high frequencies of small artifacts point to a lack of hydrological size-sorting. Altogether, our datasets suggest that the hydro-dynamic sorting of bone elements is unlikely. Instead, they suggest that these two faunal assemblages from T1 Unit 1b and T2 Unit 4 are a product of hominin activities and provide an opportunity to study their behavioral significance.

## Articulating Kalavan-2 within late MP regional settlement patterns

Kalavan 2 preserves evidence of multiple occupations by MP hunter-gatherers ca. 60–45 ka. The three main horizons in T1 and T2, in addition to the findings of low frequencies of lithic scatters at ten sedimentological units, as well as the sporadic lithic artifacts found in four more sedimentological units dug by the previous test trenches [10], suggest that the occupations at Kalavan 2 locale were repeated. In this paper, we aimed to test hypotheses involving elevational mobility at Kalavan 2, yet reconstructing the timing of the site's occupations at a seasonal scale is challenging. Micromorphological evidence suggests that, like today, the area suffered from winter freezing conditions, but there are too few faunal dental remains to conduct meso and microwear analyses to establish seasonality reconstruction. Higher elevations sites such as Kalavan 2 experienced frost conditions as well as heavy snow cover during winter. Those conditions most likely would prohibit the accessibility of the area during this time of the year. Thus, Kalavan 2 will most likely have been occupied mainly during the rest of the year and for shorter periods. Higher elevation sites such as Kalavan 2 experienced frost conditions and heavy snow cover during winter. Such conditions would most likely prohibit access to the area during this time of the year. Thus, Kalavan 2 will most likely have been occupied mainly during the rest of the year and for shorter periods. Fortunately, other behavioral proxies at the site allow us to consider occupational duration, technological organization, mobility and subsistence strategies, and the sites' role in regional settlement systems.

From a lithic economy point of view, the artifact assemblage has two complementary facets. In all units, obsidian artifacts outnumber non-obsidian ones (with the notable exception of T2 Unit 4; however, the earlier excavations of the same unit found more obsidian artifacts than non-obsidian ones [10]). The non-obsidian component includes a wide variety of raw materials including basalt, dacite, welded tuff, chert, and limestone. There are indications that early reduction stages of those materials occurred on-site (e.g., the presence of few non-obsidian cores and cortical pieces). However certain non-obsidian artifacts attest to their provisioning of blanks that were knapped off-site into the locale (e.g., a large Levallois point on basalt; Fig 9: 3). [152]. Such examples of artifact transport suggest that non- obsidian core reduction

sequences were also segregated across the landscape [153]. The non-obsidian component also exhibits a greater variety of technological variants, such as flake (non-Levallois and Levallois) and blade production.

The obsidian assemblages include a high frequency of artifacts < 2 cm in maximal dimensions, including high frequencies of shaping flakes including those that removed the tips of retouched points (Fig 8: 1–5). Abundant small flakes occur despite the lack of on-site blank-production indicators (minimal numbers of cores and CTEs). This suggests intense on-site obsidian tool use, maintenance, and reduction compared to non-obsidian artifacts. The shaping flakes are categorized according to the scars on their dorsal face removing the previous phases of retouch (see similar definitions and discussions in [154–156]). Most of the shaping flakes are < 2 cm. It is worth noting that the technological role(s) of shaping flakes were likely diverse and may reflect part of the initial stages of retouch or secondary retouch after use when the edge has become dull, jagged, or uneven (see ethnographic example [157]). These technological signatures reflect maintenance behaviors [158–160] intended to extend tool use-life. Obsidian retouched points and shaping flakes from the site exhibit use-wear, perhaps reflecting activities related to woodworking. These preliminary results suggest a scenario of continuous rejuvenation of retouched points during woodworking. Initial use-wear observations for the obsidian scrapers, in contrast, hint at activities related to processing hard animal materials (bone or antler).

Similar obsidian modes of shaping geared toward the removal and rejuvenation of the tool edges and tips of retouched points have been documented at the MP site of Yerevan 1 cave in its upper sequence [27]. The pieces include both lateral ([27], Fig 14: 5), and distal [27], Fig 14: 9, 15, 19; [4], Fig 6: 9) shaping flakes. Yeritsyan and Semenov ([161]), based on use-wear analysis, suggested that the majority of analyzed obsidian tools were used for a diverse range of activities including hide scraping, woodworking, bone processing, meat processing, and perforating or cutting activities [161]. The main characteristic of the wear on the obsidian pieces is the crushing of the working edges. Both Yerevan 1 cave and Kalavan 2 are located beyond the expected daily foraging distance from the nearest obsidian sources. Obsidian sourcing has not been conducted for Yerevan 1 cave; however, the closest sources are ca. 30–40 km from the site [4]. It is not clear if the shaping techniques observed at both sites reflect raw material scarcity, or were rather related to the mechanical properties of obsidian,that may have required constant rejuvenation of tool edges. A similar scenario was suggested when coarse-grained dolerite was contrasted with finer-grained hornfels at the Middle Stone Age layers at the site of Sibudu, South Africa, where it was suggested that dolerite edges are more durable in comparison to those made on hornfels that demanded constant rejuvenation by retouching [162]. Such suggestions regarding the effects of raw material properties upon the maintenance behaviors at Kalavan 2 will be examined in the future using in-depth experimental, technological, refitting, and comprehensive use-wear analysis.

Sourcing of the non-obsidian raw materials at Kalavan 2 remains in the initial stages; however, obsidian sourcing of MP artifacts by pXRF is well developed in Armenia. Obsidian is not available near Kalavan 2, and the closest sources lie well beyond the expected daily foraging range from the site following the principles of [163, 164]. Their model assumptions are based on ethnographic hunter-gatherer studies that suggest that daily walking distance is confined to ca. 120 minutes of one-way pedestrian travel and in a flat, homogeneous topography, this travel time is equal to a distance of ca. 10 km [163, 164]. However, applying Tobler's [165] formulae for the relationship between slope angle, distance, and walking time to the Armenian highlands topography, results in reduced distances than the commonly suggested [163, 164] of an area of a diameter of ca. 10 km.

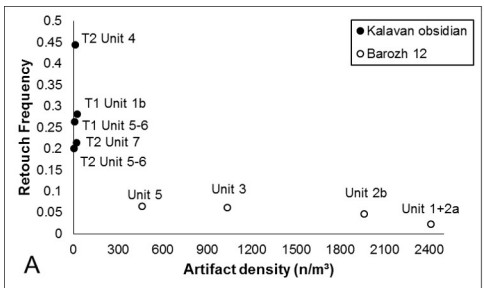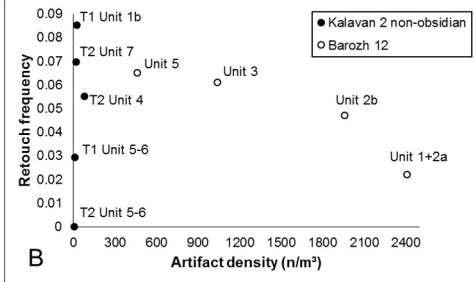

**Fig 17.** Binary plots of artifact density and retouch frequency for sedimentary units at Barozh 12 (open circles) and Kalavan 2 (B). Data exclude artifacts < 2 cm in maximal dimensions to avoid differences among sedimentary units in assemblage preservation and winnowing of small artifacts due to post-depositional processes.

Obsidian sourcing of 928 artifacts, conducted in the Kalavan 2 field lab, revealed that the Geghasar source in the Gegham Range and sources along the Hrazdan River valley, in particular Gutansar and Hatis, were routinely exploited. The Tsaghkunyats sources were utilized to a lesser degree. These source areas are a linear distance of 50–60 km from Kalavan 2 but ≥ 120–140 km on foot. Less frequently, the Syunik highlands sources were exploited as well: 120–130 km from the site linearly and at least ≥ 200 km along the most direct routes. Artifacts from the three distant obsidian sources (Pokr Arteni, Kars-Digor 1, and Chikiani) were found only in T1 Unit 1b. Pokr Arteni is located ≥ 200 km to the west along the most direct routes from Kalavan 2. Further to the west lies Kars-Digor 1 in modern-day Turkey, 160 km linearly, and ≥ 200 km on foot from Kalavan 2. To the northwest of Kalavan 2, the farthest identified obsidian source is Chikiani in Georgia, 150 km linearly, and ≥ 240 km on foot (Fig 13).

This raw material movement suggests contacts with other Late Pleistocene hominins. The source area of Gutansar and Hatis, was the most frequented and was shared with MP occupants of other sites such as Lusakert Cave 1 and Alapars-1 [14, 16] (Fig 12). The source of Pokr Arteni is located 1–2 km from the MP site of Barozh 12 [7, 8, 30]. Chikiani sources to the northwest were utilized at MP and UP occupations in the Imereti Region in the Rioni River basin (southern Caucasus), at a distance > 100 km [5, 43, 166]. The lithic assemblages at Ortvale Klde, a cave site in the Imereti Region dated to MIS 3, show a similar duality in the raw material economy to that observed at Kalavan 2 [167]. Locally available chert quantitatively dominates the Ortvale Klde assemblage, which exhibits a low frequency of retouch, and the few obsidian artifacts were transported a notable distance and appear as highly reduced tools [43, 167].

Barton and Riel-Salvatore [52] and Clark and Barton [53] modeled short-term occupations as lithic assemblages with low artifact densities, high frequencies of retouched pieces, and few cores (however see discussion in [54, 168, 169]). Plotting the frequency of retouched artifacts against artifact density for each unit with the highest artifact abundances at Kalavan 2, we observe that densities range between 20 and 307 artifacts/m$^3$ (Table 6) and the retouch frequencies range between 0.09 and 0.14 (Fig 17A). When these results are compared to those from the chronologically contemporaneous MP site of Barozh 12 (1336 m asl) [7, 8, 30], it is apparent that all of the Kalavan 2 units have very-low-density assemblages with significantly higher tool frequencies (t = 4.9, df = 7, p = 0.002; Fig 17B). Barozh 12 is a locality typified by very high artifact densities and a wide range of tool frequencies among sedimentary units [7, 8, 30]. Based on these data as well as the other lines of evidence discussed above, we propose that the Kalavan 2 assemblages represent relatively short-term occupations with activities that mainly involved tool use, maintenance and discard.

The faunal data may also be used to consider the occupation duration and site function within a wider settlement system. At task-specific sites dedicated to hunting activities, we expect monospecific faunal assemblages containing the non-nutritional bone elements (i.e., an assemblage with a narrow range of hunted animals and a high representation of non-nutritional elements), whereas residential sites should contain a diverse hunted faunal assemblage with meat-rich body parts represented (e.g., upper limbs, axial bones) [55, 57]. Hominins played a major role in the formation of the Kalavan 2 faunal assemblages, as attested by the presence of anthropogenic bone modifications, lack of evidence for carnivore activity. The varied ecological preferences of the present species could be the result of seasonal habitat overlap in the surroundings of Kalavan 2. Alternatively, it may reflect hominin use of different altitudinal zones/elevational ecotones [51] along the mountain flanks and subsequent transport of food to the site.

Our results from Kalavan 2 deviate from the expectations of a short-term hunting site, as indicated by the animal diversity represented in the faunal assemblages and the representation of their body parts. The assemblages are biased toward a high representation of long bone fragments, which are less meat-rich (with notable exception of Unit 4 in T2 [Table 14] and the bison skull found in previous excavation [10]). This pattern could be explained by either (1) post-depositional density-mediated attrition or (2) selective transport of skeletal elements to the site. Selective transport of body parts from primary kill sites to Kalavan 2 is likely given the rather steep landscape where such a behavior would reduce transport costs. If selective transport by hominins cannot be refuted, this may suggest that Kalavan 2 was a secondary meat-processing or marrow extraction locale. However, the questions of whether selected body parts were transported according to their nutritional value still need to be tested [31].

Kalavan 2 adds to a growing corpus of MIS 3/2 sites that can be situated within wider subsistence patterns, especially concerning mobility strategies. At Ortvale Klde (530 m asl) in Georgia, faunal analysis suggests repeated exploitation of migratory herds of Caucasian tur (*Capra caucasica*) in late fall and/or early spring [43, 170]. In Armenia, at Lusakert cave 1 (1420 m asl), ca. 60–35 ka [15], the faunal assemblages consist of mainly *Capra* sp. and *Equus* sp., and a few *Bos/Bison* remains [29]. High densities of lithic artifacts, together with the preservation of hearth features [171], suggest that the site might have been occupied for a longer duration rather than a task-specific locale. At the UP site of Aghitu-3 (1601 m asl), where the main occupation layers are dated to 36–32 and 29–24 ka cal BP, the site occupations are characterized as short, repeated, and seasonal, exploiting mainly *Ovis* and equids in spring or summer based on tooth wear [31]. At the UP site of Kalavan 1, located only 1 km from Kalavan 2, repeated occupations dated to 16 ka cal BP were suggested to have occurred in summer and/or autumn and focused on hunting migrating mouflon (*Ovis orientalis*) based on isotopic and faunal analysis [44, 172, 173].

At sites located at higher elevation, such as the open-air Alapars-1 (1774 m asl), no faunal remains were preserved. The lithic evidence from the layer dated to the end of MIS 4 (ca. 65 ka) was interpreted as demonstrating greater occupational intensity in comparison to the MIS 5 occupation amidst a climatic shift [16]. At the site of Hovk 1 (2040 m asl), sporadic evidence of human occupation over a long period of time ca. 100 ka to 35 ka [11]. Taphonomical analysis of the fauna at Hovk 1 suggested that the fauna accumulated in a natural pitfall/carnivore trap and is not anthropogenic [33].

Among the above mentioned MIS 3 sites, some were assigned a putative role within settlement systems. Ortvale Klde was suggested to function as a residential site where different hunter-gatherer groups aggregated for economic and social purposes. This interpretation is based on the density of anthropogenic sediments and archaeological materials, especially in Layers 7–4 [43]. At Lusakert Cave 1, a similar line of evidence was used to place the site as a

focal point within the settlement system [29, 171]. The contemporaneous site of Barozh 12 was also interpreted as a persistent place in hominin settlement systems based on high find densities [7, 8, 30]. When comparing the find densities and the retouch frequency from Kalavan 2 and Barozh 12 (Fig 17), sharp differences in technological organization and assemblage formation can be observed. At Barozh 12, relatively longer or frequent occupations involved core reduction and blank production alongside tool use and discard [7, 8, 30]. Kalavan 2 artifact assemblages, on the other hand, reflects the opposite end of the continuum: repeated, short-term occupations and a narrower range of activities. The geographic range of the settlement system in which Kalavan 2 occupants moved is illuminated by obsidian sourcing. Lithic procurement behaviors during MIS 3 in this region suggest mainly the exploitation of nearby raw material sources with infrequent transport of non-local material beyond the daily exploitation territory. Non-local obsidians were transported to sites over a range of 50–200 km as the crow flies and up to 300 km as the shortest walking distance [6–8, 14, 16, 174, 175]. Kalavan 2 is among the few MIS 3 sites where obsidian is not locally available (e.g., [4, 12, 27]). The closest sources are at least 120–140 km on foot and the most distant are greater than 200 km away on foot. Such distances appear much greater in comparison with those identified in Mediterranean Europe and the Levant in the same period [176–178]. Among the Mediterranean European and Levantine MIS 3 sites, nearly 90% of the lithic raw materials were derived from daily foraging territories [176–180]. The evidence from Kalavan 2, in contrast, attests to high-mobility lifeways over larger territories.

## Concluding remarks

Kalavan 2 is located in a rugged mountainous landscape where current winter temperatures are below freezing and as shown here, similar conditions were prevalent during the Late Pleistocene. The available resources for hominins within such an ecological niche fluctuated drastically according to the seasons. As such, the site was most likely not occupied during the harsh winter periods. The results of the renewed excavations at Kalavan 2 exposed three main occupation horizons and ten additional low densities of lithic artifacts within different sedimentary units as well as four horizons unearthed by previous expedition [10]. The current results refined the chronological, stratigraphical, and paleoenvironmental framework that enables to contextualize hominin repeated occupations within the mountainous landscape from ca. 60 to 45 ka. The rich behavioral and environmental corpus unearthed at Kalavan 2 enables us to develop hypotheses regarding elevational mobility patterns and the nature of hominin-environmental interactions in a mountainous environment. Evidence from Kalavan 2 as observed in technological organization, resource procurement, and subsistence practices, has illuminated late MP hunter-gatherer lifeways and enables us to place the site within the settlement systems in the Armenian highlands.

Currently, Kalavan 2 is the only open-air site in the region with the preserved fauna and lithic artifacts within a stratified context. The faunal assemblage comprises diverse hunted animals including *Bos*/bison, horse, wild goat/ibex, and red deer, dominated by long bone elements. The identified prey species suggest that Kalavan 2 represents a secondary processing locale, rather than a primary hunting locale. The lithic assemblages contain two main technological components: (1) an obsidian assemblage with narrow techno-typological variation and an emphasis on highly reduced retouched pieces, and use-wear suggests at a limited range of activities including woodworking and hard animal material; and (2) a non-obsidian assemblage suggest some on-site knapping with higher techno-typological variation of both flakes and blades. The geochemical sourcing of the obsidian artifacts largely circumscribe movements of the Kalavan 2 visitors to central Armenia, in particular, the vicinity of Hrazdan River

($\geq$ 120–140 km on foot); however, there are also artifacts from four sources that lie $\geq$ 200 km on foot in different directions. Considered together, these lines of evidence support our interpretation of Kalavan 2 as the site of repeated, ephemeral occupations involving task-specific activities.

These results from Kalavan 2 illustrate that this high-elevation locale played an integral part in late MP settlement dynamics and behavioral adaptations in the rugged landscapes of Armenian highlands.

## Supporting information

**S1 File. Dating methodologies.**
(DOCX)

**S2 File. Sedimentary and paleoenvironmental proxies' methodologies.**
(DOCX)

**S3 File. Use-wear methodology.**
(DOCX)

**S4 File. Faunal analysis methodology.**
(DOCX)

**S1 Fig.** 1: IR50 and pIRIR225 decay curves of the natural luminescence signal obtained from fine-grain polyminerals (sample L-Eva 1681). 2: IR50 and pIRIR225 decay curves of the natural luminescence signal obtained from coarse grain K-feldspars (sample L-Eva 1681). 3: Results of dose recovery tests (pIRIR225 signal). All measured- to given dose ratios deviate < 10% from unity. 4: IR50- and pIRIR225 related g-values obtained on samples L-Eva 1684, 1685 and 1686. 5: Chemical classification diagrams of the glass analytical data obtained from the Kalavan tephras. 6: Glass shard concentrations vs. depth at Kalavan 2 Trench 1 & 2. Grey bars denote 10 cm scan samples and orange bars represent refined 2 cm intervals. 7: Selected chemical bi-plots of non-normalised glass compositional data from visible tephra identified at Kalavan 2. Comparisons are made to the Çekmece Formation derived from Nemrut in the EAVP. Comparative data from Macdonald et al. (2015).
(ZIP)

**S2 Fig.** 1: pXRF results of elemental by trench. 2–9: Micromorphological samples pictures with details on the main micromorphological features: Ob = Obsidian flake; Bo = Bone; Co = Compaction traces; Vo = Void; Mn = Manganese; Bi = Bioturbation; Fm = Frost microstructure; Ve = Veins of Magnesium, gypsum and/or calcite; Rs = Rotation structure. 10: Odd-over-even predominance (OEP) and average chain length (ACL) of n-alkanes from Kalavan-2 (red triangles) compared to modern grass soils (yellow squares) and deciduous soils (green squares). Modern data from Bleidtner et al 2018. 11: n-alkane abundances of samples from the Kalavan-2 ~55kya sedimentary unit (red triangles) compared to modern grass soils (yellow squares) and deciduous soils (green squares). Modern data from Bleidtner et al 2018.
(ZIP)

**S3 Fig. Scatterplot of Zr and Rb, both normalized to Sr to minimize size effects, for a sample of the sourced Kalavan 2 artifacts and the corresponding geological.** Reference specimens. S2 Fig 1 in S2 Fig.
(TIF)

**S1 Table.** 1. Summary of non-normalised glass compositional data obtained from the visible and crypto-tephra preserved in the Kalavan 2 Trench 1, 2 & 4 sequences. Visible tephra are

affixed with T17- numbers whereas the cryptotephra begin T18- or T19. 2. Non-normalised glass major and minor element data.
(ZIP)

**S2 Table.** 1: pXRF results of elements by trench. 2: Micromorphological results. 3: Pollen results.
(ZIP)

**S3 Table. List of studied samples for use-wear.**
(DOCX)

**S4 Table.** 1: Faunal size class categories. 2: Teeth suitable for study for mesowear and dental microwear texture analysis (DMTA). 3: Small mammal postcranial elements per trench. Trench 4 has been subdivided in units due to the richness of post-cranial elements.
(ZIP)

## Acknowledgments

We would like to thank the Kalavan villagers for their help, support, and hospitality: especially the Ghukasyan family for providing us a home away from home. We warmly acknowledge the debt we owe to the students and volunteers that worked with us over the years. We also thank Suren Kesejyan, Hovhannes Partevyan, and Vardan Stepanyan. We express our sincere gratitude to the Institute of Archaeology and Ethnography of the National Academy of Sciences of the Republic of Armenia, and to Pavel Avetisyan, the Director of the Institute for Archaeology and Ethnography, National Academy of Sciences, Republic of Armenia, for his continued support of our research. We would like to thank Dr. C. Hayward at the University of Edinburgh for assistance with the microprobe analyses. We would like to thank Dr. Daniela E. Winkler (Institute of Geosciences, Applied and Analytical Paleontology, Johannes Gutenberg-Universität Mainz, Germany) and Prof. Gildas Merceron (PALEVOPRIM, UMR CNRS 7262, University of Poitiers, France) for their help with the meso and microwear of teeth. We thank the editor Prof. M. Petraglia, Dr. C. Egeland, and an anonymous reviewer for their comments which improved the paper greatly.

AMB and PG dedicate this paper to the memory of their grandmothers Hanna Malinsky and Bessie Singer, who laid the foundation of this project with their spirit and attitude.

## Author Contributions

**Conceptualization:** Ariel Malinsky-Buller, Philip Glauberman, Boris Gasparyan.

**Data curation:** Ariel Malinsky-Buller, Philip Glauberman, Vincent Ollivier, Tobias Lauer, Rhys Timms, Ellery Frahm, Alexander Brittingham, Benno Triller, Lutz Kindler, Monika V. Knul, Masha Krakovsky, Sebastian Joannin, Michael T. Hren, Olivier Bellier, Alexander A. Clark, Simon P. E. Blockley, Dimidry Arakelyan, João Marreiros, Eduardo Paixaco, Ivan Calandra, Robert Ghukasyan, David Nora, Nadav Nir, Ani Adigyozalyan, Boris Gasparyan.

**Formal analysis:** Ariel Malinsky-Buller, Vincent Ollivier, Tobias Lauer, Rhys Timms, Ellery Frahm, Alexander Brittingham, Benno Triller, Lutz Kindler, Monika V. Knul, Masha Krakovsky, Sebastian Joannin, Michael T. Hren, Olivier Bellier, Alexander A. Clark, Simon P. E. Blockley, Dimidry Arakelyan, João Marreiros, Eduardo Paixaco, Ivan Calandra, David Nora, Nadav Nir, Ani Adigyozalyan, Hayk Haydosyan.

**Funding acquisition:** Ariel Malinsky-Buller, Philip Glauberman, Olivier Bellier, Boris Gasparyan.

**Investigation:** Ariel Malinsky-Buller, Philip Glauberman, Vincent Ollivier, Tobias Lauer, Rhys Timms, Ellery Frahm, Alexander Brittingham, Benno Triller, Monika V. Knul, Masha Krakovsky, Sebastian Joannin, Michael T. Hren, Alexander A. Clark, Simon P. E. Blockley, Dimidry Arakelyan, João Marreiros, Eduardo Paixaco, Ivan Calandra, David Nora, Boris Gasparyan.

**Methodology:** Ariel Malinsky-Buller, Philip Glauberman, Vincent Ollivier, Tobias Lauer, Rhys Timms, Ellery Frahm, Alexander Brittingham, Benno Triller, Lutz Kindler, Monika V. Knul, Masha Krakovsky, Sebastian Joannin, Michael T. Hren, Alexander A. Clark, Simon P. E. Blockley, Dimidry Arakelyan, João Marreiros, Eduardo Paixaco, Ivan Calandra, David Nora.

**Project administration:** Ariel Malinsky-Buller, Philip Glauberman, Masha Krakovsky, Robert Ghukasyan, David Nora, Boris Gasparyan.

**Resources:** Ariel Malinsky-Buller, Philip Glauberman, Vincent Ollivier, Boris Gasparyan.

**Software:** Rhys Timms, Ellery Frahm.

**Supervision:** Ariel Malinsky-Buller, Philip Glauberman, Rhys Timms, Masha Krakovsky, Michael T. Hren, Simon P. E. Blockley, Boris Gasparyan.

**Validation:** Ariel Malinsky-Buller, Vincent Ollivier, Tobias Lauer, Rhys Timms, Ellery Frahm, Alexander Brittingham, Benno Triller, Lutz Kindler, Monika V. Knul, Masha Krakovsky, Sebastian Joannin, Alexander A. Clark, Simon P. E. Blockley, João Marreiros, Eduardo Paixaco, Ivan Calandra, Boris Gasparyan.

**Visualization:** Vincent Ollivier, Tobias Lauer, Rhys Timms, Ellery Frahm, Alexander Brittingham, Benno Triller, Lutz Kindler, Monika V. Knul, Masha Krakovsky, Sebastian Joannin, Alexander A. Clark, Simon P. E. Blockley, Dimidry Arakelyan, João Marreiros, Eduardo Paixaco, David Nora, Boris Gasparyan.

**Writing – original draft:** Ariel Malinsky-Buller, Philip Glauberman, Vincent Ollivier, Tobias Lauer, Rhys Timms, Ellery Frahm, Alexander Brittingham, Benno Triller, Lutz Kindler, Monika V. Knul, Masha Krakovsky, Sebastian Joannin, Simon P. E. Blockley, João Marreiros, Eduardo Paixaco, Ivan Calandra, Nadav Nir, Boris Gasparyan.

**Writing – review & editing:** Ariel Malinsky-Buller, Philip Glauberman, Vincent Ollivier, Tobias Lauer, Rhys Timms, Ellery Frahm, Alexander Brittingham, Benno Triller, Lutz Kindler, Monika V. Knul, Masha Krakovsky, Sebastian Joannin, Simon P. E. Blockley, João Marreiros, Eduardo Paixaco, Ivan Calandra, Nadav Nir, Boris Gasparyan.

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
