## [Decision Letter · Decision Letter 0]

6 Nov 2020

PONE-D-20-30812

Short-term occupations at high elevation during the Middle Paleolithic at Kalavan 2 (Republic of Armenia)

PLOS ONE

Dear Dr. Malinsky-Buller,

Thank you for submitting your manuscript to PLOS ONE. After careful consideration, we feel that it has merit but does not fully meet PLOS ONE’s publication criteria as it currently stands. Therefore, we invite you to submit a revised version of the manuscript that addresses the points raised during the review process.

Both reviewers felt that the paper had considerable scientific merit, as do I, but there were some concerns about paper organisation, content and implications. Reviewer 1 points out some repetition in the manuscript, and questions the organisation and content of the introductory and discussion sections. Reviewer 2 requests a more cautious approach to the conclusion that high altitude sites were a key component of Middle Palaeolithic settlement systems, given the slim evidence so far obtained. Both reviewers also make specific points about the site and on particular site descriptions that need to be considered. 

We look forward to receiving your revised manuscript.

Kind regards,

Michael D. Petraglia, Ph.D.

Academic Editor

PLOS ONE

Journal Requirements:

3. We note that Figures 1, 2 and 12 in your submission contain map/satellite images which may be copyrighted.

a. You may seek permission from the original copyright holder of Figures 1, 2 and 12 to publish the content specifically under the CC BY 4.0 license. 

4. Please amend the manuscript submission data (via Edit Submission) to include author Hren M.T.

5. Please amend your authorship list in your manuscript file to include author Micheal Wren.

6. Please amend your list of authors on the manuscript to ensure that each author is linked to an affiliation. Authors’ affiliations should reflect the institution where the work was done (if authors moved subsequently, you can also list the new affiliation stating “current affiliation:….” as necessary).

7. Please include table 17 as part of your main manuscript and remove the individual file. Please note that supplementary tables should remain as separate "supporting information" files.

8. Please include captions for your Supporting Information files at the end of your manuscript, and update any in-text citations to match accordingly. Please see our Supporting Information guidelines for more information: http://journals.plos.org/plosone/s/supporting-information

Reviewers' comments:

Reviewer's Responses to Questions

**Comments to the Author**

1. Is the manuscript technically sound, and do the data support the conclusions?

Reviewer #1: Yes

Reviewer #2: Partly

2. Has the statistical analysis been performed appropriately and rigorously? 

Reviewer #1: Yes

Reviewer #2: N/A

3. Have the authors made all data underlying the findings in their manuscript fully available?

Reviewer #1: Yes

Reviewer #2: Yes

4. Is the manuscript presented in an intelligible fashion and written in standard English?

Reviewer #1: Yes

Reviewer #2: Yes

5. Review Comments to the Author

Reviewer #1: The paper is a detailed presentation of a range of data from excavation of an important site in the Armenian highlands. It adds significantly to emerging data on hominin occupation of the region and contributes to knowledge of hominin mobility patterns. Impressively, the paper provides a wide range of data including detailed lithic and faunal analyses, obsidian sourcing (which in my opinion, clearly shows mobility of early hominin populations in the southern Caucasus), sedimentology/micromorphology and chronostratigraphy. To a certain extent, the paper also provides paleoenvironmental data from pollen and leaf wax analyses (and seasonality from dental wear of ungulates).

I believe that the paper is suitable for PLOS ONE and I recommend its publication. However, I believe that significant portions of the manuscript should be rewritten to improve on clarity, particularly the Introduction and the Discussion. The Discussion repeats, almost verbatim, statements that were already made in the Results section.

I have marked some of these sections on the attached PDF (as there are no line numbers in the manuscript). I’ve also made numerous comments in the manuscript. In addition to these comments and suggestions, I have these additional notes:

Rework the introduction (following the notes). Provide more archaeological context to the Kalavan 2 for instance by providing a survey of sites in the region as well as available paleoenvironmental proxies.

Rephrasing/elaboration of certain points (detailed in the text) could improve the paper, especially with regards to the “elevational mobility systems” which I presume refers to high-altitude mobility patterns.

The luminescence dates for the site more or less correspond to the tephrachronology, although it has to be emphasized that the exact sources of tephra in Kalavan 2 cannot be ascertained. What about the inverted dates in Trench 3 (Fig 2)?

The stratigraphic sections (Figures 4-7) perhaps could also indicate which exact sedimentary layers yielded the artefacts and faunal remains.

Just a suggestion (a matter of personal taste to be honest) but I believe it will be more effective to have a graphical representation of the lithic artefacts as well as the faunal remains (i.e., types/taxa per layer). The same for the density of materials recovered per unit (in case the 3D coordinates were recorded, a 3D distribution map will also be more effective in showing the ‘ephemeral’ nature of site occupation).

Figures for the paleovegetation proxies?

Table 17 is missing in the manuscript.

In several parts of the article (i.e. introduction), the authors emphasized that different altitudes would have different environments. Given that most of the sediment from the site were transported (i.e. by the river streams well detailed in the results), how confident are the authors in the interpretation the leaf wax/pollen as paleoenvironment proxies? I’m not sure this was clearly discussed in the paper.

Reviewer #2: This is an impressive multidisciplinary study, and I would support publication. However, I think the preliminary nature of most of the analyses preclude such sweeping conclusions. Please see the attached file for detailed comments.

6. PLOS authors have the option to publish the peer review history of their article (what does this mean?). If published, this will include your full peer review and any attached files.

Reviewer #1: No

Reviewer #2: No

---

## [Author Response · Author response to Decision Letter 0]

29 Dec 2020

We thank the reviewers and the editor, Dr. Eegland, and the additional reviewer for their detailed comments on the paper. We incorporated all the English suggestions made by the second reviewer. 

Regarding journal requirements: 

a. In regrad to the PLOS ONE submissions requirements for paleontology and archaeology research we coply with the ethical and legal requirments. We added the number of permits obtained for all aspects of the study, including the full name of the issuing authority and the permit number. 

b. In addition and in order to compy with the journal policy to make all data necessary to replicate their study’s findings we followd the common phrasing in many publications in the journal. We wrote: “The archaeological material presented in this study is stored at the Institute of Archaeology and Ethnography, Republic of Armenia in Yerevan. The material is available for study with authorization from the Institute of Archaeology and Ethnography, Republic of Armenia in Yerevan. All other relevant data are within the manuscript and its Supporting Information files.”

c. Mapping for figures 1 and 12 was carried out using Esri ArcGIS 10.6. License rights for Esri ArcGIS 10.6 software was provided by Leibniz archeological institute, Mainz. Figures 1 and 12 are containing information from ArcGis online basemaps, which terms of use and permissions are coming in the Master Agreement between the customer (Leibnitz archeological institute) and Environmental Systems Research Institute, Inc. (ESRI). We adjunct the Master Agreement. Figure 2 was products by D. Arakelyan from the Institute of Geological Sciences, National Academy of Sciences of the Republic of Armenia, Yerevan, Armenia, and it is based on an unpublished regional geologic map .

Below is our detailed response (in purple) to the critiques and points raised (in black). The lines in the revised manuscript are now numbered to facilitate following the changes made based on the suggestions of the commentators. 

The abstract was changed in light of Reviewer 1 comments and these sentences were added:

Renewed excavations at Kalavan 2 exposed three main occupation horizons and ten additional low densities lithic and faunal assemblages. The results provide a new chronological, stratigraphical, and paleoenvironmental framework for hominin behaviors between ca. 60 to 45 ka. The evidence presented suggests that the stratified occupations at Kalavan 2 locale were repeated ephemerally most likely related to hunting in a high-elevation within the mountainous steppe landscape. 

The editor and the two reviewers commented regarding the structure and organization of the introduction as well as the lack of background regarding the regional Paleolithic record. Accordingly, the introduction was rewritten and a paragraph was added concerning the current knowledge of the settlement system in the region (lines 81-102). 

Reviewer 1 commented in the introduction: “What are these challenges and what are the opportunities? The authors should elaborate more as these would provide contexts to the arguments (i.e. implications of high-altitude occupation during the MP)” and “What are the coping strategies and what are the ecological risks they mitigate? And also what are the ecological constraints?”. 

We rephrased the first paragraph of the introduction according to those comments (68-79)

Strong seasonal fluctuations as manifested in precipitation and temperature occur across elevations that range from sea level to more than 5000 m asl [1]. As a result, the temporal and spatial accessibility of resources across this terrain sharply fluctuated throughout the year. Those ecological challenges are manifested in the seasonal depletion of resources or opportunities such as animal seasonal migration, most likely impacted hunter-gatherer mobility patterns and lifeways. Hunter-gatherers' decisions regarding their mobility took also in consideration the social factors such as finding mating partners and maintaining demographic viability [2]. The archaeological record preserves an archive that echoes past cumulative decision-making regarding hunter-gatherers mobility and land-use as coping strategies to mitigate those ecological and social risks in order to adapt to specific environmental constraints [3].

Moreover, the specific ecological challenges of this terrain are detailed in the first paragraph of the section “Testing elevational mobility strategies”. 

Reviewer 1 commented regarding the last sentence of the introduction: “You mean mobility patterns in high elevation? or mobility patterns in different elevations?” – We rephrased the sentence for clarity (124- 126):

This record, therefore, enables us to test hypotheses regarding elevation-dependent seasonal mobility and subsistence strategies and the nature of hominin-environmental interactions in a mountainous environment. 

In the section “Testing elevational mobility strategies” Reviewer 2 asked: “I understand what you are trying to say here, but I'm not sure 'social resources' is the appropriate term...” We reformulated the sentence and a reference (lines 148-150): 

Thus, decision-making regarding mobility and settlement strategies likely took into account the variable spatio-temporal access to subsistence resources and/or to maintain social networks [45,47,48]. 

In line 188-190, reviewer 1 asked: “what are these habitats? And I presume this also refers to the current environment/climatic conditions.”. We changed the sentence to explain better the meaning of elevational ecotone as a result of diverse biota and fauna along different elevations (lines 173- 177). 

At 1640 m asl, Kalavan 2 is currently situated in a deciduous-mixed forest below the upland meadow-grassland, within an elevational ecotone [51]. Such an environment is located at the interface of altitudinal zonation of flora and seasonal migration of fauna (see above). This setting provides an ideal case study for integrating high elevation adaptations within hunter-gatherer`s settlement systems.

In line 181-183, Reviewer 1 commented: 

“I'm not sure I completely agree. Short term occupation sites (especially those occupied for several months/seasons) could also show selective transport/over-representation of high meat/fat-yielding animal parts.” and “So by 'ephemerally occupied, task-specific sites' are you referring to kill sites? Perhaps rephrase? Or are they seasonal camps occupied by a group of hominins/humans for several months/seasons regularly/semi-regularly (i.e., every year)? Cause I'm sure this will give very different signatures in terms of lithics distribution and animal body part representation.” 

We rephrase the dichotomy between residential and task-specific sites by explaining that task-specific we mean for example hunting sites. Hunting sites may provide a more specific signature than butchery that was written previously. 

Residential sites reflect long-term occupations with varied types of activities, whereas task-specific sites refer to short-term occupations with restricted sets of activities such as hunting kill site.

Reviewer 2 wrote in line 4074-409 that “I'd be cautious with this interpretation as you are probably looking at a layer accumulated through time”. However, the observation of the pattern seen in the thin-section is a common feature (see reference 67), that preserve occasionally in high-resolution cases such as in Kalavan 2. 

Reviewer 1, commented in lines 511-513 that we should declare in the first part of the tephra analysis that there is not precise identification of the sources of tephra at the site. We disagree with the reviewer and the tephra analysis is a part from a much larger attempt to provide a regional tephra chrono-stratigraphy and the results presented in this paper provide a building block in forming such a framework. 

In line 600 – reviewer 1 suggest to change traceology to Use Wear Analysis- agreed

Additional data related to the artifacts cleaning procedure that were requested by reviewer 2 were added (lines 604-609).

In line 624, reviewer 1 asked about the need to provide experimental specimens or published studies. The observations reported in the paper are based on Hurcombe (1992) and Walton (2018) published reference collection as is common in the field.

Following the suggestion of reviewer 1, we replaced figure 11 and added a picture of the field context of the pitted anvils. 

In line 667-674, reviewer 1 asked: “what does high knapping quality mean? more homegenous i.e. less inclusion?”. Therefore, we change the text: The knapping quality of the obsidian (e.g., glassiness, lack of mineral phenocrysts) does not differ among the identified sources. Thus, the variation in the raw material abundance does not reflect a preference based on flaking properties or mechanical performance. Instead, it reflects mobility patterns coupled with spatiotemporally distributed production and discard of different tools as needed.

In line 743, reviewer 1 asked: “But what about the results of the microwear? Can you say something about the environment? I assume that the presence of Equus suggests presence of grassland/grazing. What about the signatures in the bovids? Are they the same in terms of pits and striations with those in equids?” As detailed in S4 Table 2, only five samples per method are well-enough preserved to be analyzed (and only 5 are common for both methods). These samples represent 3 species (1 Cervus Elaphus, 2 Equus hemionus, and 2 Bos/Bison) spread across several trenches/units. This means that only one or two samples per species are available for analysis, not even considering the trenches/units. This is not enough to provide meaningful results. The current sample size is not sufficient to produce meaningful results based on these methods. Thus, the raw data were not analyzed.

In line 754 reviewer 1 requested to provide photomicrographs (if any) or taphonomic counts. We assert that the methodology for identification of the taphonomic characteristics are fundamental to all specialist working in the field as described in the reference quoted. A detailed paper regarding the micro-faunal remains is under preparation with more in-depth descriptions and documentation. 

In line 786-787 reviewer 1 state implication (i.e. be more specific). What does the pollen record show in terms of paleoenvironment? We added a sentence in lines 786-787: Recorded taxa characterized an open steppic landscape. The implications and limitation of the pollen results are stated in lines 862- 866: Pollen from the T4 deposits (51-36 ka) suggests an open landscape with a forest at lower altitudes, unlike the current forested environment of Kalavan today. Due to the preservation and taphonomic limitations, palynological data from T4 was unable to offer unequivocal support for a treeless landscape.

Reviewer 1 suggested to add “a figure showing the 3D distribution of the artefacts and faunal remains will be better, or if the 3D coordinates were not recorded, perhaps graphs could be presented instead tables. It will be easier to see the distribution patterns, especially which layers yielded the artefacts/faunal remains”. Similarly, reviewer 2 wrote in the discussion lines 919-923: “I'm not sure that this can be said without showing a 3D distribution maps of the finds? Or are you referring to temporal (i.e. stratigraphical) concentration? Then I agree. Perhaps a figure (as suggested earlier) would be helpful to clearly show this.” 

We agree with the reviewer's suggestions that it would be more approachable to demonstrate the vertical distribution in a figure rather than tables (although figs 4-7 show the vertical thickness of the sedimentary units). However, in the current stage of the research of the site, where both the in-depth analysis of the faunal and the lithic are still in progress. These studies require further spatial studies including reffiting of bones and lithics as well as the incorporation of the non-coordinate into the analysis, mainly the smaller fraction that are not recorded by the total stations. Further in-depth spatial analysis is underway and therefore, we assert that such a figure is premature. In order to specify that we mean temporal (i.e. stratigraphical) concentration we added a clarification sentence in lines 925- 935: 

The highest density archaeological horizons were stratigraphically confined within a relatively thin horizon ca. 10-20 cm thick. 92% of the lithic artifacts were derived from three horizons (T1 Unit 1b – 57.1%; T2 Unit 4 -19.9% and Unit 7- 15.0). Similarly, 74.8% of the faunal NSP were found in two of those horizons (T2 Unit 4 - 39.5% and T1 Unit 1b - 35.3%). Some archaeological horizons are confined laterally as well (e.g., the horizons in T2 do not continue into T3 five meters apart). Moreover, as can be observed by the differences between T2 Unit 4 and the previous excavations of the same archeological horizon as manifested in the density of finds (both lithic and fauna) as well as the relative frequency of the different raw materials frequencies hint toward a possible lateral changes within a single paleo-surface. 

In the discussion, we followed the suggestions of reviewer 1 regarding possible repetitions and deleted and rephrase many parts of the discussion. 

In line 860 reviewer 1 stated: “In several parts of the article (i.e. introduction), the authors emphasized that different altitudes would have different environments. Given that most of the sediment from the site were transported (i.e. by the river streams well detailed in the results), how confident are the authors in the interpretation the leaf wax/pollen as paleoenvironment proxies? I’m not sure this was clearly discussed in the paper.” And later: “In other words, are you sure that the pollen and leaf waxes you are looking at are not transported and therefore not really representative of the local environment.” 

The samples taken for pollen and leaf wax were taken according to stratigraphical and chronological framework and are studied accordingly. Thus, those samples are time-averaged, but in confined stratigraphical order. Therefore, the analyzed proxies is studied according to their stratigraphical and chronological ordering. Sediment transport of the leaf wax signal is indeed a concern that is currently under investigation with a manuscript in preparation from one of the co-authors on this paper (Brittingham et al., in prep). Sampling in the modern watershed does not show the Dani River significantly incorporating leaf waxes from the alpine vegetation once the stream passes below treeline. This indicates that, in this setting, leaf waxes as a proxy are not influenced as much by up-stream vegetation changes, but rather by more local vegetation at and around the site. These sentences were added to the text lines 858- 862.

Reviewer 2 commented that in Table 18, is it possible to have 101% grass representation? The results of the table are based on an equation published by Bliedtner (et al 2018), where the grass% estimate is derived (): “The observed scattering of the individual samples around the degradation lines, potentially leading to negative or >100% values when calculating grass/herb percentages, could be caused by species-specific and environmental variability: even within one species, environmental stress like different temperatures, moisture conditions, radiation levels, shading or soil nutrient availability can cause some variation of the n-alkane synthesis”. They also address this in a follow-up paper, where this equation was applied to a loess-paleosol sequence in Armenia (Trigui et al 2019): “Grass percentages that exceed 100% illustrate the uncertainties and limitations of the end-member modeling approach because percentages are just semi-quantitative estimations.” 

Line 895 reviewer 1 wrote: “Intriguingly, despite the manner of accumulation differences, the spectrum of microfaunal species is similar among all trenches. “ We rephrased this sentences (895-899)” T1 and T2 represent predator-accumulated assemblages and are not dominated by the common hamster and provide a more diverse species composition, together with signs of digestion and breakage. These micro-faunal remains in T1 and T2 may, therefore, reflect deposition during the time of the sediment accumulation. 

Line 1032 reviewer 1 wrote: “Wouldn't the first statement of this paragraph contradict later elaboration? i.e., if you have limited bone element identification, how confident would your interpretation of carcass transport activities by hominins be? or are you refering to limited taxon identification?” we rephrased (941- 958): 

Fauna from T1 Unit 1b and T2 Unit 4 are characterized by high fragmentation, resulting in limited taxon identification. The few anthropogenic modifications of fauna were found in these two units, in which long bone shafts dominate (Table 14). Such a pattern can be explained by three possible scenarios: (1) poor syn- and post-depositional conditions for bone perseveration [145,146]; (2) bone accumulation as a result of a high hydro-dynamic input [146,147]; and (3) selective transport of animal elements by hominins [148]. Micromorphological analyses, however, reveal the initial stages of stabilization and no indications of hydrodynamic sorting. Moreover, lithics < 2 cm in maximal dimensions occur in high frequencies (i.e., T1 Unit 1b: 85.1% of the obsidian, 46.7% of the non-obsidian assemblages; T2 Unit 4: 57.4% of the obsidian, 42.4% of non-obsidian assemblages). In experimental core reduction and tool production assemblages, this size class comprises ca. 60-80% of lithic artifacts [149,150]. Given that lithics < 2 cm are more susceptible to displacement than larger artifacts [149–151], the high frequencies of small artifacts point to a lack of hydrological size-sorting. Altogether, our datasets suggest that the hydro-dynamic sorting of bone elements is unlikely. Instead, they suggest that these two faunal assemblages from T1 Unit 1b and T2 Unit 4 are a product of hominin activities and provide an opportunity to study their behavioral significance.

Reviewer 2 asked in regards to lines 940-952: “The authors argue that there was no hydrodynamic sorting of the lithics or the bones but, unlike the lithics, they provide no data on the size distribution of the faunal remains to support that claim. Were data on the orientation and inclination of the artifacts or faunal remains recorded?”. This information regarding the faunal assemblage’s orientation, size distribution and mainly weight comparison between different clasts and possible effects of water action are all still under investigation and future publications will present this information. 

In line 1034, reviewer 2 asserted: I'm not sure the data supports this statement, i.e., repeated. What about 'multiple occupations around ca. 60-45 ka.' Similarly, reviewer 1 wrote: “I think the authors’ concluding statement, that “[t]hese results from Kalavan 2 illustrate that high-elevation locales played a pivotal and integral role in late MP settlement dynamics and behavioral adaptations in the rugged landscapes of [the] Armenian highlands” is probably overstated. Do short-term occupations of one site imply a “pivotal role”? Even if we include other high-altitude sites in the area (e.g., Hovk 1), it seems premature (however reasonable) to conclude that such occupations were a key component of MP settlement-subsistence systems. I would refer to this as a working hypothesis rather than a broadly applicable conclusion.” We rephrase and expanded this statement to clarify our intentions (960-964): Kalavan 2 preserves evidence of multiple occupations by MP hunter-gatherers ca. 60-45 ka. The three main horizons in T1 and T2, in addition to the findings of low frequencies of lithic scatters at ten sedimentological units, as well as the sporadic lithic artifacts found in four more sedimentological units dug by the previous test trenches [10], suggest that the occupations at Kalavan 2 locale were repeated.

In line 1098, reviewer 1 asked: Is butchery (i.e. processing of hide and flesh) considered as processing of "hard animal materials"? It is common to refer to hard animal material as a comprehensive term including either bone or antler. As usually, most specialists cannot distinguish between the two raw materials, thus using a more comprehensive term.

---

## [Editor Report · Decision Letter 1]

6 Jan 2021

Short-term occupations at high elevation during the Middle Paleolithic at Kalavan 2 (Republic of Armenia)

PONE-D-20-30812R1

Dear Dr. Malinsky-Buller,

We’re pleased to inform you that your manuscript has been judged scientifically suitable for publication and will be formally accepted for publication once it meets all outstanding technical requirements.

Kind regards,

Michael D. Petraglia, Ph.D.

Academic Editor

PLOS ONE
---

## [Editor Report · Acceptance letter]

14 Jan 2021

PONE-D-20-30812R1 

Short-term occupations at high elevation during the Middle Paleolithic at Kalavan 2 (Republic of Armenia) 

Dear Dr. Malinsky-Buller:

I'm pleased to inform you that your manuscript has been deemed suitable for publication in PLOS ONE. Congratulations! Your manuscript is now with our production department. 

Kind regards, 

on behalf of

Professor Michael D. Petraglia 

Academic Editor

PLOS ONE